# Deep Generative Spatiotemporal Engression for Probabilistic Forecasting of Epidemics

**Rajdeep Pathak**                                                    *rajdeep.pathak@sorbonne.ae*
**Tanujit Chakraborty**                                               *tanujit.chakraborty@sorbonne.ae*
*SAFIR, Sorbonne University Abu Dhabi, United Arab Emirates*
*SCAI, Sorbonne Université, Paris, France*

**Reviewed on OpenReview:** *https://openreview.net/forum?id=7AfAztCd5A*

## Abstract

Accurate and reliable forecasting of epidemic incidences is critical for public health preparedness, yet it remains a challenging task due to complex nonlinear temporal dependencies and heterogeneous spatial interactions. Often, point forecasts generated by spatiotemporal models are unreliable in assigning uncertainty to future epidemic events. Probabilistic forecasting of epidemics is therefore crucial for providing the best or worst-case scenarios rather than a simple, often inaccurate, point estimate. We present deep spatiotemporal engression methods to generate accurate and reliable probabilistic forecasts on low-frequency epidemic datasets. The proposed methods act as distributional lenses, and out-of-sample probabilistic forecasts are generated by sampling from the trained models. Our frameworks encapsulate lightweight deep generative architectures, wherein uncertainty is quantified endogenously, driven by a pre-additive noise component during model construction. We establish geometric ergodicity and asymptotic stationarity of the spatiotemporal engression processes under mild assumptions on the network weights and pre-additive noise process. Comprehensive evaluations across six epidemiological datasets over three forecast horizons demonstrate that the proposal consistently outperforms several temporal and spatiotemporal benchmarks in both point and probabilistic forecasting. Additionally, we explore the explainability of the proposal to enhance the models' practical application for informed, timely public health interventions. Code is available at `https://github.com/PyCoder913/stengression`, and the `stengression` Python package offers an end-to-end implementation of our proposed approaches.

## 1 Introduction

Epidemics have persistently posed one of the most serious threats to global public health, societal stability, and economic sustainability. Throughout history, infectious diseases have periodically swept through populations, leaving behind devastating consequences. For instance, the bubonic plague or Black Death in the 14th century claimed nearly one-third of Europe's population (Glatter & Finkelman, 2021). The Great Influenza epidemic or Spanish Flu of 1918 (Trilla et al., 2008) infected about one-third of the world's population and caused over 50 million deaths. More recent events such as the 2003 SARS epidemic (Cherry & Krogstad, 2004), the 2014-2016 West African Ebola outbreak (Houlihan et al., 2017), the 2015 Zika epidemic (Ferguson et al., 2016), and the COVID-19 pandemic (Chakraborty & Ghosh, 2020) have demonstrated that even in an era of advanced science and communication, humanity remains vulnerable to fast-evolving pathogens that can overwhelm healthcare systems, disrupt economies, and destabilize societies. The increasing pace of globalization, climate change, and human mobility has intensified the spatial reach and frequency of such outbreaks. Hence, epidemic forecasting is not merely an academic pursuit but an indispensable component of public health preparedness and response.

The imperative to anticipate epidemic dynamics has driven the development of forecasting models essential for strategic resource allocation and the formulation of interventions. A plethora of forecasting models are available in the literature, and a vast majority of them can generate point forecasts based on the data of past epidemic incidence cases (Panja et al., 2023). However, public health officials can take better decisions with an estimate of the uncertainty in the forecasted target rather than solely relying on point forecasts (Ioannidis et al., 2022). Due to this, probabilistic forecasts are more desirable for high-stakes decision making (Rumack et al., 2022). A probabilistic forecast quantifies predictive uncertainty by providing prediction intervals or selected quantiles around a point forecast, capturing the range of plausible future outcomes. Epidemics are driven by highly nonlinear interacting processes, including pathogen mutation rates, spatiotemporal mobility patterns, and rapid shifts in human behavior following non-pharmaceutical interventions. It is therefore important for an epidemic forecaster (also for a wide variety of other applications) to produce spatiotemporal probabilistic forecasts, to inform public health officials for tailored responses to outbreaks of infectious disease.

Previous works on epidemic modeling generally bifurcate into mechanistic (compartmental) frameworks, which simulate the physical laws of disease transmission (Kermack & McKendrick, 1927), and phenomenological approaches, which utilize historical data to forecast future trends without explicit biological assumptions. Early approaches to forecast epidemics were predominantly temporal, focusing on time series forecasting of incidence data (Benvenuto et al., 2020). More recently, machine learning techniques such as Recurrent Neural Networks (RNNs) and Long Short-Term Memory (LSTM) networks (Hochreiter & Schmidhuber, 1997) have emerged as powerful nonlinear approximators capable of learning intricate patterns from epidemiological data (Datilo et al., 2019). However, most of these methods are suitable for producing point forecasts, which is just a scalar value for each target and has no measure of uncertainty. Model-agnostic methods such as conformal prediction (Shafer & Vovk, 2008) are often used to create post-hoc prediction intervals from a trained time series model (Barman et al., 2025; Panja et al., 2026). However, they require exchangeability of data and rely on holdout calibration set for calculation of non-conformity scores. Current probabilistic time series models often depend on parametric distributions (Salinas et al., 2020; Herzen et al., 2022), discrete conditional percentiles via quantile regression (Jensen et al., 2022), or computationally exhaustive Markov Chain Monte Carlo (MCMC) sampling for Bayesian inference (Klein et al., 2023). Crucially, these purely temporal architectures overlook the spatial factors of epidemic spread. By isolating single geographical units, they fail to model the complex diffusion of diseases across networks, effectively ignoring the spatial transmission pathways.

To overcome this limitation, recent works are focused on spatiotemporal modeling, which accounts for temporal dynamics under spatial dependence. Spatiotemporal models represent the epidemic process as a dynamic system evolving over a network or a spatial lattice. The Space-Time Autoregressive Moving Average (STARMA) model (Pfeifer & Deutrch, 1980) provides a statistical framework for modeling spatiotemporal dependence, expressing the current state of a region as a function of its own past and that of its spatial neighbors. More recently, Graph Neural Networks (GNNs) have gained significant traction in modeling and forecasting epidemics (Liu et al., 2024) and spatial time series in general. Models such as Graph-Structured RNN (GSRNN; Li et al. (2019)), Cross-location Attention based GNN (Cola-GNN; Deng et al. (2020)), and Epi-GNN (Xie et al., 2022) have combined GCNs and RNNs for effectively forecasting infectious diseases such as the COVID-19 and influenza. However, a significant portion of current spatiotemporal literature focused on epidemics (Barman et al., 2025; Deng et al., 2020) remains restricted to point estimation. The existing probabilistic models in the spatiotemporal domain are predominantly tailored for climate and energy sectors, where high-frequency, high-volume data (e.g., sub-hourly intervals) are readily available. Notable examples include Spatio-temporal Neural Network (STNN) for wind-speed forecasting (Liu et al., 2020), DiffSTG for traffic and air quality modeling (Wen et al., 2023), and spatiotemporal echo-state networks (STESNs) for wind power prediction (Huang et al., 2022). Conversely, epidemic datasets typically exhibit much lower temporal granularity - ranging from daily or weekly to monthly observations, and are often constrained by a paucity of data points. Moreover, the operational utility of models like DiffSTG and STESN is constrained by the high computational overhead of their sampling mechanisms, which renders the generation of forecast ensembles computationally expensive during inference. Thus, there is a critical need for spatiotemporal models for data-driven epidemic forecasting that performs well on low-frequency data, are lightweight, and offer model-intrinsic uncertainty quantification through a probabilistic framework.

This work focuses on building spatiotemporal frameworks based on deep generative architectures that can produce accurate and reliable probabilistic forecasts for low-frequency epidemic datasets. This is done via integrating the idea of pre-additive noise mechanism, a spatial module, temporal module (with LSTM as the base model), and energy score loss. Traditional point-based post-additive noise models ($Y = g(X) + \eta$, discussed in Sec. 3) generally focus on the conditional mean estimation, operating under the restrictive assumption that error distributions are symmetric (typically Gaussian) and centered around the mean. Engression, a neural network-based distributional regression methodology (Shen & Meinshausen, 2025), vouches for pre-additive noise models and learns to sample from the training data distribution via a transformation of a simple distribution. Building on this, we introduce three probabilistic frameworks for spatiotemporal forecasting of epidemics, namely *Multivariate Engression Network* (MVEN), *Graph Convolutional Engression Network* (GCEN), and *Spatio-Temporal Engression Network* (STEN). The MVEN architecture is purely temporal and based on LSTM-engression (Kraft et al., 2026), comprising LSTM networks processing noise-perturbed inputs. In contrast, STEN and GCEN are probabilistic spatiotemporal frameworks that can capture both temporal dynamics and spatial correlation structures. STEN's spatial module relies on prior knowledge of the spatial connectivity in terms of a distance-based weights matrix. This approach produces interpretable spatial embeddings by capturing the linear trajectory among the spatial locations. The graph convolution operations of GCEN offers higher flexibility in modeling the dynamic, complex, nonlinear spatial dependencies across several spatial units.

Our proposed models function as distributional lenses; multiple 'plausible' multi-step-ahead forecast trajectories can be sampled from the trained models to form a forecast ensemble, thereby quantifying the uncertainty of the disease dynamics. By integrating uncertainty quantification as an intrinsic feature of the architectures, the proposed frameworks provide prediction intervals (PIs) without the need for independent, model-agnostic wrappers like conformal prediction, or the prohibitive computational overhead inherent in Bayesian posterior sampling. Moreover, generating PIs directly from the forecast ensemble eliminates the necessity for external calibration using a holdout set. We further investigate the asymptotic properties of the spatiotemporal engression processes by formulating them as closed-loop Markov chains, demonstrating their geometric ergodicity and asymptotic stationarity under mild assumptions. This theoretical guarantee yields significant practical implications for reliable long-term epidemic forecasting. Finally, we conduct an explainability analysis by understanding the internal model dynamics; we explore how the latent stochasticity induced by the pre-additive noise propagates to generate a diverse ensemble of plausible forecast trajectories. Additionally, the spatial architecture of the proposed STEN inherently offers interpretability by explicitly quantifying the contribution of individual spatial lags to local epidemic incidence. This yields practical and actionable evidence for public health interventions, clearly separating the influence of ongoing local patterns from cross-regional spread.

To validate our models, we consider six diverse epidemic datasets consisting of airborne and vector-borne infectious diseases with daily, weekly, and monthly granularities. Each spatiotemporal dataset consists of disease incidence cases for all the states or prefectures for the respective countries with varying sample sizes. The proposed deep spatiotemporal engression methods, viz. MVEN, GCEN, and STEN are applied to these low-frequency datasets to produce multi-step-ahead probabilistic forecasts. These lightweight architectures are compared to existing spatiotemporal probabilistic architectures such as Fast Gaussian Process (GpGp), DiffSTG, and STESN, that are extremely computationally heavy. When compared to other distinct temporal and spatiotemporal state-of-the-art methods using multiple point and probabilistic performance metrics for different forecast horizons, the proposals either outperform the benchmarks, or remain highly competitive. Thus, deep spatiotemporal engression offers model-intrinsic uncertainty quantification on spatiotemporal data with theoretical guarantees, and consistently improves not only the prediction intervals, but also predictive accuracy for epidemic datasets.

Our main contributions can be summarized as follows.

1. Existing architectures such as GCN-LSTM and STARMA-based networks are predominantly deterministic, and probabilistic methods like LSTM-engression are traditionally confined to pure temporal sequences. This work introduces the first spatiotemporal deep generative frameworks - MVEN, GCEN, and STEN, that integrate a pre-additive noise mechanism with energy score optimization for

probabilistic forecasting, which is highly relevant for epidemic series. Hence, existing GCN-LSTM architectures and LSTM-engression appear as special cases of the 'more general' proposed models.

2. The statistical properties of classical engression were studied under an i.i.d. (independently and identically distributed) assumption on the covariates. In contrast, we establish the asymptotic properties of closed-loop, pre-additive spatiotemporal recurrent network processes, where the i.i.d. assumption is violated. To the best of our knowledge, this is the first time that asymptotic stationarity is studied for pre-additive noise-based spatiotemporal deep learning models.

3. From a practical perspective, our proposed architectures provide a highly efficient alternative to current resource-heavy probabilistic models. Compared to state-of-the-art diffusion-based forecasters, our lightweight models achieve strong probabilistic calibration with a significantly reduced computational footprint, making them well-suited for processing low-frequency datasets, such as epidemic incidence records.

4. We provide a comprehensive benchmarking resource for epidemiological forecasting by establishing a rigorous evaluation framework characterized by: six diverse datasets (varying in transmission modes, spatial structures, and temporal granularities), three forecasting horizons, twelve established temporal and spatiotemporal baseline models, and ten point-based and probabilistic metrics.

The remainder of this paper is structured as follows. Sec. 2 discusses the probabilistic forecasting problem for an epidemic data and motivates the use of pre-additive noise for time series forecasting. Sec. 3 provides an overview of the engression framework. Sec. 4 details the proposed methodologies, including the spatiotemporal problem formulation and the MVEN, GCEN, and STEN architectures. Theoretical results regarding geometric ergodicity and asymptotic stationarity are established in Sec. 5.1 and verified through numerical simulations in Sec. 5.3. The experimental evaluation and comparative analyses on real-world epidemic datasets are presented in Sec. 6, followed by an analysis of model explainability in Sec. 7. We conduct an ablation study in Sec. 8 to evaluate the components of proposed model architectures. In Sec. 9, we conclude the paper, highlighting the limitations of the proposal and directions for future research.

## 2 Motivating Example

Our work is motivated by real-world epidemic datasets where point forecasts are often inadequate to help decision makers in public health (Ioannidis et al., 2022). To illustrate, we consider weekly dengue incidence data in Guainia, Colombia (Clarke et al., 2024). To motivate the problem of probabilistic forecasting, we review a simple univariate time series forecasting task, where only the past lag $y_{t-1} \in \mathbb{R}$ is used to forecast the value for the next time-step, $y_t \in \mathbb{R}$ for the dengue dataset. One can observe the dependence of $y_t$ on $y_{t-1}$ using a lag plot, which is the visualization of $y_t$ against $y_{t-1}$. A vertical slice of this plot at any fixed lag value $y_{t-1}$ empirically reveals the conditional distribution of the future state $y_t$. Due to external factors driving the process, we may observe different future values for the same past lag. For example, in an epidemic process, the number of cases can jump from $y_{t-1} = 20$ to $y_t = 30$ in a given week, $y_t = 50$ in another week considering disease spread due to its contagious nature, and may dip to $y_t = 10$ in another, assuming effective control measures are imposed over time to curb the disease spread. Fig. 1 (A) illustrates this, with a lag plot on the dengue data of Guainia. With 835 observations in the range of $[0, 22]$ each, it is clear (in fact, follows from the pigeonhole principle; Rebman (1979)) that (at least) one past lag ($y_{t-1}$) value will correspond to multiple values for the future observation $y_t$.

In standard point forecasting frameworks, including classical AR(p) and autoregressive neural networks (ARNN), the modeling paradigm typically assumes a post-additive noise structure, formulated as $y_t = f(y_{t-1}) + \eta_t$. This formulation inherently imposes a rigid constraint on the predictive distribution. It presumes that the stochasticity of the system is invariant across the state space, manifesting as a symmetric, often Gaussian, error distribution centered around the conditional mean. Consequently, such models effectively enforce an uncertainty profile where the shape of the distribution remains constant regardless of the input. This limitation is visually evident in the inability of the mean fit to encompass the dispersion of incidence data (Fig. 1 (B)). Hence, making them generative by repeatedly sampling noise values to obtain 'plausible

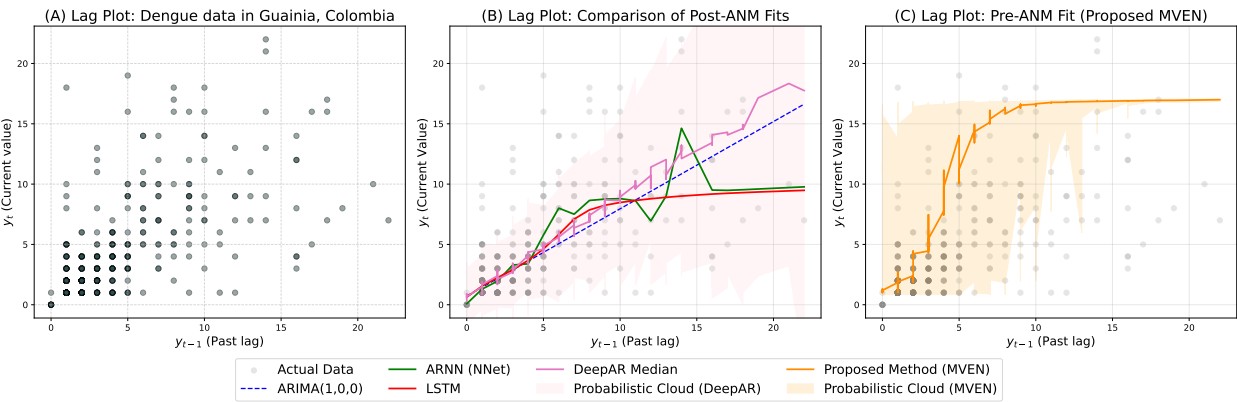

Fig. 1: (A) Lag plot $y_t$ vs. $y_{t-1}$ for weekly dengue data observed in Guainia, Colombia, (B) comparison of post-ANM fits on the lag plots, and (C) proposed MVEN (and a generated probabilistic cloud based on the proposal) fit on the data.

future trajectories' might result in a spurious uncertainty profile. Other than that, there are post-additive noise models such as DeepAR that allow input-dependent predictive distributions, applying rigid parametric assumptions on the data. Because a slight deviation in a sample influences the parameters of the next sample, the trajectories naturally fan out, making uncertainty predicted by such models grow the further into the future one predicts (see Figs. 1 (B) and 17). To address these issues, this work motivates a pre-additive noise mechanism: $y_t = f(y_{t-1} + \eta_t)$ coupled with optimization of a strictly proper scoring rule - the energy score loss (Gneiting & Raftery, 2007). By injecting stochasticity prior to the non-linear transformation $f$, the model empowers the neural network to function as a complex distributional lens by actively propagating noise through the model's non-linear layers. Moreover, training by optimizing the energy score loss mathematically ensures that the predicted distribution converges to the true distribution, thereby resolving the 'fanning out' uncertainty of models like DeepAR. The resulting 'probabilistic cloud' (see Fig. 1 (C)) obtained via repeated sampling of noise thus provides a rigorous approximation of the predictive density, ensuring that generated trajectories remain dynamically plausible and structurally consistent with the underlying physical process. Thus, Fig. 1 motivates the use of pre-additive noise models with strictly proper scoring rule as compared to post-additive noise models for epidemic forecasting[1]. Consequently, this work introduces spatiotemporal methods with a pre-additive noise structure, designed to generate reliable and accurate probabilistic forecasts for low-frequency epidemic datasets.

## 3 Background

Distributional regression fundamentally seeks to estimate the full conditional distribution $P(Y|X = x)$ of a target variable $Y$ given a set of covariates $X$, rather than a specific functional such as the conditional mean or median. Several established methodologies successfully achieve this goal by quantifying predictive uncertainty and capturing complex data distributions. For instance, Mixture Density Networks (MDNs) (Bishop, 1994) parameterize specific distribution families; normalizing flows (Papamakarios et al., 2021) utilize invertible transformations to learn highly complex densities; autoregressive architectures like DeepAR (Salinas et al., 2020) yield probabilistic forecasts by parameterizing probability distributions over sequential data. These methods often face significant challenges when extrapolating beyond the training data. To address this limitation within the distributional regression family, the engression framework was recently introduced as a deep generative methodology (Shen & Meinshausen, 2025). Engression distinguishes itself through two primary contributions: a strictly proper scoring rule (energy score loss) formulation and a pre-additive noise model (pre-ANM). Rather than relying on likelihood maximization, the model learns the conditional

---

[1]We note that the uncertainty produced by a post-additive noise model can also be shaped by training with the energy score loss. We compare this structure with the pre-additive formulation in an ablation study in Sec. 8.

distribution by minimizing the energy score loss. Concurrently, the pre-ANM injects noise $\eta$ directly into the covariates $X$ prior to a nonlinear transformation $g$, represented as $Y = g(X + \eta)$. This contrasts with the conventional post-additive noise model (post-ANM) where noise is appended after the transformation $(Y = g(X) + \eta)$.

The synthesis of a distributional fit via the energy score with a pre-ANM architecture equips engression with theoretically grounded advantages in extrapolation. While the extrapolative properties of engression provide strong theoretical guarantees, they rely on monotonicity assumptions which are violated in real-world spatiotemporal setups. However, the pre-additive structure of engression is particularly advantageous for time series forecasting because it renders the model inherently generative with a greater distributional flexibility through the non-linear propagation of noise. This is a critical mechanism for robust uncertainty quantification. To draw a single sample from the trained model, a random noise vector $\eta$ is sampled from a pre-defined distribution (e.g., uniform or Gaussian) and used to compute the perturbed input $X' = X + \eta$. This perturbed input is then passed through the learned function $g$ to yield a sample output $Y_{sample} = g(X')$. By repeating this stochastic process, the model can generate an entire ensemble of plausible future trajectories, effectively sampling from the learned distribution to provide a rich and dynamically consistent characterization of predictive uncertainty. Although point-based post-ANMs are also generative in the sampling sense, generating trajectories from them through this noise sampling strategy would result in a spurious uncertainty profile, as discussed in the preceding section. The generative capability of engression has been leveraged in environmental domains, such as rainfall-runoff prediction using LSTM-engression (Kraft et al., 2026). The present work develops deep spatiotemporal engression frameworks specifically designed for the probabilistic forecasting of multivariate time series under complex spatial dependencies. Our proposed models exploit the generative sampling mechanism of the pre-ANM and the energy score formulation to produce model-intrinsic prediction intervals, accurately quantifying uncertainty for forecasting spatiotemporal epidemic series.

## 4 Proposed Methodology

Spatiotemporal data, which archives measurements across both geographical locations and temporal intervals, is fundamental to the study of time series forecasting. This research focuses on predicting epidemic disease incidences, addressing the critical challenge of jointly modeling two distinct phenomena: the inherent temporal dependencies evident in historical cases data and the complex spatial relationships that exist between administrative regions, such as states or prefectures. The contagious nature of diseases like tuberculosis, COVID-19, dengue, and influenza-like illnesses necessitates this dual focus, as the transmission dynamics in one region are often influenced by its geographical neighbors. Assuming that there are $N$ administrative regions, the spatiotemporal dataset is a three dimensional tensor, $\mathcal{Y} \in \mathbb{R}^{T \times N \times D}$, which encapsulates measurements of $D$ features across $N$ discrete spatial nodes over a sequence of $T$ time-steps. An individual element in the tensor $\mathcal{Y}$, denoted as $y_{t,n,d}$, represents the value of the $d^{th}$ feature ($d \in \{1, 2, ..., D\}$), recorded at the $n^{th}$ spatial node ($n \in \{1, 2, ..., N\}$), at time-step $t$ ($t \in \{1, 2, ..., T\}$). A slice along the first axis at time $t$, denoted as $\mathbf{Y}_t$, is an $N \times D$ matrix representing a snapshot of all features across all nodes at that specific moment. In the current context, we have $D = 1$ as we are interested in modeling only the epidemic incidence cases for multiple spatial nodes; however, the setting can be generalized, where the last dimension can accommodate multiple target variables to forecast along with exogenous covariates (see Sec. 9). Then, given the historical epidemic data $\mathcal{Y} = \{\mathbf{Y}_t\}_{t=1}^{T} \in \mathbb{R}^{T \times N \times 1}$ up to time $T$, the aim is to design a model that can produce multi-step ahead forecasts for $q$ future time-steps ($q \in \mathbb{N}$): $\widehat{\mathcal{Y}} = \left\{\widehat{\mathbf{Y}}_t\right\}_{t=T+1}^{T+q} \in \mathbb{R}^{q \times N \times 1}$, where each $\widehat{\mathbf{Y}}_t \in \mathbb{R}^{N \times 1}$ is the prediction for incidence cases at the $t^{th}$ time-step across all $N$ nodes.

In this section, we present three spatiotemporal probabilistic forecasting frameworks, motivated by the idea of engression (developed for regression examples in Shen & Meinshausen (2025)) and LSTM-engression (developed for temporal forecasting in Kraft et al. (2026)).

### 4.1 Graph Convolutional Engression Network

We propose Graph Convolutional Engression Network (GCEN), a deep learning architecture designed for probabilistic spatiotemporal forecasting, particularly on data exhibiting graphical or lattice structures. Its

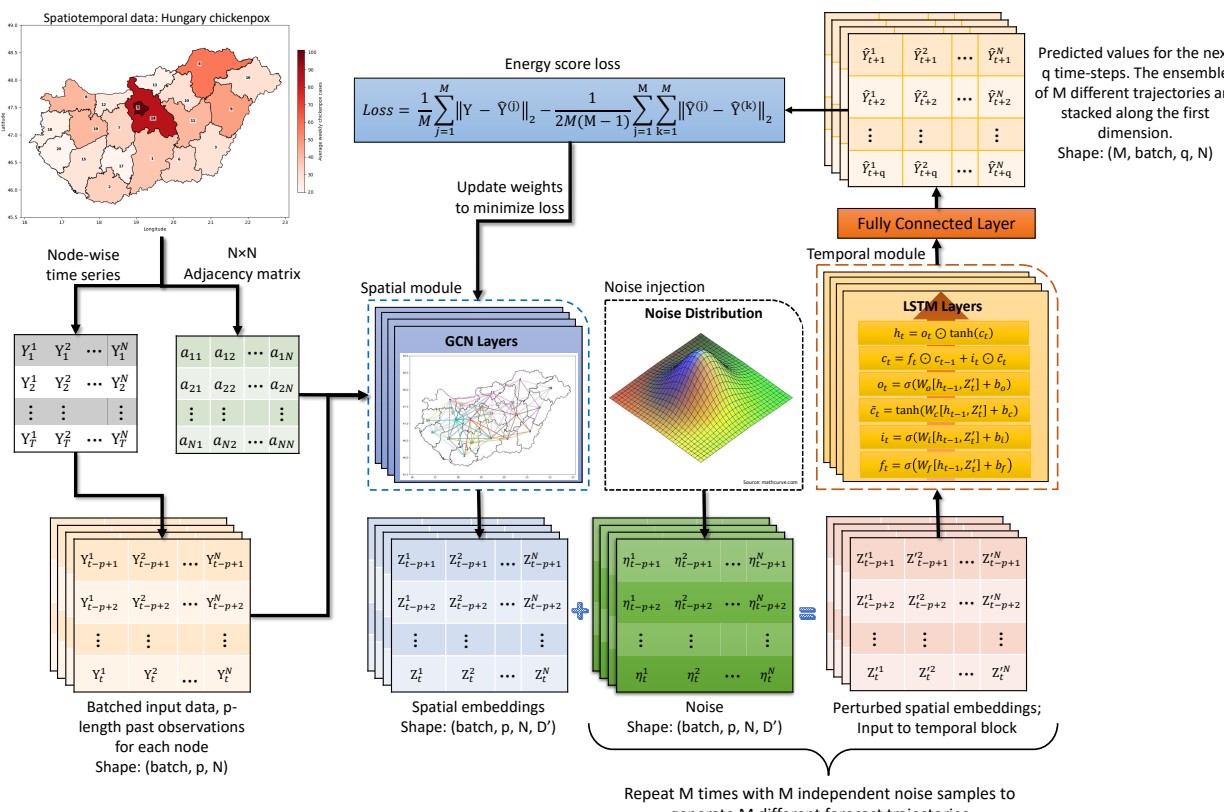

Fig. 2: A visual representation of the end-to-end training loop of the proposed GCEN architecture. Batches of input sequences along with the static adjacency matrix is first processed by the spatial module consisting of GCN layers to produce the spatial embeddings. Noise is sampled from a pre-determined distribution and added to the spatial embeddings, which are then passed to the temporal module consisting of LSTM layers, followed by a fully connected layer. The process is repeated with $M$ independent noise samples to generate an ensemble of $M$ forecasts for the next $q$ time-steps for each node. The network is trained by minimizing the energy score loss function.

design, inspired by Spatio-Temporal Graph Convolutional Networks (Yu et al., 2018), integrates spatial feature extraction with a probabilistic temporal forecasting mechanism. This creates a principled separation between the modeling of spatial dependencies and temporal dynamics. GCEN is made up of three key components: a spatial embedding function $f_{GCN}$, latent noise injection, and a temporal evolution function $f_{Temp}$. A visual illustration of the GCEN architecture is presented in Fig. 2.

The spatial module of the GCEN architecture comprises GCNs, that functions as a spatial feature extractor. It learns a spatially-aware representation of the system's state at each point in time. The underlying spatial relationships between the nodes are encoded in a static adjacency matrix, $A \in \mathbb{R}^{N \times N}$, which represents an undirected graph. First, the weighted Haversine distance ($d_{ij}$) between the geographical locations of nodes $i$ and $j$ with latitude-longitude pair ($lat_i, lon_i$) and ($lat_j, lon_j$), respectively, is computed as:

$$d_{ij} = 2R \sin^{-1} \left( \sqrt{\sin^2 \left( \frac{lat_i - lat_j}{2} \right) + \cos(lat_i) \cos(lat_j) \sin^2 \left( \frac{lon_i - lon_j}{2} \right)} \right), \tag{1}$$

where $R$ represents the radius of the Earth. The spatial interdependencies among the network nodes are formally characterized by a weighted adjacency matrix $A$, wherein the pairwise similarity $a_{ij}$ between distinct

nodes is derived from a thresholded Gaussian kernel:

$$a_{ij} = \exp\left(-\frac{d_{ij}^2}{\tilde{\sigma}^2}\right), \text{ when } i \neq j \text{ and } \exp\left(-\frac{d_{ij}^2}{\tilde{\sigma}^2}\right) \geq \epsilon,$$

where the hyperparameters $\tilde{\sigma}^2$ and $\epsilon$ serve as critical regulators of the matrix's distribution and sparsity, respectively. This Gaussian radial basis function effectively establishes a finite interaction radius; specifically, if the distance between two nodes exceeds the threshold $\sqrt{-\tilde{\sigma}^2 \ln \epsilon}$, the edge weight is truncated to zero, and thus no connectivity is considered between those nodes.

The spatial module operates in a sequence-to-sequence paradigm, utilizing a window of $p$ lagged time steps, $\mathbf{Y}_{t-p+1:t}$, as the input at time $t$. At each time-step $t$, it processes the feature slice $\mathbf{Y}_{t-p+1:t} \in \mathbb{R}^{p \times N \times D}$ and the adjacency matrix $A$, to produce a latent embedding $\mathbf{Z}_{t-p+1:t} \in \mathbb{R}^{p \times N \times D'}$, where $D'$ is the dimension of the node features in the embedding domain. This transformation can be formally expressed as:

$$\mathbf{Z}_{t-p+1:t} = f_{GCN}(\mathbf{Y}_{t-p+1:t}, A; \Theta_{GCN}),$$

where $\Theta_{GCN}$ represents the learnable weight matrix of the GCN layers. The function $f_{GCN}$ is typically a composition of graph convolution operations that propagate and transform node features across the graph, allowing each node's resulting representation to be informed by its local spatial context. To maintain localization and efficiency, GCNs approximate polynomial filters using a first-order Chebyshev expansion (Kipf & Welling, 2017). For an input signal $\mathbf{Y}_t \in \mathbb{R}^{N \times D}$, the graph convolution is defined as:

$$\mathbf{Y}'_t = \theta_0 \mathbf{Y}_t + \theta_1 \left(\frac{2L}{\lambda_{\max}} - I_N\right) \mathbf{Y}_t,$$

where $\theta_0, \theta_1$ are shared filter weights, $L$ is the graph Laplacian, $\lambda_{\max}$ is its largest eigenvalue, and $I_N$ is the identity matrix of order $N$. By stacking $K$ such layers and adhering to a neural message passing framework (Gilmer et al., 2017), the model generates spatial embeddings. Each node $i$ updates its representation $h_t^{i,(k)}$ through 1-hop mean aggregation of its neighborhood $\mathcal{N}(i)$ and its own prior state (Panja et al., 2026):

$$h_t^{i,(0)} = Y_t^i,$$

$$h_t^{i,(k)} = f_t^{(k)}\left(\Theta_t^{(k)} \frac{\sum_{j \in \mathcal{N}(i)} h_t^{j,(k-1)}}{|\mathcal{N}(i)|} + B_t^{(k)} h_t^{i,(k-1)}\right), \quad k = 1, \ldots, K,$$

$$Z_t^i = \text{Dense}\left(h_t^{i,(K)}\right),$$

where $f_t^{(k)}$ is the update function and $\Theta_t^{(k)}$ and $B_t^{(k)}$ are learnable parameters. The final spatiotemporal representation $Z_t^i$ is produced by a fully connected layer following the $K^{th}$ iteration. The resultant embedding matrix, $\mathbf{Z}_t = \left[Z_t^1, \ldots, Z_t^N\right]^\top \in \mathbb{R}^{N \times D'}$, thereby captures information from the $(K-1)$-order neighborhood of each node combined with their temporal information.

A core architectural element, adapted from the engression methodology, is the injection of stochastic noise into the learned latent space, that helps the model to produce probabilistic forecasts. For each spatial embedding $\mathbf{Z}_\tau$ within the $p$-length historical sequence ($\tau = \{t-p+1, \ldots, t\}$), a corresponding noise tensor $\boldsymbol{\eta}_\tau \in \mathbb{R}^{N \times D'}$ is sampled from a pre-defined distribution (e.g., standard Gaussian or Uniform). This noise is additively combined with the deterministic embedding to yield a perturbed latent state:

$$\mathbf{Z}'_\tau = \mathbf{Z}_\tau + \boldsymbol{\eta}_\tau.$$

This perturbation is applied to all time-steps in the lookback window, producing a stochastic embedding sequence, $\mathbf{Z}'_{t-p+1:t}$. An alternative injection method involves sampling a noise tensor $\boldsymbol{\eta}_\tau \in \mathbb{R}^{N \times D_{noise}}$ and concatenating it with the embedding, yielding a perturbed representation $\mathbf{Z}'_\tau \in \mathbb{R}^{N \times (D'+D_{noise})}$. For ease of presentation, this text will primarily assume the additive method; however, our model implementation flexibly supports both approaches. The choice of the noise distribution is data dependent; we empirically

observe that datasets depicting (local) periodicity are better modeled using uniform distribution, whereas for more noisy spatiotemporal datasets, Gaussian distribution provides better coverage during prediction.

The sequence of perturbed spatial embeddings, $\mathbf{Z}'_{t-p+1:t}$, serves as the input to the second module: a temporal model. In the temporal module of GCEN, we use an LSTM network, following prior works (Panja et al., 2026; Kraft et al., 2026). This module is responsible for capturing the complex temporal dynamics of the system's evolution and generating probabilistic forecasts through the noise injection mechanism. Hence, the temporal evolution governed by the function $f_{Temp}$ and parameterized by $\Theta_{Temp}$ is given by:

$$\mathcal{H}_\tau = f_{Temp}(\mathbf{Z}'_\tau, \ \mathcal{H}_{\tau-1}; \Theta_{Temp}) = \text{LSTM}(\mathbf{Z}'_\tau, \ \mathcal{H}_{\tau-1}; \Theta_{\text{LSTM}}),$$

where $\mathcal{H}_\tau \in \mathbb{R}^{N \times D_{hidden}}$ denotes the hidden state of the temporal model at time-step $\tau = \{t-p+1, \ldots, t\}$ for all $N$ nodes, and $\Theta_{\text{LSTM}}$ is the learnable weight matrix in LSTM. The final multi-step forecast for a future horizon of $q$ steps (starting after time-step $t$), denoted by $\widehat{\mathbf{Y}}_{t+1:t+q} \in \mathbb{R}^{q \times N \times D}$, is then generated from the final hidden state of the sequence via a fully connected (dense) output layer:

$$\widehat{\mathbf{Y}}_{t+1:t+q} = \text{Dense}(\mathcal{H}_t).$$

The process from the noise sampling step is repeated $M$ times, each with an independently drawn noise sequence $\left\{\boldsymbol{\eta}^{(j)}\right\}_{j=1}^M$ to generate an ensemble of $M$ distinct future trajectories $\left\{\widehat{\mathbf{Y}}^{(j)}_{t+1:t+q}\right\}_{j=1}^M$. This ensemble constitutes the model's probabilistic forecasts, where each $\widehat{\mathbf{Y}}^{(j)}_{t+1:t+q}$ represents a sample forecast (trajectory) drawn from its learned distribution. The GCEN model can be viewed as a generative process, mapping a historical input sequence consisting of values for $p$ time-steps, $\mathbf{Y}_{t-p+1:t}$, the graph structure $A$, and a sequence of $M$ noise samples $\left\{\boldsymbol{\eta}^{(j)}_{t-p+1:t}\right\}_{j=1}^M$ to an ensemble of $M$ forecast trajectories $\left\{\widehat{\mathbf{Y}}^{(j)}_{t+1:t+q}\right\}_{j=1}^M$. A forward pass is the process of passing input data through a neural network's layers - input, hidden, and output - to generate a prediction. A single ($j^{th}$) forward pass of the GCEN model (pseudo code presented in Algorithm 1 in Appendix B) produces a single forecast trajectory

$$\widehat{\mathbf{Y}}^{(j)}_{t+1:t+q} = f_{GCEN}\left(\mathbf{Y}_{t-p+1:t}, \ A, \ \boldsymbol{\eta}^{(j)}_{t-p+1:t}\right),$$

and this process is repeated $M$ times ($M$ forward passes) to generate $M$ forecast trajectories for the same forecast horizon, which constitute the probabilistic forecast ensemble. If the model is trained with the entire available historical data $\mathbf{Y}_{1:T}$, then a forecast for the next $q$ time-steps can be generated using the historical sequence $\mathbf{Y}_{T-p+1:T}$ as input to the trained model. The resulting forecast ensemble $\left\{\widehat{\mathbf{Y}}^{(j)}_{T+1:T+q}\right\}_{j=1}^M$ facilitates the derivation of point forecast trajectory (e.g., the ensemble median), as well as the estimation of prediction intervals through specific quantiles. A detailed discussion about estimation of the model parameters using energy score-based backpropagation is given in Sec. 4.4.

## 4.2 Spatio-Temporal Engression Network

The STEN architecture incorporates spatial dependencies based on a predefined spatial weights matrix, motivated by the idea of Spatiotemporal Autoregressive Moving Average (STARMA) (Pfeifer & Deutrch, 1980) model. By adapting this framework into a differentiable neural layer, it becomes possible to explicitly define the nature of spatial dependencies, such as the influence of second-order neighbors or the relative importance of different spatial scales. The STEN model comprises two sequential modules - a STAR-layer: STAR-inspired spatial embedding layer (see Rathod et al. (2018) for details), and a probabilistic engression-LSTM module.

At each time-step $\tau$, the STAR-layer receives the matrix of node features, $\mathbf{Y}_\tau \in \mathbb{R}^{N \times D}$, and a pre-computed, static $N \times N$ spatial weights matrix $W$. The Haversine distance $d_{ij}$ is computed between each node using equation 1, and the weights matrix is constructed as $W_{ij} = \frac{1}{d_{ij}^\alpha}$, where $\alpha$ is the decay parameter which can be tuned using a holdout validation set. The core operation of the layer is to compute a set of spatially lagged feature representations. For a pre-defined maximum spatial lag $L$, the layer calculates the sequence

of matrices $\{\mathbf{Y}_\tau, W\mathbf{Y}_\tau, W^2\mathbf{Y}_\tau, \dots, W^L\mathbf{Y}_\tau\}$. Each matrix $W^l\mathbf{Y}_\tau \in \mathbb{R}^{N \times D}$ represents the features of the $l^{th}$-order neighbors aggregated to each node. These computations can be performed very efficiently, especially if $W$ is a sparse matrix. The layer then combines these multi-scale spatial representations through a learnable, weighted aggregation. The final spatial embedding for time $\tau$, denoted by $\widetilde{\mathbf{Z}}_\tau \in \mathbb{R}^{N \times D'}$, is computed as:

$$\widetilde{\mathbf{Z}}_\tau = \text{ReLU}\left(\sum_{l=0}^{L}(W^l\mathbf{Y}_\tau)\Phi_l\right), \tag{2}$$

where each $\Phi_l \in \mathbb{R}^{D \times D'}$ is a learnable weight matrix, effectively acting as a linear transformation specific to the $l^{th}$ spatial lag and $D'$ is the dimensionality of the learned spatial embedding. This formulation allows the model to learn the relative importance of information from different spatial scales when creating the final embedding. The STAR-layer is applied to a batch of input sequences of length $p$ (for each batch) in each forward pass, transforming the raw input sequence $\mathbf{Y}_{t-p+1:t}$ into a sequence of rich spatial embeddings, $\widetilde{\mathbf{Z}}_{t-p+1:t}$. Each embedding $\widetilde{\mathbf{Z}}_\tau$ ($\tau = \{t-p+1, \dots, t\}$), encodes not just the features at time $\tau$, but also the learned multi-scale spatial context surrounding each node.

The spatial embeddings from the STAR-layer are then perturbed with noise before being passed to the temporal module, an LSTM network, to generate a hidden state tensor $\mathcal{H}_{t-p+1:t} \in \mathbb{R}^{p \times N \times D_{hidden}}$, similar to GCEN. Finally, a single dense layer acts as the forecasting head, which takes the final hidden state of the temporal module as input and maps it to the desired output shape for a $q$-step-ahead forecast horizon: $\widehat{\mathbf{Y}}_{t+1:t+q} \in \mathbb{R}^{q \times N \times D}$. The complete, end-to-end process for a single forward pass of the STEN model is detailed in Algorithm 2 (Appendix B). To generate probabilistic forecasts, the entire process from the noise sampling step is repeated $M$ times with different noise samples $\{\boldsymbol{\eta}^{(j)}\}_{j=1}^{M}$ to create an ensemble of trajectories $\left\{\widehat{\mathbf{Y}}_{t+1:t+q}^{(j)}\right\}_{j=1}^{M}$.

### 4.3 Multivariate Engression Network

MVEN is a multivariate time series framework based on engression. It utilizes a generative approach characterized by the direct injection of stochastic noise into the input features instead of spatial embeddings, which are subsequently processed by LSTM layers for forecasting. LSTM-engression, proposed by Kraft et al. (2026), maps the LSTM hidden states to a single value and uses softplus activation to transform them into the positive domain. In contrast, MVEN utilizes a fully connected layer after the LSTM network as a forecasting head, which maps the LSTM hidden states into the final forecast shape. MVEN uses energy score loss (see Sec. 4.4) during backpropagation for model parameter estimation, and the forecast ensemble consisting of $M$ trajectories is generated as a stacked tensor of shape $(M, q, N, D)$. Formulated as a sequence-to-sequence framework similar to GCEN and STEN, a single forward pass of MVEN ingests a historical observation tensor $\mathbf{Y}_{t-p+1:t} \in \mathbb{R}^{p \times N \times D}$ and generates a $q$-step ahead forecast $\widehat{\mathbf{Y}}_{t+1:t+q} \in \mathbb{R}^{q \times N \times D}$ for all the nodes (see Algorithm 3 in Appendix B). As a purely (multivariate) temporal framework, this model can be viewed as a baseline for evaluating spatiotemporal architectures, as it skips spatial dependencies, thereby treating the nodes as independent, to isolate the contribution of temporal dynamics in the dataset. MVEN will be particularly useful for spatiotemporal forecasting problems where locational information are seldom available, such as movements of robots while performing manipulation tasks.

### 4.4 Optimization

The training dataset $\mathcal{D} = \{(\mathbf{Y}_{t-p+1:t}, \mathbf{Y}_{t+1:t+q})\}$, $p \leq t \leq T - q$, is constructed by taking sequential overlapping slices from the original multivariate time series, where each input sample $\mathbf{Y}_{t-p+1:t} \in \mathbb{R}^{p \times N \times D}$ and its corresponding ground truth $\mathbf{Y}_{t+1:t+q} \in \mathbb{R}^{q \times N \times D}$ are temporally contiguous. The proposed models are trained by minimizing the empirical energy score loss (Gneiting & Raftery, 2007), a strictly proper scoring rule that evaluates the quality of the entire predictive distribution against the ground truth observation. The loss function comprises two terms: an accuracy term and a sharpness term, which allows the model to learn not just a single 'best guess' prediction, but the entire probable distribution of future states, while still ensuring the average prediction is accurate. This is similar to the idea of balancing exploration and

exploitation in reinforcement learning (Sutton et al., 1998). For a single slice of ground truth sequence and its prediction (an ensemble of $M$ forecasts), the accuracy term is given by:

$$\mathcal{L}_{accuracy} = \frac{1}{M} \sum_{j=1}^{M} \left\| \mathbf{Y}_{t+1:t+q} - \widehat{\mathbf{Y}}_{t+1:t+q}^{(j)} \right\|_2^{\beta},$$

and the sharpness term is given by

$$\mathcal{L}_{sharpness} = \frac{1}{2M(M-1)} \sum_{j=1}^{M} \sum_{k=1}^{M} \left\| \widehat{\mathbf{Y}}_{t+1:t+q}^{(j)} - \widehat{\mathbf{Y}}_{t+1:t+q}^{(k)} \right\|_2^{\beta},$$

where $\mathbf{Y}_{t+1:t+q}$ is the ground truth (observed target vector) during training, $\widehat{\mathbf{Y}}_{t+1:t+q}^{(j)}$ is the in-sample predicted $j^{th}$ trajectory (an independent sample drawn from the model's predicted distribution), $M$ is the total number of forecasts in the forecast ensemble ($j = 1, 2, \ldots, M$), and the exponent $\beta \in (0, 2)$. The empirical energy score (ES) loss computes the loss between all (batches of) sequences of ground truth observations and in-sample predictions in $\mathcal{D}$ as follows:

$$\mathcal{L}_{ES} = \frac{1}{|\mathcal{D}|} \sum_{i=1}^{|\mathcal{D}|} \left[ \frac{1}{M} \sum_{j=1}^{M} \left\| \mathbf{Y}_i - \widehat{\mathbf{Y}}_i^{(j)} \right\|_2^{\beta} - \frac{1}{2M(M-1)} \sum_{j=1}^{M} \sum_{k=1}^{M} \left\| \widehat{\mathbf{Y}}_i^{(j)} - \widehat{\mathbf{Y}}_i^{(k)} \right\|_2^{\beta} \right], \tag{3}$$

where $\mathbf{Y}_i$ is the ground truth sequence of length $q$ (second element of the $i^{th}$ tuple in $\mathcal{D}$) and $\widehat{\mathbf{Y}}_i^{(j)}$ is its predicted value.

We empirically illustrate the benefits of ES loss over standard data loss such as mean squared error (MSE) in probabilistic forecasting through an ablation study in Sec. 8 and a toy example in Appendix C. Since the loss function in equation 3 is differentiable almost everywhere, standard backpropagation is used to optimize the objective and learn the weights of the proposed models. The full end-to-end training loop of the proposed models is given in Algorithm 4 in Appendix B.

*Remark* 1 (Model selection). Model selection among the proposed frameworks is contingent upon the specific trade-off between data size, interpretability, and the complexity of spatial dependencies inherent in the data. Because of the multiplication operations between input tensors and powers of spatial weights matrices in STEN (equation 2), the computational complexity of STEN is directly proportional to data size. In contrast, GCEN possesses better scalability because of weight-sharing in the graph convolution operations. Although the MCB analysis (Figs. 9 and 12) establishes the empirical superiority of GCEN over STEN, GCEN's graph-based convolutions operates as a black box, masking the explicit pathways of spatial dissemination. STEN addresses this limitation by offering structural transparency; its parameterization explicitly isolates and quantifies the impact of individual spatial lags on disease propagation (see Sec. 7.2). Thus, STEN is preferable when interpretability is paramount and data is scarce, while GCEN is superior for capturing complex, nonlinear spatial heterogeneity through data-driven graph convolutions and is more scalable. Finally, MVEN is optimal for datasets with weak or noisy spatial signals, as it isolates temporal dynamics without imposing potentially confounding spatial structures.

## 5 Geometric Ergodicity and Asymptotic Stationarity

### 5.1 Theoretical Results

Geometric ergodicity is a fundamental property for evaluating the stability and predictive reliability of stochastic forecasting models. It confirms that the underlying Markov chain converges to a unique stationary distribution at an exponential rate (Meyn & Tweedie, 2012). This geometric convergence implies that the influence of initial conditions and distant past observations decays exponentially fast. Establishing this property is important, as it ensures the model is stable and its forecasts are asymptotically independent of arbitrary starting conditions, which is a prerequisite for reliable long-term prediction. For the theoretical

studies, we consider a closed-loop (i.e., no exogenous covariates) and 1-step recurrent (input and output sequence lengths $p, q = 1$) setup without loss of generality, for simplicity. The results can be extended for a setup with $p, q > 1$ and exogenous inputs. We first prove that such a temporal network process driven by pre-additive noise is geometrically ergodic, following Zhao et al. (2020). Using that, we then proceed to demonstrate that the proposed MVEN, GCEN, and STEN processes, when formulated as homogeneous Markov chains in their closed-loop forms, are geometrically ergodic.

To establish the geometric ergodicity and asymptotic stationarity of the engression processes, we first show that they are irreducible, meaning that all the states in the Markov chain communicate with each other. The concept of irreducibility is essential when considering the structure of the Markov chain for any nonlinear time series model. By constructing a suitable non-negative continuous function and verifying Tweedie's drift criterion (Tweedie, 1983) along with irreducibility, we ensure the existence of a unique invariant distribution to which the process converges geometrically fast, thus establishing ergodicity. Before stating the main results, we formally define the notions of irreducibility and geometric ergodicity.

**Definition 1** (Irreducibility). Let $\lambda_k$ be the Lebesgue measure on the state space $\mathcal{S} \subseteq \mathbb{R}^k$. The Markov chain $\{S_t\}$ is $\lambda_k$-irreducible if, for every initial state $s \in \mathcal{S}$ and every set $A \subseteq \mathcal{S}$ with $\lambda_k(A) > 0$, there exists an integer $n \geq 1$ such that $P^n(\mathbf{s}, A) > 0$. This means any set $A$ with positive measure can eventually be reached from any starting state $s$.

**Definition 2** (Geometric ergodicity). Let $\mathcal{B}$ and $\lambda$ denote the Borel $\sigma$-field and Lebesgue measure, respectively. A Markov chain $\{S_t\}$ is called geometrically ergodic if there exists a probability measure $\Pi$ on a probability triple $(\mathcal{S}, \mathcal{B}, \lambda)$ and a constant $\rho > 1$ such that $\lim_{n \to \infty} \rho^n \|\mathbb{P}^n(\mathbf{s}, \cdot) - \Pi(\cdot)\| = 0$, $\forall \mathbf{s} \in \mathcal{S}$ where $\|\cdot\|$ denotes the total variation norm. Then, we say the distribution of $S_t$ converges to $\Pi$ and $S_t$ is asymptotically stationary.

Denote by $\mathcal{S} \subseteq \mathbb{R}^k$ the state space, $\mathcal{B}^k$ the class of Borel sets on $\mathbb{R}^k$, and $\lambda_k$ the Lebesgue measure on $(\mathbb{R}^k, \mathcal{B}^k)$. Let $\{\mathbf{y}_t\}_{t \in \mathbb{N}}$ be the target sequence, where $\mathbf{y}_t \in \mathbb{R}^N$. Let $\mathbf{h}_t \in \mathbb{R}^{N'}$ denote general hidden states, so that $k = N + N'$. We define a network process with pre-additive noise and no exogenous inputs (for simplicity), where the input at the $t^{th}$ step is the previous observed value $\mathbf{y}_{t-1}$. Let $\boldsymbol{\eta}_t$ be the noise added to the input at the $t^{th}$ time-step. To establish irreducibility, without loss of generality, we add a vanishingly small noise $\boldsymbol{\xi}_t$ to the hidden states. This can be formulated as a homogeneous Markov chain with transition function $\mathcal{F}$:

$$\begin{pmatrix} \mathbf{y}_t \\ \mathbf{h}_t \end{pmatrix} = \mathcal{F}\left( \begin{pmatrix} \mathbf{y}_{t-1} \\ \mathbf{h}_{t-1} \end{pmatrix} + \begin{pmatrix} \boldsymbol{\eta}_t \\ \boldsymbol{\xi}_t \end{pmatrix} \right),$$

or equivalently,

$$S_t = \mathcal{F}\left( S_{t-1} + \boldsymbol{\varepsilon}_t \right), \tag{4}$$

where $S_t = \left( \mathbf{y}_t^\top, \mathbf{h}_t^\top \right)^\top \in \mathbb{R}^k$ denotes the current state, $\boldsymbol{\eta}_t \in \mathbb{R}^N$ is a random noise vector sampled from a pre-determined distribution with pdf $f_{\boldsymbol{\eta}}(\cdot)$, and $\boldsymbol{\varepsilon}_t = \left( \boldsymbol{\eta}_t^\top; \boldsymbol{\xi}_t^\top \right)^\top$, where $\boldsymbol{\xi}_t \in \mathbb{R}^{N'}$, $\boldsymbol{\xi}_t \sim \mathcal{N}(\mathbf{0}_{N'}, \sigma^2 I)$, where $\sigma_{ii}^2$ are vanishingly small positive scalars.

To prove the geometric ergodicity of the temporal network process with pre-additive noise, we impose the following assumptions:

(A1) The joint density function of $\boldsymbol{\eta}_t$ is continuous and positive everywhere.

(A2) For any initial state $\mathbf{s} \in \mathbb{R}^k$ and any Borel set $A \in \mathcal{B}^k$ with positive Lebesgue measure ($\lambda_k(A) > 0$), the set $\left\{ \left( \boldsymbol{\eta}^\top, \boldsymbol{\xi}^\top \right)^\top \in \mathbb{R}^k : \mathcal{F}\left( \mathbf{s} + \left( \boldsymbol{\eta}^\top, \boldsymbol{\xi}^\top \right)^\top \right) \in A \right\}$ has positive Lebesgue measure.

(A3) $\mathbb{E}(\|\boldsymbol{\eta}_t\|^c) < \infty$ for some $c \geq 1$.

(A1) and (A3) are standard assumptions in Zhao et al. (2020), that can be satisfied by, for example, standard Gaussian noise; hence, these assumptions trivially hold for $\boldsymbol{\xi}_t$. (A3) assumes that the noise term $\boldsymbol{\eta}_t$ has a finite $c^{th}$-order moment. In our case, it suffices to consider a finite first moment. (A2) is necessary to prove irreducibility in our pre-additive setup, which means that for any state $\mathbf{s} \in \mathbb{R}^k$ and any Borel set $A$ with $\lambda_k(A) > 0$, the image set $\left\{ \mathcal{F}\left( \mathbf{s} + \left( \boldsymbol{\eta}^\top, \boldsymbol{\xi}^\top \right)^\top \right) : \left( \boldsymbol{\eta}^\top, \boldsymbol{\xi}^\top \right)^\top \in \mathbb{R}^k \right\}$ must intersect $A$ such that the preimage of this intersection has a positive Lebesgue measure in $\mathbb{R}^k$.

The theorem below states that a recurrent network process with pre-additive noise is geometrically ergodic under the assumptions (A1)-(A3) stated above. The proof is given in Appendix A, and involves showing that the Markov chain $\{S_t\}$ is irreducible, followed by establishing Tweedie's drift criterion (Tweedie, 1983).

**Theorem 1.** *Suppose that Assumptions (A1)-(A3) hold. If there exist real numbers $a \in (0,1)$ and $b$ such that $\|\mathcal{F}(z)\| \leq a\|z\| + b$, then the pre-additive recurrent network process defined in equation 4 is geometrically ergodic.*

Next, we proceed to prove that closed-loop 1-step recurrent GCEN, STEN, and MVEN processes are geometrically ergodic under mild assumptions. The output of the GCEN and STEN processes are determined by an LSTM network, which processes (perturbed) spatial embeddings as input. The hidden unit computations in an LSTM network involve several gating operations, as outlined below.

$$
\begin{aligned}
\mathbf{f}_t &= \sigma(W_{fh}\mathbf{h}_{t-1} + W_{fy}\mathbf{y}_{t-1} + b_f) && \text{(forget gate)} \\
\mathbf{i}_t &= \sigma(W_{ih}\mathbf{h}_{t-1} + W_{iy}\mathbf{y}_{t-1} + b_i) && \text{(input gate)} \\
\mathbf{o}_t &= \sigma(W_{oh}\mathbf{h}_{t-1} + W_{oy}\mathbf{y}_{t-1} + b_o) && \text{(output gate)} \\
\widetilde{\mathbf{c}}_t &= \tanh(W_{ch}\mathbf{h}_{t-1} + W_{cy}\mathbf{y}_{t-1} + b_c) && \text{(candidate cell state)} \\
\mathbf{c}_t &= \mathbf{i}_t \odot \widetilde{\mathbf{c}}_t + \mathbf{f}_t \odot \mathbf{c}_{t-1} && \text{(cell state)} \\
\mathbf{h}_t &= \mathbf{o}_t \odot \tanh(\mathbf{c}_t) && \text{(hidden state)} \\
\mathbf{z}_t &= g(W_{zh}\mathbf{h}_t + b_z) && \text{(output)}
\end{aligned}
$$

where, $\mathbf{y}_0 = \mathbf{0}_N, \mathbf{h}_0 = \mathbf{0}_{N'}$, $\mathbf{y}_t, \mathbf{z}_t \in \mathbb{R}^N$, $\mathbf{h}_t, \mathbf{c}_t, \widetilde{\mathbf{c}}_t, \mathbf{o}_t, \mathbf{i}_t, \mathbf{f}_t \in \mathbb{R}^{N'}$ for $t = \{1, \ldots, T\}$, $g, \sigma$, and tanh are respectively an elementwise output function and the elementwise sigmoid and hyperbolic tangent functions, and $\odot$ is the elementwise (Hadamard) product. The hidden weight matrices $W_{fh}, W_{ih}, W_{oh}, W_{ch} \in \mathbb{R}^{N' \times N'}$, input weight matrices $W_{fy}, W_{iy}, W_{oy}, W_{cy} \in \mathbb{R}^{N' \times N}$, output weight matrix $W_{zh} \in \mathbb{R}^{N \times N'}$, and bias terms $b_f, b_i, b_o, b_c \in \mathbb{R}^{N'}$ and $b_z \in \mathbb{R}^N$. Let $f_{spatial} : \mathbb{R}^N \to \mathbb{R}^N (\mathbb{R}^{1 \times N \times 1} \to \mathbb{R}^{1 \times N \times 1})$ denote the transformation that maps the input sequence to spatial embeddings, which is achieved by the graph convolution operations in GCEN and STAR-layer in STEN. Hence, the GCEN and STEN processes can be defined as:

$$
\begin{pmatrix} \mathbf{y}_t \\ \mathbf{h}_t \\ \mathbf{c}_t \end{pmatrix} = \mathcal{F}_{LSTM}\left( \begin{pmatrix} f_{spatial}(\mathbf{y}_{t-1}) \\ \mathbf{h}_{t-1} \\ \mathbf{c}_{t-1} \end{pmatrix} + \begin{pmatrix} \boldsymbol{\eta}_t \\ \boldsymbol{\xi}_t \\ \boldsymbol{\psi}_t \end{pmatrix} \right), \tag{5}
$$

where $\mathcal{F}_{LSTM}$ denotes the transition function of the LSTM network and $\boldsymbol{\psi}_t \in \mathbb{R}^{N'}$ is a vanishingly small noise component added to the cell state, whose distribution follows that of $\boldsymbol{\xi}_t$. To prove that the simplified versions of GCEN and STEN processes are geometrically ergodic, we make another set of assumptions as follows.

(A4) The noise-perturbed output generated by the spatial function $f_{spatial}$ is bounded everywhere: $f_{spatial}(\mathbf{x}) + \boldsymbol{\eta} \in B_\infty^N$.

(A5) $\widetilde{M} := \sup_{x \in B_\infty^{N'}} \|g(W_{zh}x + b_z)\|_1 < \infty$.

(A6) $\sigma(\|W_{fh}\|_\infty + \|W_{fy}\|_\infty + \|b_f\|_\infty) \leq a$ for some $a < 1$.

Assumption (A4) can be made true, for instance, with bounded activation functions such as tanh and sigmoidal, used inside $f_{spatial}$ (GCN or STAR layers). Assumption (A5) ensures that the $\ell_1$-norm of the final output from the process is finite, given that the hidden state value is in $B_\infty^{N'}$, which denotes the $N'$-dimensional $\ell_\infty$-ball. Assumption (A6) means that the forget gate is contractive with respect to the matrix $\ell_\infty$-norm, denoted by $\|\cdot\|_\infty$ and defined as $\|P_{m \times n}\|_\infty = \max_{1 \leq i \leq m} \sum_{j=1}^n |p_{ij}|$. Now, we state the main results for geometric ergodicity and asymptotic stationarity of the closed-loop spatiotemporal engression Markov processes.

**Theorem 2.** *Suppose that Assumptions (A1)-(A6) hold. Then the GCEN and STEN processes defined in equation 5 are geometrically ergodic.*

**Corollary 1** (Asymptotic stationarity)**.** *Let $S_t = \left(\mathbf{y}_t^\top, \mathbf{h}_t^\top, \mathbf{c}_t^\top\right)^\top$ be the GCEN or STEN process satisfying the conditions of Theorem 2, then $\{S_t\}$ is asymptotically stationary.*

*Remark* 2. If the spatial function $f_{spatial}$ is the identity function, then equation 5 represents the MVEN process.

**Corollary 2.** *Under assumptions (A1)-(A6), the MVEN process is geometrically ergodic and asymptotically stationary.*

*Remark* 3.     1. Corollaries 1 and 2 are a direct consequence of Theorem 2. The proof for Theorem 2 is provided in Appendix A.

  2. The theoretical injection of (vanishingly small) noise into the hidden states to ensure irreducibility is inspired by the 'wobbling trick' introduced by Jaeger (2001) for echo-state networks (ESNs). In ESNs, injection of uniform noise inside the network states makes them wobble around the perfect periodic state sequence and stabilizes the fixed points. In our setup, the noise added to the spatial embeddings naturally propagates through the temporal module's hidden states; therefore, we omit explicit noise re-injection in our practical model implementation to avoid redundancy.

  3. In practice, the LSTM forget gate $f_t = \sigma(\cdot)$ inherently outputs values in $(0, 1)$, naturally favoring a contractive state update. The only scenario where contractivity fails is if the pre-activations grow unboundedly positive, causing $f_t \to 1$. This unbounded growth can be prevented in practical implementations by standard input scaling and $L_2$ weight regularization, which naturally bounds the pre-activation magnitude. Consequently, the maximum expected value of the forget gate remains strictly bounded away from 1.

### 5.2   Practical Implications of Theoretical Results

The theoretical results derived for the simplified, closed-loop recurrent processes provide an essential mathematical foundation for MVEN, GCEN, and STEN. While our empirical forecasting tasks operate in a finite-sample, multi-step regime on real epidemic data, and without the explicit injection of hidden-state noise used in the mathematical proofs, evaluating this idealized setting allows us to formally demonstrate the stability of the underlying networks. For example, establishing irreducibility in the theoretical setting indicates that the core network structure is not fundamentally confined to a limited portion of the state space; it possesses the architectural capacity to transition between disparate states (e.g., from a low endemic period to a high-incidence outbreak). Furthermore, the results on geometric ergodicity demonstrate that the autonomous versions of these networks are structurally stable and resistant to 'explosive behavior' over time. This foundational stability ensures that, under the specified bounded and contractive conditions, the influence of arbitrary initial conditions decays exponentially fast. Moreover, the theoretical assumptions can be enforced in practice by taking suitable noise distribution, activation functions, and appropriately scaling the inputs. This suggests that the probabilistic forecasts observed empirically are grounded in a mathematically non-explosive architecture, making them trustworthy tools for decision-making.

### 5.3   Simulation Study

To empirically verify the theoretical stability guarantees of the closed-loop processes, we modeled the generative processes as autonomous, homogeneous Markov chains. For each process, we conducted simulations consisting of 200 independent trials to empirically verify the asymptotic stationarity and geometric ergodicity. The 1-step recurrent GCEN and STEN architectures were instantiated with GCN and STAR spatial modules featuring hidden dimensions of $D_{hidden} = 16, 32,$ and 64, integrated with an LSTM cell for the temporal module. The MVEN architecture was instantiated with an LSTM cell, with the same settings for hidden units. For each trial in GCEN, we generated a random graph with a random number of spatial nodes $N$ from 10 to 60. For STEN, we instantiated the $N \times N$ spatial weights matrix (where $N \in \{10, \dots, 60\}$) with random entries such that the diagonal elements are zero and each row sums to one. The hidden and cell states $\mathbf{h}_0$ and $\mathbf{c}_0$, of shape $N \times D_{hidden}$, in all cases, were initialized with zeros. The input state vector $\mathbf{Y}_0$ of shape $1 \times N \times 1$ (the first dimension corresponds to time-step, and the last to that of input dimension) was also initialized with zeros. The noise injection was carried out with a standard Gaussian noise. The models

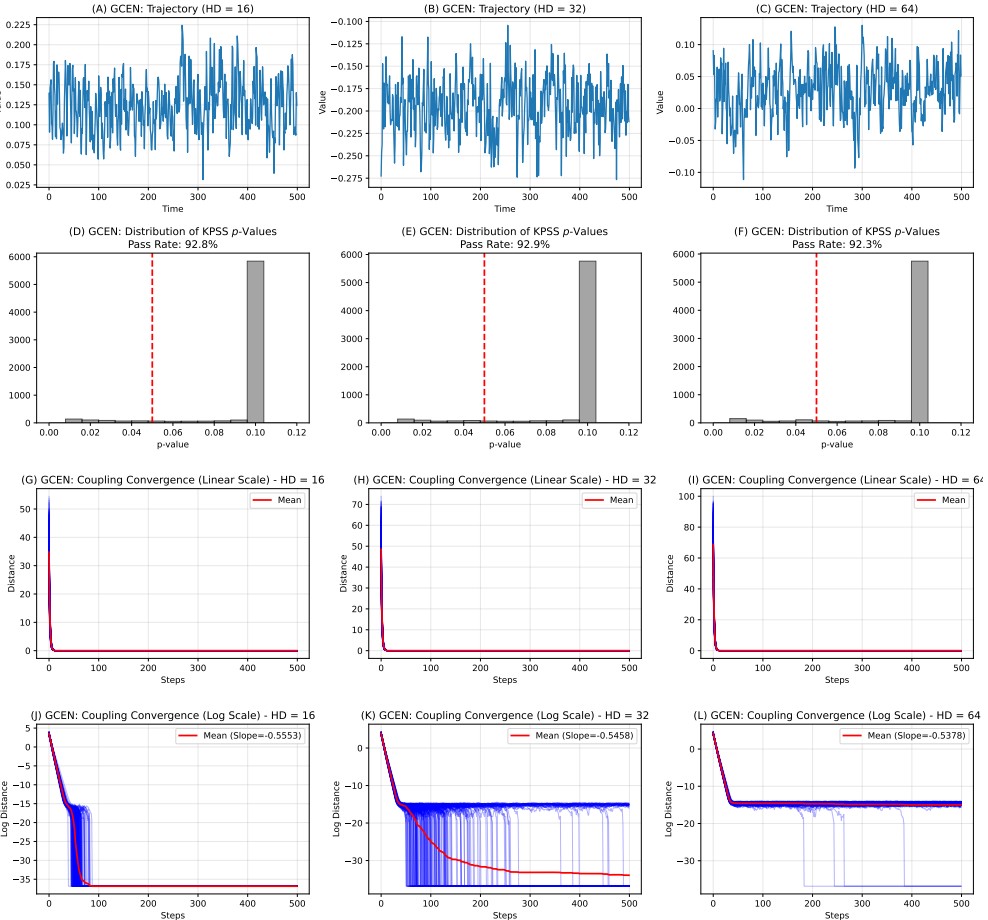

Fig. 3: Simulation results for GCEN: empirical verification of (marginal) asymptotic stationarity and geometric ergodicity of the GCEN process through simulations for 500 time steps and 200 trials.

were executed in a closed-loop autoregressive mode over a horizon of $T = 500$ time steps for each trial, after a burn-in period of 50.

The following analysis details the simulation results of the closed-loop GCEN process; results for the STEN and MVEN processes are substantially similar and are deferred to Appendix G (Figs. 15 and 16, respectively). First, we verify the marginal asymptotic stationarity for each spatial node. We applied the Kwiatkowski-Phillips-Schmidt-Shin (KPSS) test to the generated trajectories of all $N$ nodes, at a significance level of 0.05. Unlike standard unit root tests, the KPSS null hypothesis posits stationarity; thus, we aggregated the 'pass rate' defined as the proportion of nodal time series where the null hypothesis could not be rejected ($p > 0.05$), i.e., the proportion of time series that are stationary. We obtained pass rates of 92.8%, 92.9%, and 92.3%, respectively for GCEN models simulated with 16, 32, and 64 hidden dimensions, signifying strong evidence of asymptotic stationarity. In Fig. 3, panels (A)-(C) present the simulated values of the time series for an example node (the last node in the last trial) for each model (hidden dimension setting). Panels (D)-(F) illustrate the distribution of $p$-values of the KPSS test, with the dashed red vertical line denoting $p = 0.05$.

To verify geometric ergodicity, we employed a synchronous coupling strategy. In each of the 200 trials, two independent chains were initialized at divergent states, $\mathbf{S}_0 = (\mathbf{Y}_0, \mathbf{h}_0, \mathbf{c}_0)$ and $\mathbf{S}'_0 = (\mathbf{Y}'_0, \mathbf{h}'_0, \mathbf{c}'_0)$, but were driven by an identical sequence of noise realizations $\{\boldsymbol{\eta}_t\}_{t \geq 1}$. We monitored the Euclidean distance between the trajectories as additive norm, $d_t = \|\mathbf{Y}_t - \mathbf{Y}'_t\|_2 + \|\mathbf{h}_t - \mathbf{h}'_t\|_2 + \|\mathbf{c}_t - \mathbf{c}'_t\|_2$. Ideally, for a geometrically ergodic system, this distance should contract exponentially, adhering to $d_t \leq C\zeta^t$ for some $\zeta < 1$. We quantified this by fitting a linear regression to the logarithmic distance, $\ln(d_t) \approx \gamma t + \kappa$, and reporting the

mean decay rate $\bar{\gamma}$ across all trials. The regression window was restricted to the regime where $d_t > 10^{-6}$ (an arbitrarily chosen small float) to avoid artifacts arising from floating-point machine precision floors. The resulting negative slopes of the mean logarithmic distance, $\bar{\gamma} \approx -0.56, -0.55$, and $-0.54$, respectively, for the models with hidden dimensions 16, 32, and 64, confirms exponential contraction before the coupled chains effectively merge (Fig. 3 (J)-(L) and (G)-(I)), which indicates that the system forgets initial conditions at a geometric rate, validating the contractive drift of the GCEN process. Simulation analyses indicate that data generated from the closed-loop spatiotemporal engression processes exhibit asymptotic stationarity and temporal stability, lacking explosive divergence. Nevertheless, in practical applications, models trained on finite samples yield robust forecasts across both stationary and nonstationary time series dynamics.

## 6 Application to Real-World Epidemic Datasets

This section outlines the experimental framework for evaluating the proposed models on real-world epidemic datasets.

### 6.1 Datasets

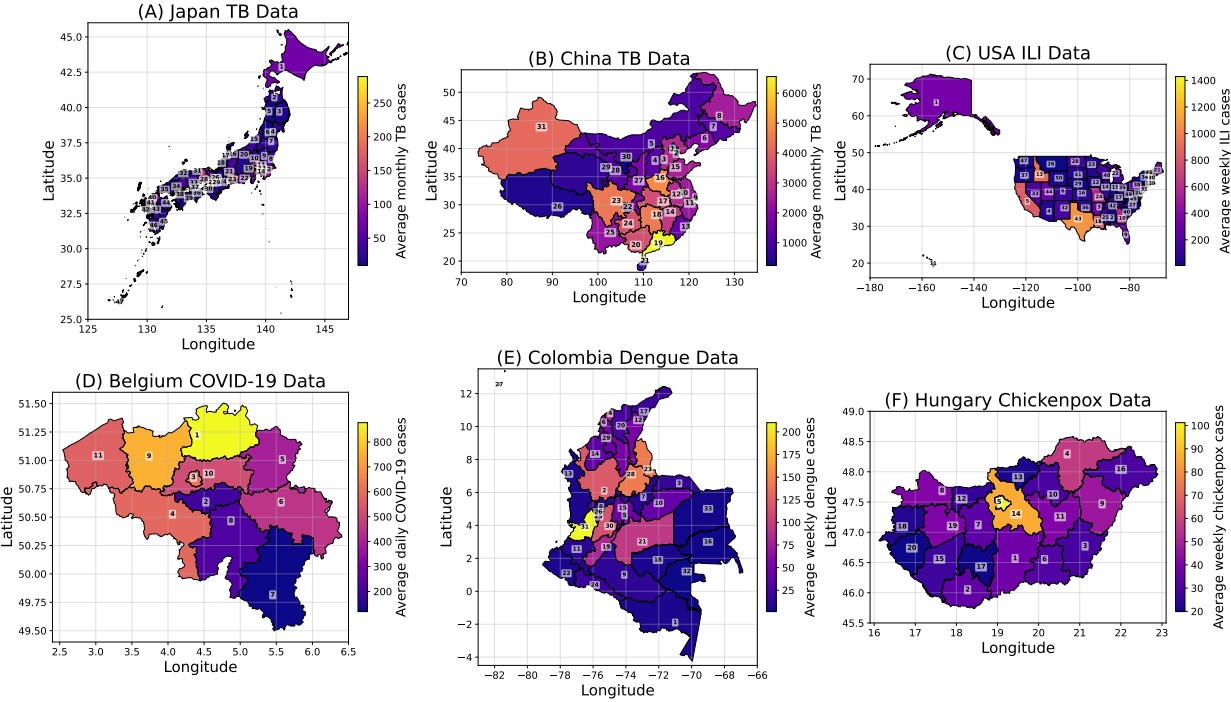

Fig. 4: Geographical maps of the countries under study: (A) Japan, (B) China, (C) USA, (D) Belgium, (E) Colombia, and (F) Hungary - illustrating the mean incidence cases at each labeled node (see Tables 3-8 in Appendix D for the corresponding node names). The territorial boundaries of the countries are shown for illustrative purposes only, without implying any political assertions.

To ensure a rigorous and comprehensive evaluation, the proposed models were benchmarked on six different spatiotemporal datasets encompassing diverse diseases and geographical regions. They vary in sizes, granularity, and number of nodes, as detailed in Table 1. We are particularly interested about five diseases characterized by diverse etiological agents and transmission mechanisms. Dengue is a vector-borne disease with a very high fatality rate (2-5%; Sah et al. (2023)); Tuberculosis (TB) is a bacterial infection, whereas COVID-19, influenza-like illnesses (ILI), and varicella (chickenpox) are viral pathogens. The latter four diseases spread predominantly through direct anthropogenic pathways, including aerosolized respiratory droplets, person-to-

person contact, and fomite contact. Therefore, models that integrate human mobility and the underlying spatial topology of the region can produce near real-time risk maps and case projections.

Table 1: Datasets used in this study.

| Country | Disease | No. of observations | No. of locations | Frequency | Time frame | Reference |
|---------|---------|--------------------|-----------------|-----------|-----------|-----------|
| Japan | Tuberculosis (TB) | 216 | 47 | Monthly | January 1998-December 2015 | Sumi & Kobayashi (2019) |
| China | Tuberculosis (TB) | 60 | 31 | Monthly | January 2014-December 2018 | Ma & Fan (2023) |
| USA | Influenza-Like Illnesses (ILI) | 328 | 50 | Weekly | October 2010-January 2017 | CDC[2] |
| Belgium | COVID-19 | 776 | 11 | Daily | September 2020-October 2022 | Epistat[3] |
| Colombia | Dengue | 835 | 33 | Weekly | January 2007-December 2022 | Clarke et al. (2024) |
| Hungary | Chickenpox | 522 | 20 | Weekly | January 2005-December 2014 | Rozemberczki et al. (2021b) |

Fig. 4 presents the spatial distribution of incidence cases at all locations for different datasets considered in this study, utilizing a color gradient to represent the mean case counts at each node. The global statistical properties of each dataset such as mean, standard deviation, range, skewness, kurtosis, stationarity tests, and nonlinearity tests are presented in Tables 3-8 in Appendix D, along with a brief discussion on the temporal properties.

### 6.1.1 Spatial Autocorrelation

We study the evolution of spatial autocorrelation over time for all the datasets. Spatial autocorrelation measures the extent to which a variable's geographic distribution deviates from spatial randomness. Moran's I is the definitive global statistic for this assessment, yielding a standardized coefficient typically ranging from $-1$ to $+1$. Positive values signify spatial clustering, i.e., nodes with high (low) values are surrounded by neighbors that also have high (low) values; negative values denote dispersion, and values near the expected null indicate randomness (Moraga, 2023). The statistic is defined as:

$$I = \frac{n \sum_i \sum_j w_{ij}(Y_i - \overline{Y})(Y_j - \overline{Y})}{(\sum_{i \neq j} w_{ij}) \sum_i (Y_i - \overline{Y})^2}$$

where $n$ is the number of spatial nodes, $Y_i$ is the observed value in region $i$, and $\overline{Y}$ is the global mean. The connectivity between units is defined by a weights matrix $W \in \mathbb{R}^{N \times N}$ (where $w_{ii} = 0$), constructed as in STEN (see Sec. 4.2). To capture the dynamic behavior of spatiotemporal systems, calculating Moran's I for each discrete interval reveals the temporal evolution of spatial structure. As illustrated in Fig. 5, this approach identifies critical transitions, showing how clustering patterns emerge, stabilize, or fragment over time. Each plot reveals a unique spatial pattern for the respective region and disease. Japan (TB) exhibits a gradually intensifying positive spatial autocorrelation from 1998 to 2016, while China (TB) maintains a lower, stable clustering pattern from 2014 to 2019. Conversely, the USA (ILI) shows significant volatility, with Moran's I frequently fluctuating between positive clustering and spatial randomness. Belgium (COVID-19) displays oscillatory cycles, where autocorrelation peaks during infection waves and dips into negative territory during troughs. Colombia (dengue) is characterized by high-intensity clustering spikes, with deep spatial dispersion (negative Moran's I) in majority of the period (2007-2022). Finally, Hungary (chickenpox) demonstrates extreme volatility, with rapid transitions between strong clustering and dispersion throughout the decade.

The analysis suggests a significant neighborhood effect in disease transmission. Accurate probabilistic forecasting of such infectious diseases is of our main interest since it is often difficult for the government and public health officials to know the exact disease mutation and transmission mechanism, and anticipate outbreaks while taking decisions. Data-driven statistical and machine learning systems can work with temporal dynamics in the presence of spatial dependencies to produce multi-step-ahead forecasts of the disease cases in a real-time setting. Moreover, effective epidemic forecasting necessitates robust probabilistic frameworks and uncertainty quantification to provide actionable insights for public health policy.

---

[2]https://gis.cdc.gov/grasp/fluview/fluportaldashboard.html
[3]https://epistat.sciensano.be/covid/

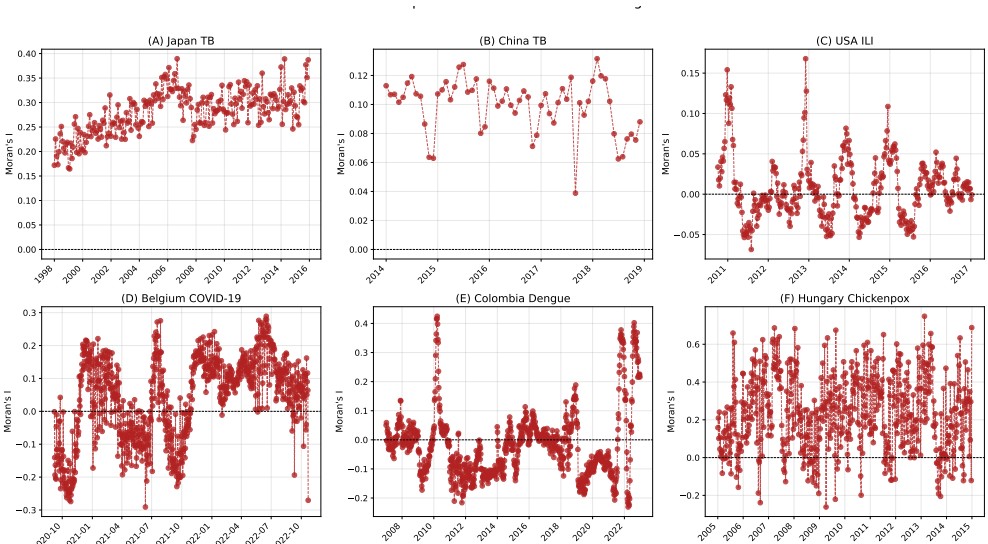

Fig. 5: Temporal evolution of global spatial autocorrelation (Moran's I) across the six epidemiological datasets.

## 6.2 Experimental Setup

The experimental framework is designed for a rigorous, multi-horizon evaluation tailored to the temporal granularity of each dataset. The monthly datasets (Japan and China TB) are evaluated on 6, 12, and 24-month forecast horizons by producing out-of-sample multi-step-ahead forecasts, with the 24-month horizon excluded for China due to its limited length. The daily dataset (Belgium COVID-19) is assessed over 30, 60, and 90-day forecast horizons, while the weekly datasets (USA ILI, Colombia dengue, and Hungary chickenpox) are evaluated over 4, 9, and 13-week horizons. To prevent data leakage, a strict temporal split is employed (as discussed in Appendix E.1), where the final segment of each time series corresponding to the forecast horizon is held out as the test set. The training data is standardized before training the deep learning models. Specifically, each column is standardized using the individual mean and standard deviation of the corresponding column (node). The training process for the proposed models incorporated uniform noise for the Belgium dataset (due to pronounced localized sinusoidal patterns; see Fig. 7, panels (I)-(L)) and Gaussian noise for all other datasets, for comprehensive prediction interval coverage. Selecting the noise distribution was based on a validation approach. More implementation details such as the choice of validation set lengths and hyperparameter optimization are provided in Appendix E.1. The spatial topology used in GCEN is encoded in a static adjacency matrix $A$ for each country. For all the datasets, the connectivity between nodes was determined using the Haversine distance, calculated as per equation 1 and the adjacency matrix was formed using the methodology described in Sec. 4.1.

Model performance is evaluated against a suite of competitive baselines, presented in Table 2. Brief specifications for these baseline architectures along with relevant references and implementation details are provided in Appendix E. We use a set of 10 metrics (see Appendix F for details) to assess both point and probabilistic forecast accuracy. Metrics used to evaluate point forecast performance are Symmetric Mean Absolute Percentage Error (SMAPE), Mean Absolute Error (MAE), Root Mean Squared Error (RMSE), Mean Absolute Scaled Error (MASE), and Root Mean Squared Scaled Error (RMSSE). Probabilistic metrics include Pinball loss at the $80^{th}$ and $95^{th}$ percentiles (Pinball-80, Pinball-95), $\rho$-risks at $\rho = 0.5$ and 0.9, Continuous Ranked Probability Score (CRPS), Winkler Score, and visualization of calibration through Probability Integral Transform (PIT) Q-Q plots. For the sake of completeness, we also report the empirical coverage obtained by the 95% prediction intervals (PIs) produced by the probabilistic frameworks, although we argue that it is a misleading metric.

We observe that setting $\beta = 1$ in the energy score loss (equation 3) gives the best results for our use-case. We use $M = 2$ during the training process, following prior works (Kraft et al., 2026; Shen & Meinshausen, 2025).

Table 2: Comparison of forecasting models. The columns assess whether each model can handle non-linearity, non-stationarity, low frequency data, is spatiotemporal and scalable, computationally lightweight, and finally can produce probabilistic forecasts.

| Models | Non-linear | Non-stationarity | Scalability | Low-frequency | Spatiotemporal | Computationally lightweight | Probabilistic forecasting |
|---|---|---|---|---|---|---|---|
| LSTM (Hochreiter & Schmidhuber, 1997) | ✓ | ✓ | ✓ | ✓ | ✗ | ✓ | ✗ |
| NHiTS (Challu et al., 2023) | ✓ | ✓ | ✓ | ✓ | ✗ | ✓ | ✗ |
| Transformers (Vaswani et al., 2017) | ✓ | ✓ | ✓ | ✗ | ✗ | ✓ | ✗ |
| TCN (Bai, 2018) | ✓ | ✓ | ✓ | ✓ | ✗ | ✓ | ✗ |
| GSTAR (Ruchjana et al., 2012) | ✗ | ✓ | ✗ | ✓ | ✓ | ✓ | ✗ |
| STARMA (Pfeifer & Deutrch, 1980) | ✗ | ✗ | ✗ | ✓ | ✓ | ✓ | ✗ |
| STGCN (Yu et al., 2018) | ✓ | ✓ | ✓ | ✗ | ✓ | ✗ | ✗ |
| DeepAR (Salinas et al., 2020) | ✓ | ✓ | ✓ | ✓ | ✗ | ✓ | ✓ |
| Prob-iTransformer (Liu et al., 2023) | ✓ | ✓ | ✓ | ✗ | ✗ | ✗ | ✓ |
| GpGp (Guinness, 2018) | ✗ | ✗ | ✓ | ✓ | ✓ | ✗ | ✓ |
| DiffSTG (Wen et al., 2023) | ✓ | ✓ | ✓ | ✗ | ✓ | ✗ | ✓ |
| STESN (Huang et al., 2022) | ✓ | ✓ | ✓ | ✗ | ✓ | ✗ | ✓ |
| **MVEN** (Proposed) | ✓ | ✓ | ✓ | ✓ | ✓ | ✓ | ✓ |
| **GCEN** (Proposed) | ✓ | ✓ | ✓ | ✓ | ✓ | ✓ | ✓ |
| **STEN** (Proposed) | ✓ | ✓ | ✓ | ✓ | ✓ | ✓ | ✓ |

This means that in the training loop, we call the model's forward pass $M = 2$ times. Hence, two independent noise tensors are sampled and two distinct (in-sample) forecast trajectories $\widehat{\mathbf{Y}}^{(1)}$ and $\widehat{\mathbf{Y}}^{(2)}$ are generated, with which the loss is calculated. Essentially, the training time depends on the value of $M$; a larger value of $M$ will correspond to more training time. For evaluation, an ensemble of 100 forecast trajectories is generated, from which the median is extracted as the point prediction. Consistent with prior literature (Kraft et al., 2026), we employ the median rather than the mean due to its superior robustness against outliers, thereby ensuring a more reliable point forecast trajectory.

Due to the inherent stochasticity of an engression model, a single trained instance yields non-deterministic forecasts. To ensure a robust and rigorous evaluation, we account for this variability by generating 50 independent prediction ensembles from the single trained model for each test case. The final results obtained by the performance metrics across three distinct forecast horizons, presented in Tables 11–13 (Appendix G), are reported as the mean and standard deviation computed across these 50 runs. To summarize, we repeat the process from forecast generation to metric calculation for 50 times and report the mean value for the metrics; for each forecast, we generate 100 trajectories and take the median as the point prediction. We also report the total computational time (both training and inference) measured in seconds in Table 9 in Appendix E.1. Time consumption by the proposed models in the total training-to-evaluation loop is significantly less, as compared to baselines such as STGCN, Prob-iTransformer, GpGp, DiffSTG, and STESN.

Moreover, the generation of multiple forecast trajectories to form a forecast ensemble leads to model-intrinsic uncertainty quantification. This property is leveraged to construct 95% PIs, with the lower and upper bounds defined by the $2.5^{th}$ and $97.5^{th}$ quantiles of the empirical forecast distribution (ensemble of 100 trajectories), respectively. The quality of the intervals produced by MVEN, GCEN, and STEN are assessed on the test set using the Winkler score (Winkler et al., 1996), PIT analysis, and empirical coverage, all of which are briefly described in Appendix F. The experimental results demonstrate the superiority of the proposed models in both point and probabilistic forecasting. Fig. 6 presents a comparative analysis with respect to CRPS, illustrating the performance of the proposed methods relative to the benchmark models across the six epidemic datasets. It is observed that, based on CRPS, the proposed models consistently outperform the state-of-the-art temporal and spatiotemporal methods.

### 6.3 Point Forecast Performance

To compute the point forecasts in MVEN, GCEN, and STEN, we consider the median of the forecast ensemble. Experimental results on the point forecast metrics are reported in Tables 11-13, which depict that overall, the proposed methods either outperform the benchmarks, or remain competitive for short, medium, and long-term forecasting.

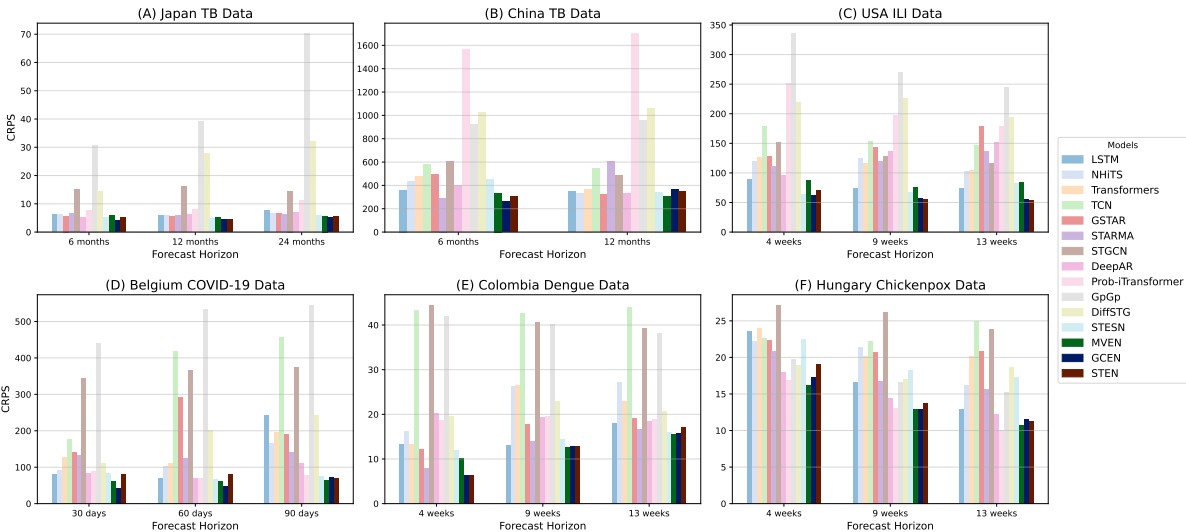

Fig. 6: CRPS of the proposed and benchmark models on the test sets of the (A) Japan TB, (B) China TB, (C) USA ILI, (D) Belgium COVID-19, (E) Colombia Dengue, and (F) Hungary Chickenpox datasets. Lower values indicate better performance. To maintain a comprehensible scale, results for TCN and Transformers on the Japan TB data are not shown due to their significantly larger CRPS magnitudes.

For Japan's TB dataset, GCEN achieved the lowest SMAPE and MASE scores for the 6- and 12-month horizons, as well as the best MAE in all horizons. At the 24-month horizon, Transformers exhibited signs of instability with extremely high MAE and RMSE values (475.54 and 516.28, respectively). China's TB dataset is characterized by low-frequency monthly observations and a restricted volume of 60 data points per province. In such data-constrained contexts, statistical models conventionally outperform deep learning architectures. For the 6-month forecast horizon, STARMA yielded the optimal values across several point metrics. Notably, GCEN remained highly competitive, frequently achieving the first and second ranks.

Our proposed models achieved high predictive accuracy on USA's ILI and Belgium's COVID-19 datasets. At the 4-week horizon for USA's ILI data, STESN showed highly competitive performance, achieving the best scores in all point metrics. STEN performed the best at the 9-week horizon, excelling in all point metrics except SMAPE, where it is shy of LSTM by a very short margin. At the 13-week horizon, STEN attained the best scores in all metrics but RMSE. Across all the horizons, GCEN and STEN achieved better scores than MVEN in terms of metrics such as MAE and RMSE, reflecting the importance of the spatial modules. For 30- and 60-day forecasts on Belgium's COVID-19 dataset, GCEN was unequivocally the best model, achieving the best scores in terms of all the point metrics. For the longest horizon of 90 days, GCEN and MVEN showed highly competitive performance, where MVEN frequently achieved the best and second best scores.

For Colombia's dengue data, model performance varied across metrics and forecasting horizons. On the 4-week horizon, GCEN outperformed others in terms of SMAPE, and achieved the second best RMSE, MASE, and RMSSE. Overall, MVEN exhibited strong and consistent performance at the 9- and 13-week horizons, ranking first in five and two metrics respectively. In this particular context, the spatial modules (in GCEN and STEN) exhibit suboptimal performance, likely attributable to high levels of noise or the sparse spatial dependencies characteristic of the Colombia Dengue dataset (Fig. 5 (E)). On Hungary's chickenpox data, MVEN achieved the best SMAPE, and GCEN scored the lowest MAE and MASE for the 4-week horizon. NHiTS achieved the best RMSE, MASE (jointly with GCEN), and RMSSE scores. The 13-week horizon was characterized by mixed performance, with STEN achieving the best SMAPE, LSTM the best MAE and MASE, and NHiTS the best RMSE and RMSSE.

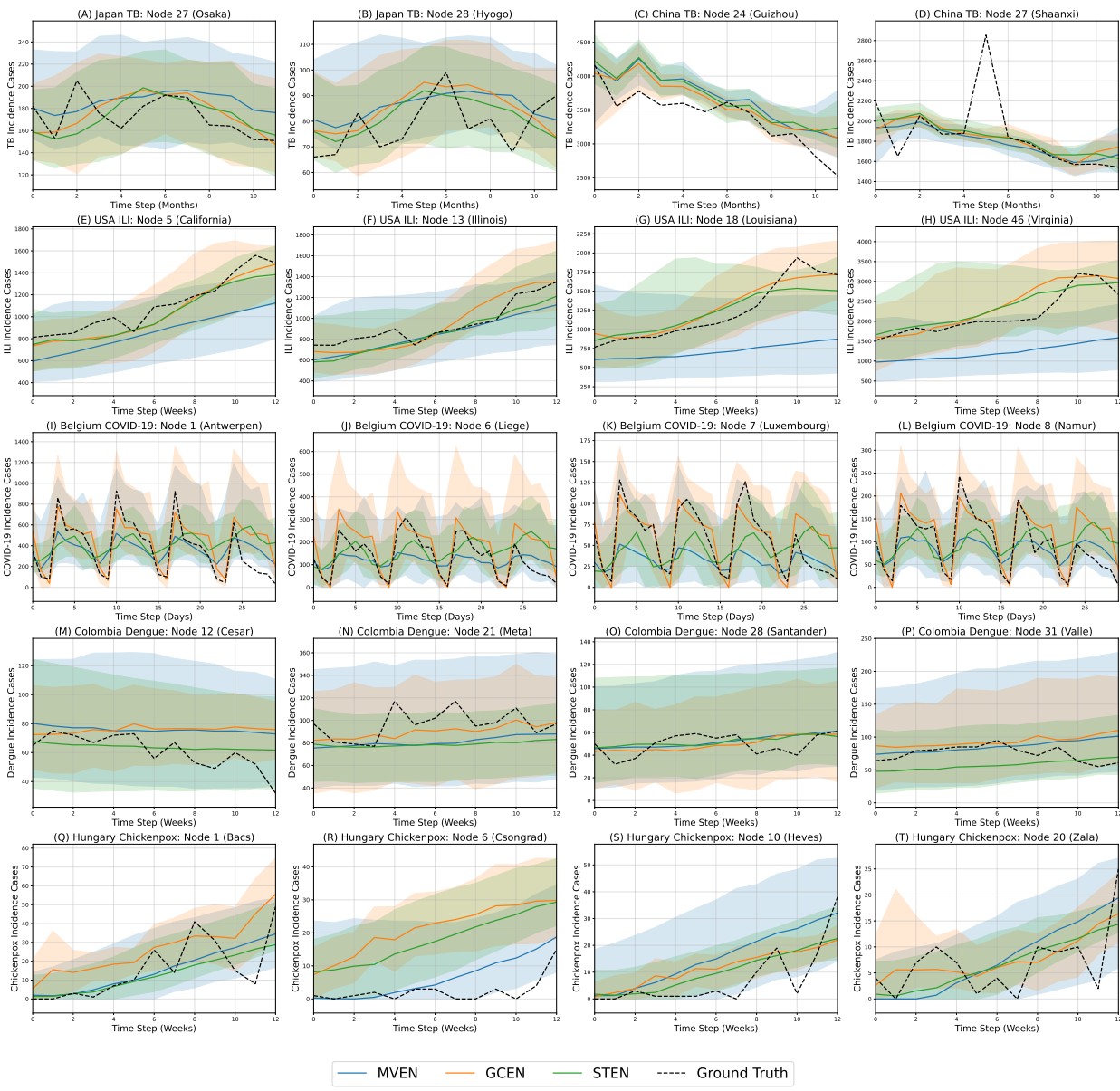

Fig. 7: Comparison of out-of-sample forecasts produced by MVEN (blue), GCEN (orange), and STEN (green) against the ground truth (test data) on selected nodes across the six datasets. The shaded regions denote the 95% PIs produced by the respective models.

## 6.4 Probabilistic Forecast Performance

Beyond point predictions, metrics like Pinball scores, $\rho$-risks, and CRPS evaluates probabilistic forecasts. The proposed models offer a natural way of uncertainty quantification through a probabilistic framework underpinned by a pre-additive noise-driven stochastic formulation. Fig. 7 presents the forecasts along with 95% PIs generated by MVEN, GCEN, and STEN on selected nodes across the datasets. The quality of the generated PIs is crucial for decision-making under uncertainty, and are evaluated using Winkler score and PIT Q-Q plots. Winkler score is a reliable metric for evaluating prediction intervals, which simultaneously penalizes both excessively wide intervals and low coverage. Overall, the point forecasts are well-aligned with the ground truth and the intervals provide sufficient coverage.

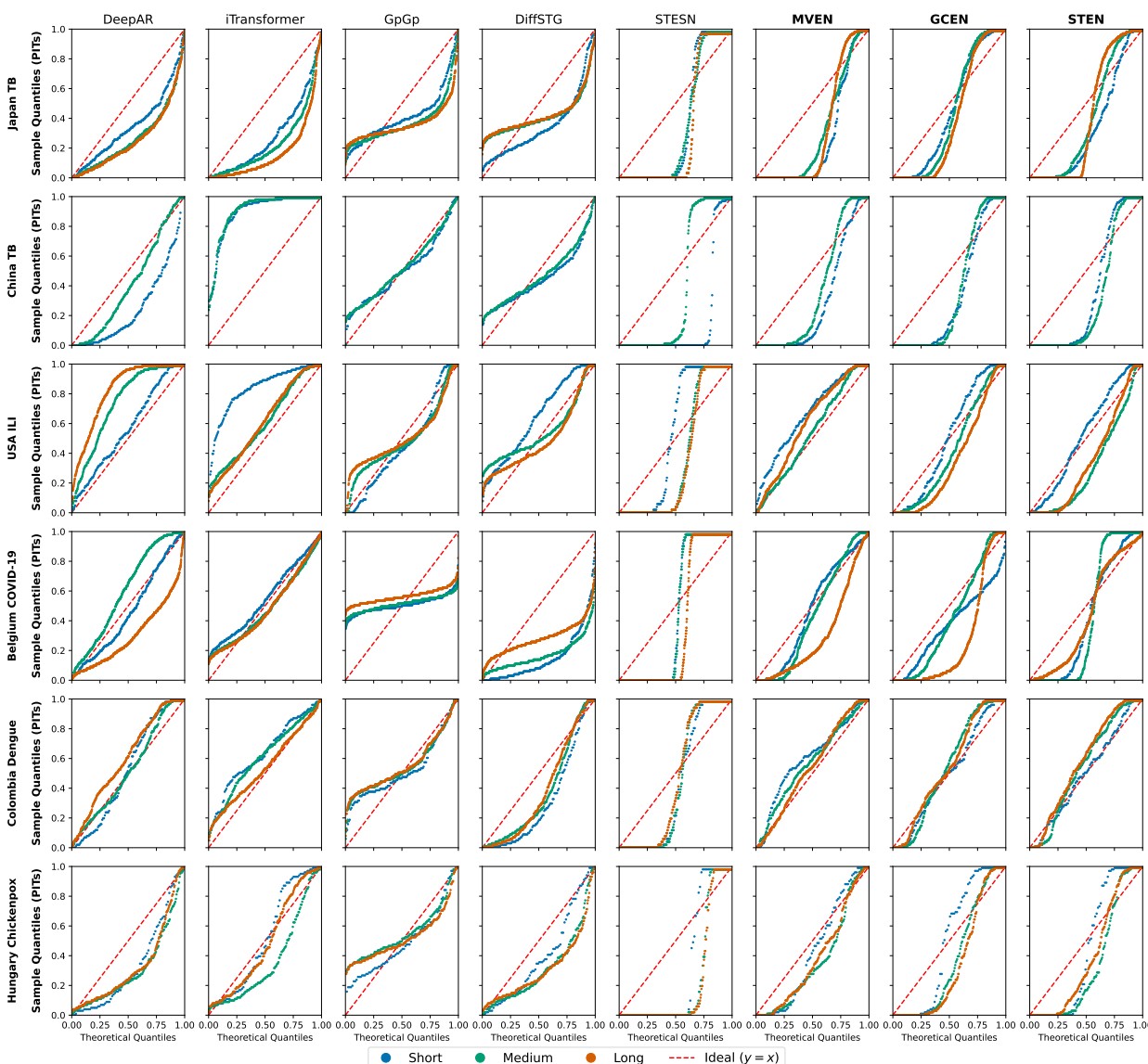

Fig. 8: Multi-horizon PIT Q-Q plots for qualitative evaluation of the probabilistic calibration of the proposed models across six epidemic datasets. Each subplot maps the empirical quantiles of the calculated PIT values (y-axis) against the theoretical quantiles of a standard Uniform(0,1) distribution (x-axis). The dashed red diagonal line ($y = x$) represents ideal calibration. The solid lines are color-coded by forecast horizons: short (blue), medium (green), and long (orange), to illustrate the stability of calibration over time. The China TB dataset excludes the long (24-month) horizon due to limited data length.

Across all evaluated datasets and temporal horizons, the proposed engression-based models demonstrated superior performance, achieving the best Pinball scores, $\rho$-risks, and CRPS values in 70.59%, 67.65%, and 94.11% of the instances, respectively. GpGp, DiffSTG, and STESN showed diminished performance on Winkler scores. Furthermore, although GpGp attained high empirical coverage (97–100%; see Table 10 in Appendix G), its intervals were characterized by excessive width, resulting in suboptimal Winkler scores. The iTransformer architecture also frequently produced excessively wide intervals (which is not useful in decision making), especially for Colombia's dengue dataset. A visualization of the forecasts and uncertainty intervals for DeepAR, Prob-iTransformer, and GpGp, for the same nodes as in Fig. 7, is provided in Fig. 17

in Appendix G. It is clear from the plots on the out-of-sample examples that our proposed models produced reasonable probabilistic bands (Fig. 7) as compared to the wide and uninformative bands produced by the benchmark spatiotemporal probabilistic models (Fig. 17).

To qualitatively evaluate the probabilistic calibration of our forecasting models, we employ PIT Q-Q plots and visualize the empirical PIT values against a theoretical Uniform(0, 1) distribution (see Appendix F for details on PIT). If the forecast ensembles are perfectly calibrated, these PIT values will follow a standard Uniform(0, 1) distribution. The Q-Q plots visualize this by mapping the empirical quantiles of these calculated PIT values (y-axis) against the theoretical quantiles of the Uniform distribution (x-axis). The plots are presented in Fig. 8. Points falling on the $y = x$ diagonal (dashed red line) indicate perfect calibration. Deviations into an S-shaped curve suggest overconfidence where PIs are narrow, while an inverse S-shape reveals underconfidence due to excessively wide intervals. The proposed models exhibit a characteristic S-shaped calibration curve, reflecting a high-precision forecasting approach that yields narrow, informative prediction intervals. While this indicates a degree of overconfidence, the curves remain closely aligned with the ideal diagonal across all datasets without requiring calibration using a validation set.

## 6.5 Statistical Test for Model Comparison

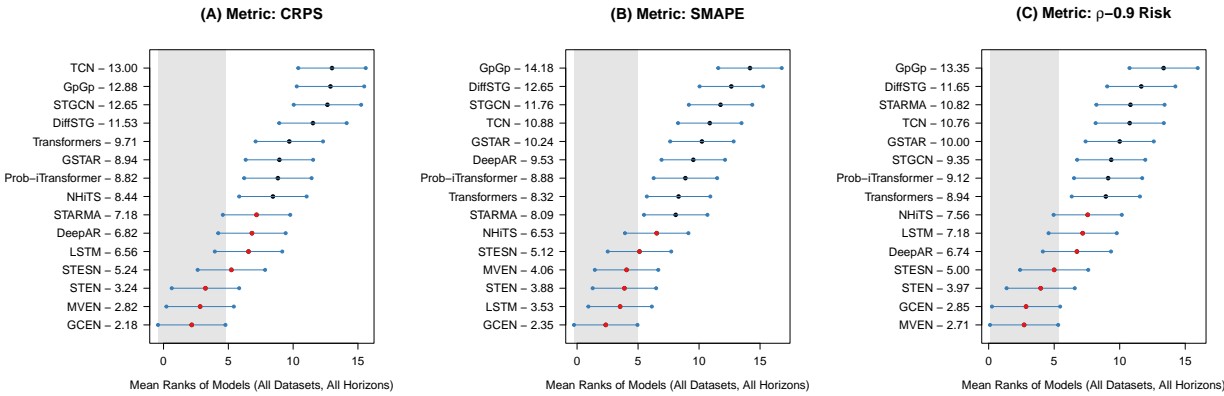

Fig. 9: MCB test results based on average ranks across six datasets and multiple forecast horizons for (A) CRPS, (B) SMAPE, and (C) $\rho$-0.9 Risk. '$\mu$-$r$' in the y-axis of each plot indicates that model $\mu$'s average rank is $r$ according to that metric.

To evaluate the comparative performance of the forecasting models across all datasets and horizons, we employ the non-parametric Multiple Comparisons with the Best (MCB) test (Koning et al., 2005). The MCB test is a statistical procedure used to identify which models are significantly better than others by computing their mean ranks and establishing a critical distance (Nemenyi, 1963). Any model whose critical distance does not overlap with the best-performing model is considered statistically inferior at the specified confidence level. This analysis is conducted across three distinct evaluation metrics: CRPS, SMAPE, and $\rho$-0.9 Risk, considering all datasets and forecast horizons. As illustrated in Fig. 9, our proposed models demonstrate superior robustness by consistently remaining within the top three positions across all of these metrics. Specifically, in terms of CRPS, GCEN achieved the lowest average rank of 2.18, establishing it as the overall best performer for probabilistic forecasting. MVEN and STEN secured the second and third spots respectively for the CRPS metric. For SMAPE, GCEN secured the top position with an average rank of 2.35. Finally, in the $\rho$-0.9 Risk evaluation, which emphasizes tail performance and high-quantile accuracy, the GCEN and STEN models maintain their competitive edge by achieving the second and third best ranks, following MVEN. The MCB analysis indicates that LSTM and NHiTS remain competitive among temporal forecasting methods. In contrast, the remaining baseline models such as Transformers, STGCN, DiffSTG, and GpGp, exhibit mean ranks that exceed the critical threshold, signifying statistically inferior performance

relative to the proposed models across the three evaluation metrics. This is due to the fact that most of these models work well with high-frequency (and high volume) data, as opposed to the nature of epidemic datasets.

# 7 Model Explainability

Temporal dependencies in our spatiotemporal engression models are handled using LSTM networks, which are generally less interpretable and considered as 'black-box' architectures (Guo et al., 2019). Since the proposed architectures (MVEN, GCEN, and STEN) comprise pre-additive noise structure, we can look into the dynamics of the random noise values that are driving the non-linear layers of the model to produce an ensemble of forecast trajectories. Moreover, the learned weights of the STAR-layer in STEN can also produce insights on the learned importance of each spatial lag, as discussed below.

## 7.1 Noise-Driven Generation of Forecast Trajectories

We take a peek into the internal dynamics of how injecting random noise values to the spatial embeddings (inputs to the temporal module) results in different plausible forecast trajectories for the same horizon. For the visualization, we sample $M = 30$ forecast trajectories from the trained GCEN, STEN, and MVEN models across three specific spatial nodes for demonstration purposes: Illinois (Node 13) for USA ILI, Santander (Node 28) for Colombia Dengue, and Osaka (Node 27) for Japan TB, respectively.

In each forward pass for GCEN and STEN, the stochastic driving force for each trajectory is derived from the exogenous random noise tensor $\boldsymbol{\eta} \sim \mathcal{N}(\mathbf{0}, \mathbf{I})^{p \times N \times D'}$ (for MVEN, $\boldsymbol{\eta} \sim \mathcal{N}(\mathbf{0}, \mathbf{I})^{p \times N \times D}$), where $p$ is the temporal lag, $N$ is the number of nodes, and $D'(D)$ is the dimension of spatial embeddings (inputs). By isolating a specific node $i$, the noise contribution is reduced to a tensor $\boldsymbol{\eta}_i \in \mathbb{R}^{p \times D'}$. To quantify the magnitude of the perturbation driving each ensemble member, we compute the Frobenius norm ($L_2$ norm across the

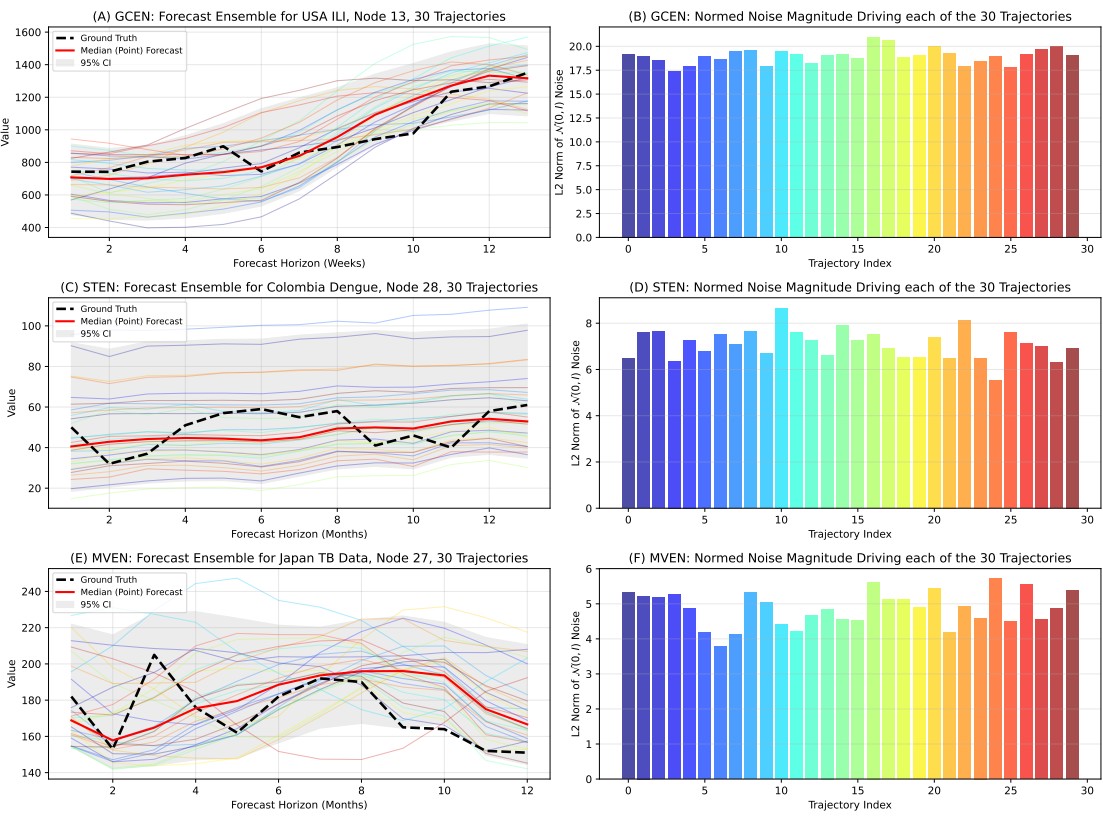

Fig. 10: Analysis of ensemble forecast dispersion and latent stochasticity in GCEN, STEN, and MVEN.

temporal and embedding dimensions):

$$\varepsilon_m = \|\boldsymbol{\eta}_i\|_F = \sqrt{\sum_{j=1}^{p} \sum_{k=1}^{D'} (\eta_{j,k})^2}.$$

This is done for all $M = 30$ trajectories, and we end up with a magnitude of noise value per trajectory, $\{\varepsilon_m\}_{m=1}^{M}$. Each noise $\varepsilon_m$ is responsible for the generation of the $m^{th}$ forecast trajectory. We visualize these dynamics in a dual-panel framework in Fig. 10: the left panel (A, C, E) presents the ensemble spaghetti plots, superimposing the median (solid red line), 95% confidence interval (gray shaded region), and ground truth (dashed black line) in original units; while the right panel (B, D, F) depicts the corresponding noise magnitudes $\varepsilon_m$ that produces each of the 30 trajectories. A unified color scheme enables a direct visual mapping between the latent noise intensity and the resulting forecast dispersion. The noise tensors are integrated into the standardized feature space during the forward pass. While the right panel depicts the Frobenius norms of these raw noise tensors, the forecast trajectories in the left panel are unstandardized to their original units to maintain physical interpretability and facilitate direct comparison with the ground truth. From the plots, it is evident that noise tensors with higher magnitudes produce forecasts that generally lie above the median, and vice versa.

## 7.2 Importance of Spatial Lags in STEN

A key feature of STEN is the set of learnable weight matrices ($\Phi_l \in \mathbb{R}^{D \times D'}$) that combine multi-scale spatial lags $W^l \mathbf{Y}_t$ (the notations are as before). The explanation of spatial lag importance offers a transparent view into the specific spatiotemporal dynamics learned by the STEN model for each epidemic. To quantify this, the importance score for each spatial lag $l$ is calculated as the Frobenius norm (Nie et al., 2010) of its corresponding learned weight matrix, $\Phi_l$:

$$\|\Phi_l\|_F = \sqrt{\sum_{i=1}^{D} \sum_{j=1}^{D'} \left(\phi_{i,j}^{(l)}\right)^2}.$$

These scores are then normalized by dividing each lag's individual score by the sum of all scores and converting it to a percentage, which represents its relative contribution to the final embedding:

$$P_l = \left(\frac{\|\Phi_l\|_F}{\sum_{k=0}^{L} \|\Phi_k\|_F}\right) \times 100\%,$$

where $L$ is the maximum number of spatial lags considered, and $P_l$ represents the normalized percentage importance of lag $l$ to the final spatiotemporal embedding. A high importance at the $l = 0$ (self) lag indicates a strong autoregressive component, suggesting that a region's future incidence is predominantly driven by its own historical momentum and internal dynamics. In contrast, high importance at $l = 1$ (neighbors) signifies a classic spatial diffusion effect, where the epidemic is primarily spread through contagion from immediately adjacent regions. Finally, significant importance at higher-order lags ($l > 1$) reveals more complex, non-local transmission routes, which can be attributed to multiple factors, such as transportation routes, hierarchical spread, and shared environmental or demographic factors.

Fig. 11 presents a comparative analysis of the learned spatial dependencies within the STEN model, visualized for each of the six epidemic datasets. The forecast horizons chosen are 30-day, 13-week, and 12-month for the daily, weekly, and monthly datasets, respectively. The $l = 0$ lag serves as the primary predictor for USA ILI (43.8%) and China TB (40.3%), indicating that these diseases are driven largely by local historical incidence. In contrast, Belgium COVID-19 exhibits a clear spatial diffusion profile, where the influence of first-order ($l = 1$, 35.2%) and second-order neighbors ($l = 2$, 35.3%) outweighs local history ($l = 0$, 29.6%). A more balanced importance distribution across all spatial lags is observed in Japan TB and Hungary chickenpox. For the Colombia dengue dataset, the prevalence of significant weights at $l = 0$ and $l = 2$ suggests that the forecast is driven by a combination of strong local autoregressive dependencies ($l = 0$) and second-order spatial influences ($l = 2$), the latter potentially reflecting shared regional climatic drivers across non-immediate neighbors. These results validate the STEN architecture's capability as an explainable framework, successfully learning and quantifying the diverse, disease-specific spatial contagion patterns inherent in the data.

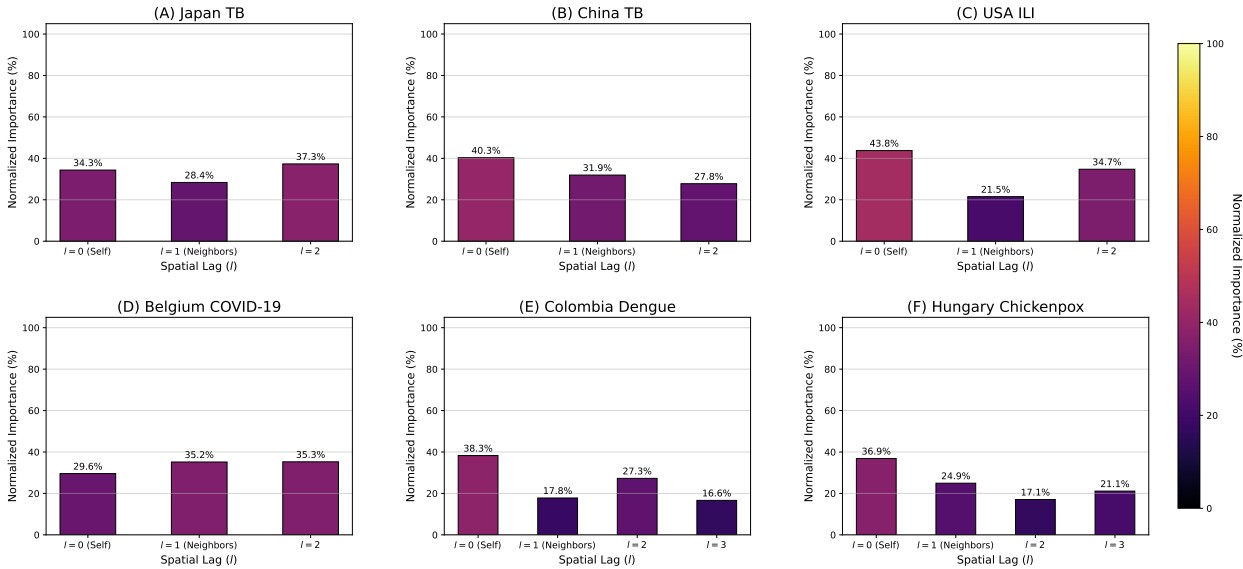

Fig. 11: Analysis of learned spatial lag importance from the STEN model across all six epidemic datasets. The maximum lag shown for each subplot (e.g., $l = 2$ for (A) and (D), $l = 3$ for (E)) corresponds to the `max_spatial_lag` hyperparameter selected for that specific dataset's model.

## 8 Ablation Study

In this section, we conduct an ablation study to evaluate the contributions of the spatial module, the pre-additive noise mechanism, and energy score optimization in probabilistic spatiotemporal forecasting. Specifically, we modify the proposed architectures in two key ways.

First, we replace the pre-additive noise structure with a post-additive formulation. In this configuration, inputs are sequentially processed by the spatial module (excluding MVEN) and the temporal module, with noise injected only into the final deterministic forecasts. We denote these models as MVEN-Post, GCEN-Post, and STEN-Post. Because these variants are still trained using the energy score, their forecast uncertainty profiles are shaped exclusively by the loss function, removing the temporal module's role in driving noise dynamics, as opposed to the pre-additive variants. Second, we retain the pre-additive noise structure but train the models using MSE rather than the energy score loss. These models are denoted by MVEN-MSE, GCEN-MSE, and STEN-MSE. During training, a forecast ensemble of 2 samples (consistent with our main experiments) is generated for the output window, and the expected MSE is computed by averaging the individual errors of each sample against the ground truth. Unlike the energy score, which incorporates a sharpness or dispersion term to prevent mode collapse and degenerate predictive distributions, this MSE objective explicitly penalizes sample variance, thereby increasing the model's vulnerability to these failure modes.

The results for the modified models, evaluated across all six epidemic datasets using identical hyperparameters, are detailed for short, medium, and long-term horizons in Tables 14-16 (Appendix G). The proposed architectures utilizing the pre-additive noise structure and energy score optimization consistently outperform both ablation variants. This is further supported by the MCB plot in Fig. 12, which demonstrates that MVEN, GCEN, and STEN are statistically superior to their post-additive and MSE-trained counterparts. PIT Q-Q plots for the modified models, shown for the USA ILI and Colombia dengue datasets in Fig. 14 (Appendix G), qualitatively reveal that the ablation variants yield inferiorly calibrated prediction intervals. Specifically, a staircase pattern emerges in these Q-Q plots, indicating that empirical quantiles remain static while theoretical quantiles increase. This denotes a degeneration of the forecast distribution into localized point masses, characterized by predictive trajectories that collapse toward their conditional mean with an

Fig. 12: MCB plots with respect to (A) CRPS and (B) MAE, ranking the proposed models and their modified variants considered in the ablation experiments.

acutely constrained variance. This behavior is particularly observed in the MSE-trained models, explicitly highlighting the role of the dispersion term within the energy score loss for maintaining well-calibrated predictive distributions. Finally, the overall importance of the spatial module is evident in the MCB plot (Fig. 12), where the purely temporal MVEN architecture is statistically surpassed by both GCEN and STEN based on CRPS and MAE.

## 9 Conclusion

This study addresses the challenge of probabilistic spatiotemporal forecasting by introducing three lightweight and generative deep learning frameworks: MVEN, GCEN, and STEN. By integrating the engression methodology with robust neural architectures, we established probabilistic frameworks capable of modeling both spatial and temporal dependencies while quantifying uncertainty. A theoretical contribution of this work is the formal derivation of geometric ergodicity and asymptotic stationarity for the closed-loop versions of the proposed generative processes. These proofs provide essential mathematical guarantees regarding the stability of the models. Empirically, the models were rigorously benchmarked across six diverse epidemiological datasets, covering various diseases and geographical topologies with varying sizes and granularities. The results consistently demonstrated the superiority of the spatiotemporal engression approaches over state-of-the-art baselines, even when data is scarce. Furthermore, the probabilistic nature of these frameworks enabled the generation of well-calibrated model-intrinsic PIs, as evidenced by the PIT analysis and Winkler scores. Our models consistently achieved better CRPS, Pinball scores, and Winkler scores against the baseline probabilistic models. More recently developed models such as NBeats (Oreshkin et al., 2019) and NHiTS (Challu et al., 2023) can also be used instead of LSTM as the temporal module for the proposed models. Moreover, the construction of the spatial weights matrix for STEN can be made flexible. Instead of using the inverse of Haversine distances, one may use other established techniques, such as the $k$-nearest neighbors, or neighbors of order $k$ based on contiguity (Moraga, 2023).

A primary advantage of the proposed engression-based models lies in their computational parsimony during both the training and inference phases. The popularly used STGCN (Yu et al., 2018) requires a large number of training samples, and is computationally heavy. The baseline spatiotemporal probabilistic frameworks such as GpGp, DiffSTG, and STESN, encounter prohibitive temporal overhead during ensemble generation from the trained models (see Table 9 in Appendix E.1). Consequently, the high latency inherent in traditional models precludes their effective deployment in near-real-time epidemiological surveillance and forecasting.

The STEN model, in particular, demonstrated distinct explainability advantages. By leveraging a STAR-inspired spatial layer, it allowed for the quantification of specific spatial lag importances, offering public health officials actionable insights into the relative dominance of local autoregressive trends versus spatial diffusion mechanisms. While the MCB test establishes the statistical superiority of GCEN in terms of predictive accuracy, STEN provides a more explainable framework for characterizing the underlying spatial dynamics of the process. These findings establish MVEN, GCEN, and STEN as robust, computationally viable tools for real-time disease surveillance, offering a reliable methodology for policy formulation under uncertainty. Beyond the current scope, these methodologies possess the inherent flexibility to model various phenomena, such as traffic flow dynamics and air quality forecasting in spatiotemporal domain.

**Limitations and Future Directions**

While the proposed frameworks offer distinct advantages in terms of intrinsic uncertainty quantification and computational efficiency, they are not without limitations. The current study only leverages the pre-ANM structure and the energy score loss to generate probabilistic forecasts for spatiotemporal data. A key benefit of engression as a distributional regression method lies in its extrapolability: ability to predict for inputs beyond the training support. Spatiotemporal extrapolability is not well-defined in the literature, to the best of our knowledge. Spatial extrapolation is generally defined as generating data or forecasting for unobserved locations (Hu et al., 2023), while the concept of extrapolation in time series is structurally analogous to handling distributional shifts and extreme events (Panja et al., 2026). In epidemiological contexts, this challenge frequently emerges when models trained predominantly on periods of low incidence are tasked with forecasting severe, high-spread outbreaks. While the classical engression setup imposes monotonicity-like assumptions on the non-linear function, such assumptions are usually violated for real-world spatiotemporal dynamics. Moreover, standard autoregressive or sequence-to-sequence models without explicit extreme-value handling fails to capture sudden surges. Thus, a primary constraint lies in the models' diminished capacity to accurately forecast extreme peaks, which is often a requirement in epidemiological modeling for anticipating and managing peak outbreak scenarios. This behavior is empirically evident in the results (for example, Shaanxi in China TB data; see Fig. 7 (D)). This can be considered as a future scope of research where extrapolation capabilities of engression can be explored by integrating extreme value modeling to capture extremes and handle distributional shifts in spatiotemporal data.

Second, the current study exclusively relies on historical epidemic incidence data, which omits the influence of exogenous determinants that fundamentally drive disease dynamics. While factors such as climatic variability for dengue or demographic density for COVID-19 and TB are known to modulate transmission, this work does not consider them as exogenous features. However, the proposed architecture is designed to be inherently generalizable to a multifaceted input space $\mathcal{Y} \in \mathbb{R}^{T \times N \times D}$, where the last dimension may encompass both the primary endogenous variables intended for forecasting and auxiliary exogenous covariates that provide supportive contextual information. Fully connected layers can be utilized to project the high-dimensional hidden states back into a refined output space that corresponds strictly to the target variables of interest.

Furthermore, our proposals are data-centric models. However, in epidemic forecasting, deep learning models integrated with epidemic knowledge (Barman et al., 2025) are often of interest. The current work can be extended to include epidemiological factors and integrated with compartmental models in future research. Such an integration would likely enhance predictive accuracy and provide more robust contextual information for public health decision-making under uncertainty.

Finally, in engression, the discrepancy between the target and the functional space is sensitive to the choice of the distance metric. For example, Shen & Meinshausen (2025); Kraft et al. (2026), and the present work utilizes the $L2$ norm in the energy score loss. Copula-based density regression might be viewed as an alternative. A copula operates on a topological space, avoiding such a choice of specific distance metric.

**Broader Impact Statement**

The proposed spatiotemporal engression models - MVEN, GCEN, and STEN, provide a robust framework for researchers and public health officials to generate reliable probabilistic forecasts of epidemic incidences. In a broad sense, the deployment of such models can significantly enhance public health preparedness by quantifying

uncertainty and providing best or worst-case scenarios, thereby informing strategic resource allocation and the design of tailored, timely interventions. The proposed models are capable of generating multiple plausible future trajectories for a given spatiotemporal problem, thereby omitting the need for model-agnostic wrappers such as conformal prediction for uncertainty quantification. Sometimes, our proposed models produce narrow and somewhat overconfident intervals (Fig. 8). However, this behavior can be readily resolved by applying established recalibration techniques (Rumack et al., 2022), guaranteeing robust uncertainty quantification. On the theoretical side, we investigate the ergodic properties of a pre-additive recurrent network process, supporting our analytical findings with numerical simulations and real data experiments.

## Data and Code Availability

The datasets used in this study are publicly available and briefly described in the main text and appendix. Code is available at `https://github.com/PyCoder913/stengression`, as well as through the `stengression` Python package[4].

## Acknowledgement

We sincerely thank the anonymous reviewers and the action editor for their constructive comments. We believe that incorporating their suggestions have greatly improved the presentation of this work.

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

## Appendix

## A Mathematical Proofs for Section 5.1

### A.1 Proof of Theorem 1

*Proof.* This proof follows a similar structure to that of Theorem 1 in Zhao et al. (2020), albeit they considered a post-additive noise process. Given that $S_t = \left(\mathbf{y}_t^\top, \mathbf{h}_t^\top\right)^\top \in \mathbb{R}^k$, where $k = N + N'$. The process is defined as $S_t = \mathcal{F}(S_{t-1} + \boldsymbol{\varepsilon}_t)$, where $\mathcal{F}: \mathbb{R}^k \to \mathbb{R}^k$ is a general non-linear function and $\boldsymbol{\varepsilon}_t = (\boldsymbol{\eta}_t^\top; \boldsymbol{\xi}_t^\top)^\top$, where $\boldsymbol{\eta}_t \in \mathbb{R}^N$, $\boldsymbol{\xi}_t \in \mathbb{R}^{N'}$. $\{S_t\}$ is a homogeneous Markov chain on the state space $(\mathbb{R}^k, \mathcal{B}^k, \lambda_k)$. We shall establish geometric ergodicity by first showing that $\{S_t\}$ is $\lambda_k$-irreducible, followed by proving that it satisfies Tweedie's drift criterion (equation 6; Tweedie (1983)).

For a set $A \in \mathcal{B}^k$ and $\mathbf{s} \in \mathbb{R}^k$, the transition probability is given by

$$\mathbb{P}(\mathbf{s}, A) = \mathbb{P}(S_t \in A | S_{t-1} = \mathbf{s}) = \int_{\mathbb{R}^k} \mathbb{I}_A\left(\mathcal{F}(\mathbf{s} + z)\right) f_{\boldsymbol{\varepsilon}}(z) \; dz,$$

where $f_{\boldsymbol{\varepsilon}}(\cdot)$ denotes the density of $\boldsymbol{\varepsilon}_t$, which is given by $f_{\boldsymbol{\varepsilon}}(z) = \prod_{i=1}^N f_{\boldsymbol{\eta}}(z_i) \prod_{j=N+1}^k f_{\boldsymbol{\xi}}(z_j)$, because the noise coordinates are independent. By assumptions (A1) and (A2), the transition kernel admits a density that is positive on sets of positive Lebesgue measure. Thus, $\{S_t\}$ is $\lambda_k$-irreducible.

Finally, it remains to show that there exists a small set $G$ with $\lambda_k(G) > 0$ and a non-negative continuous function $V(\mathbf{s})$ such that

$$\mathbb{E}[V(S_t) | S_{t-1} = \mathbf{s}] \leq \begin{cases} (1 - \delta)V(\mathbf{s}), & \mathbf{s} \notin G, \\ \widehat{M}, & \mathbf{s} \in G \end{cases} \tag{6}$$

for some $0 < \delta < 1$ and $0 < \widehat{M} < \infty$. By hypothesis, we have constants $0 < a < 1$ and $b$ such that $\|\mathcal{F}(z)\| \leq a\|z\| + b$. Define test function $V(\mathbf{s}) = 1 + \|\mathbf{s}\| > 0$. Then,

$$\begin{aligned} \mathbb{E}[V(S_t) | S_{t-1} = \mathbf{s}] &= \mathbb{E}[V(\mathcal{F}(\mathbf{s} + \boldsymbol{\varepsilon}_t))] \\ &= \mathbb{E}[1 + \|\mathcal{F}(\mathbf{s} + \boldsymbol{\varepsilon}_t)\|] \\ &\leq 1 + \mathbb{E}[a\|\mathbf{s} + \boldsymbol{\varepsilon}_t\| + b] \quad \text{(by hypothesis)} \\ &\leq 1 + \mathbb{E}[a\|\mathbf{s}\| + a\|\boldsymbol{\varepsilon}_t\| + b] \quad \text{(triangle inequality)} \\ &= 1 + a\|\mathbf{s}\| + b + a\mathbb{E}(\|\boldsymbol{\varepsilon}_t\|) \\ &= aV(\mathbf{s}) + 1 - a + b + a\mathbb{E}(\|\boldsymbol{\varepsilon}_t\|) \end{aligned}$$

Define

$$\delta = 1 - a - \frac{1 - a + b + a\mathbb{E}(\|\boldsymbol{\varepsilon}_t\|)}{V(\mathbf{s})}$$

and $G = \{\mathbf{s} : \|\mathbf{s}\| \leq L\}$ such that $V(\mathbf{s}) > 1 + \frac{b + \mathbb{E}(\|\boldsymbol{\varepsilon}_t\|)}{1-a} \; \forall \|\mathbf{s}\| > L$, which ensures that equation 6 holds true. Finally, as $\mathbb{E}[g(S_t) | S_{t-1} = \mathbf{s}]$ is continuous with respect to $\mathbf{s}$ for any bounded and continuous function $g$, it follows that $\{S_t\}$ is a Feller chain (Feigin & Tweedie, 1985). As $G$ is compact, by Feigin & Tweedie (1985), we have that $G$ is a small set. Hence, by Theorem 4(ii) and 1 in Tweedie (1983) and Feigin & Tweedie (1985) respectively, it follows that $\{S_t\}$ is geometrically ergodic. $\square$

### A.2 Proof of Theorem 2

*Proof.* This proof focuses on showing that the LSTM update equation satisfies the contraction property required by Theorem 1. Let $u = f_{spatial}(\mathbf{y}_{t-1}) + \boldsymbol{\eta}_t \in B_\infty^N$ by assumption (A4), $v = \mathbf{h}_{t-1} + \boldsymbol{\xi}_t \in B_\infty^{N'}$, $w = \mathbf{c}_{t-1} + \boldsymbol{\psi}_t$, and $x = o(u, v) \odot \tanh[i(u, v) \odot \tanh(W_{ch}v + W_{cy}u + b_c) + f(u, v) \odot w] \in B_\infty^{N'}$. Note that $x \in B_\infty^{N'} \implies \|x\|_1 \leq N'$. Our objective is to find $0 < a < 1$ and $b$ such that $\|\mathcal{F}_{LSTM}(u, v, w)\|_1 \leq$

$a \left\| (u, v, w) \right\|_1 + b$. Let $a = a_0 \in (0, 1)$ and $b = \widetilde{M} + 2N'$, where $\widetilde{M} := \sup_{x \in B_\infty^{N'}} \left\| g(W_{zh} x + b_z) \right\|_1 < \infty$, under assumption (A5). Thus, we have

$$
\begin{aligned}
&\left\| \mathcal{F}_{LSTM}(u, v, w) \right\|_1 - a_0 \left\| (u, v, w) \right\|_1 \\
&\leq \left\| g(W_{zh} x + b_z) \right\|_1 + \left\| x \right\|_1 + \left\| \mathbf{1}_{N'} + f(u, v) \odot w \right\|_1 - a_0 \left\| (u, v, w) \right\|_1 \\
&\leq \widetilde{M} + 2N' + \left\| f(u, v) \right\|_\infty \left\| w \right\|_1 - a_0 \left( \left\| u \right\|_1 + \left\| v \right\|_1 + \left\| w \right\|_1 \right) \\
&\leq \widetilde{M} + 2N' - a_0 \left\| u \right\|_1 - a_0 \left\| v \right\|_1 + \left( \left\| f(u, v) \right\|_\infty - a_0 \right) \left\| w \right\|_1.
\end{aligned}
$$

We have by Assumption (A6), that

$$
\begin{aligned}
\left\| f(u, v) \right\|_\infty &= \left\| \sigma(W_{fh} v + W_{fy} u + b_f) \right\|_\infty \\
&\leq \sigma \left( \left\| W_{fh} \right\|_\infty + \left\| W_{fy} \right\|_\infty + \left\| b_f \right\|_\infty \right) \\
&\leq a_0.
\end{aligned}
$$

Putting together, we have

$$
\left\| \mathcal{F}_{LSTM}(u, v, w) \right\|_1 - a_0 \left\| (u, v, w) \right\|_1 \leq \widetilde{M} + 2N' = b.
$$

By Theorem 1, the GCEN and STEN processes as defined in equation 5 is geometrically ergodic. $\qquad \square$

## B    Algorithms

Algorithms 1, 2, and 3 present the pseudo codes for the forward passes of GCEN, STEN, and MVEN respectively. Algorithm for the training procedure is provided in Algorithm 4.

## C    Energy Score Loss

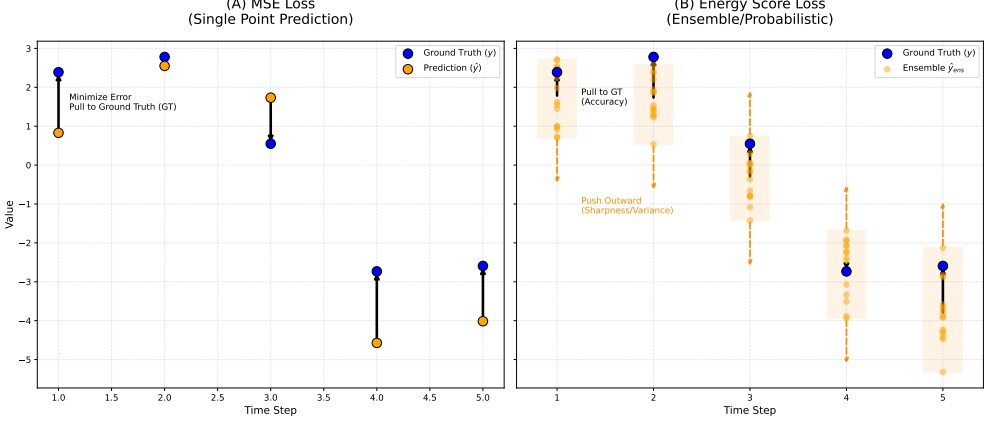

Fig. 13: A visual comparison between MSE and energy score loss using toy data. The accuracy term ($\mathcal{L}_{accuracy}$) in ES loss is analogous to MSE. The sharpness term is what makes energy score a probabilistic loss function.

Fig. 13 illustrates the components of the energy score loss in contrast to MSE. In the figure, the ground truth data points (blue) are generated using $y_t = 3 \sin(t) + \epsilon_t$, where $t = \{1, \dots, 5\}$ and $\epsilon_t \sim \mathcal{N}(0, 0.5)$. The corresponding predictions (orange) are generated using $\widehat{y}_t = y_t + \eta_t$, where $\eta_t \sim \mathcal{N}(0, 1.5)$. In panel (B), for each prediction $\widehat{y}_t$, the ensemble points constituting the orange shaded region is created by sampling 15 random points from $\mathcal{N}(\widehat{y}_t, 0.6)$.

The accuracy (first) term in the ES loss, analogous to MSE, minimizes the distance between the ground truth and predictions, effectively centering the sample cloud around the observed outcome. Conversely, the

---

**Algorithm 1** Graph Convolutional Engression Network (GCEN) Forward Pass

---

**Require:** Input tensor $\mathbf{Y} = \mathbf{Y}_{t-p+1:t} \in \mathbb{R}^{B \times p \times N \times D}$ (Batch, Lag, Nodes, Features); Adjacency info $\mathcal{G} = (\mathcal{V}, \mathcal{E})$; Noise distribution $\boldsymbol{\eta} \sim \mathcal{P}_z$.

**Ensure:** Single forecast trajectory $\widehat{\mathbf{Y}} = \widehat{\mathbf{Y}}_{t+1:t+q} \in \mathbb{R}^{B \times q \times N \times D}$.

    **// Phase 1: Spatial feature extraction (GCN)**
1:  Permute $\mathbf{Y}$ to node-first format: $\mathbf{Y} \leftarrow \text{Permute}(\mathbf{Y}, (N, B, p, D))$
2:  Compute transformed node features: $\mathbf{H}_{nodes} = \mathbf{Y}\Theta_{GCN}$
3:  Gather neighbor features $\widetilde{\mathbf{Y}}$ based on edges $\mathcal{E} \in \mathcal{G}$
4:  Aggregate neighbor messages: $\mathbf{M} = \text{Aggregate}(\mathbf{Y}, \widetilde{\mathbf{Y}})$                             ▷ Mean, Sum, or Max
5:  Combine and apply non-linearity: $\mathbf{Z} = \sigma(\text{Combine}(\mathbf{H}_{nodes}, \mathbf{M}\Theta_{GCN}))$             ▷ Add or Concat
6:  Permute back to batch-first format: $\mathbf{Z} \leftarrow \text{Permute}(\mathbf{Z}, (B, p, N, D'))$
    **// Phase 2: Stochastic noise injection (Engression)**
7:  **if** NoiseMode is *Additive* **then**
8:     Sample $\boldsymbol{\eta} \sim \mathcal{P}_z$ with shape of $\mathbf{Z}$
9:     $\mathbf{Z}_{noisy} = \mathbf{Z} + \boldsymbol{\eta}$
10: **else**
11:     **if** NoiseMode is *Concatenation* **then**
12:         Sample $\boldsymbol{\eta} \sim \mathcal{P}_z$ with shape $(B, p, N, D_{noise})$
13:         $\mathbf{Z}_{noisy} = [\mathbf{Z} \parallel \boldsymbol{\eta}]$                  ▷ Concatenate along feature dimension
14:     **end if**
15: **end if**
    **// Phase 3: Temporal processing (LSTM)**
16: Reshape $\mathbf{Z}_{noisy}$ for sequential processing: $\mathbf{Z}_{noisy} \in \mathbb{R}^{p \times (B \cdot N) \times D_{net}}$
17:                                            ▷ $D_{net} = D'$ if NoiseMode is Additive, else $D_{net} = D' + D_{noise}$
18: $(\mathbf{h}_n, \mathbf{c}_n) = \text{LSTM}(\mathbf{Z}_{noisy})$
19: Extract final hidden state: $\mathbf{h}_{final} = \mathbf{h}_n[-1]$                     ▷ Shape: $(B \cdot N) \times D_{hidden}$
    **// Phase 4: Forecast generation**
20: Project to forecast horizon: $\widehat{\mathbf{Y}} = \text{Dense}(\mathbf{h}_{final})$            ▷ Shape: $(B \cdot N) \times (q \cdot D)$
21: Reshape and permute: $\widehat{\mathbf{Y}} \leftarrow \text{Reshape}(\widehat{\mathbf{Y}}, (B, q, N, D))$
22: **return** $\widehat{\mathbf{Y}}$

---

**Algorithm 2** Spatio-Temporal Engression Network (STEN) Forward Pass

---

**Require:** Input tensor $\mathbf{Y} = \mathbf{Y}_{t-p+1:t} \in \mathbb{R}^{B \times p \times N \times D}$ (Batch, Lag, Nodes, Features); Spatial weights matrices $\{W^l\}_{l=0}^{L}$ for spatial lags 0 to $L$; Noise distribution $\boldsymbol{\eta} \sim \mathcal{P}_z$.

**Ensure:** Single forecast trajectory $\widehat{\mathbf{Y}} = \widehat{\mathbf{Y}}_{t+1:t+q} \in \mathbb{R}^{B \times q \times N \times D}$.

    **// Phase 1: Spatial embedding (STAR-layer)**
1:  $\mathbf{Z}_{total} \leftarrow \mathbf{0} \in \mathbb{R}^{B \times p \times N \times D'}$
2:  **for** $l = 0$ **to** $L$ **do**                                  ▷ Iterate through spatial lags
3:     Compute spatial aggregation: $\mathbf{Y}_\tau^{(l)} = W^l \mathbf{Y}_\tau$     ▷ Performed for each $\tau \in \{t-p+1, \ldots, t\}$; $\mathbf{Y}_\tau \in \mathbb{R}^{N \times D}$
4:     Apply learnable transformation: $\mathbf{H}^{(l)} = \mathbf{Y}_\tau^{(l)} \Phi_l$     ▷ $\Phi_l \in \mathbb{R}^{D \times D'}$ is weight matrix for lag $l$
5:     $\mathbf{Z}_{total} = \mathbf{Z}_{total} + \mathbf{H}^{(l)}$                   ▷ Summation across all spatial lags
6:  **end for**
7:  $\widetilde{\mathbf{Z}} = \text{ReLU}(\mathbf{Z}_{total})$                               ▷ Final spatial embedding
8:  Phase 2 to Phase 4 with the spatial embedding $\widetilde{\mathbf{Z}}$ follows from Steps 7-22 in Algorithm 1.

---

---

**Algorithm 3** Multivariate Engression Network (MVEN) Forward Pass

---

**Require:** Input tensor $\mathbf{Y} = \mathbf{Y}_{t-p+1:t} \in \mathbb{R}^{B \times p \times N \times D}$ (Batch, Lag, Nodes, Features); Noise distribution $\boldsymbol{\eta} \sim \mathcal{P}_z$.

**Ensure:** Single forecast trajectory $\widehat{\mathbf{Y}} = \widehat{\mathbf{Y}}_{t+1:t+q} \in \mathbb{R}^{B \times q \times N \times D}$.

     **// Phase 1: Stochastic noise injection (Engression)**

1: **if** NoiseMode is *Additive* **then**
2:     Sample $\boldsymbol{\eta} \sim \mathcal{P}_z$ with shape $(B, p, N, D)$
3:     $\mathbf{Y}_{noisy} = \mathbf{Y} + \boldsymbol{\eta}$
4: **else**
5:     **if** NoiseMode is *Concatenation* **then**
6:         Sample $\boldsymbol{\eta} \sim \mathcal{P}_z$ with shape $(B, p, N, D_{noise})$
7:         $\mathbf{Y}_{noisy} = [\mathbf{Y} \parallel \boldsymbol{\eta}]$            ▷ Concatenate along feature dimension $D$
8:     **end if**
9: **end if**
10: The rest of the algorithm follows from steps 16-22 in Algorithm 1 with $\mathbf{Y}_{noisy}$ instead of $\mathbf{Z}_{noisy}$.

---

**Algorithm 4** Training Procedure of the Proposed Models

---

**Require:** Training dataset: $\mathcal{D} = \{(\mathbf{Y}_{t-p+1:t}, \mathbf{Y}_{t+1:t+q})\}$, $p \leq t \leq T - q$; Ensemble size: $M$; Number of epochs: $E$; Learning rate: $\varepsilon$; Loss function: $\mathcal{L}$ (energy score loss).

**Ensure:** Trained model parameters $\Theta = \{\Theta_{\text{Spatial}}, \Theta_{\text{LSTM}}, W_{out}\}$, where $\Theta_{\text{Spatial}}, \Theta_{\text{LSTM}}$, and $W_{out}$ are the weight matrices for the spatial component (GCN/STAR; ignore for MVEN), LSTM, and dense layers respectively.

1: Initialize $\Theta$ using Xavier initialization;
2: Form batches of inputs and targets. Denote by $\mathcal{X} = [\mathbf{Y}_{t-p+1:t}] \in \mathbb{R}^{B \times p \times N \times D}$ the input sequences for different time points $t$ stacked over the first (batch) dimension, and let $\mathcal{Y} = [\mathbf{Y}_{t+1:t+q}] \in \mathbb{R}^{B \times q \times N \times D}$ be the corresponding batch of target sequences;
3: **for** epoch $= 1$ **to** $E$ **do**
4:     **for** each batch $(\mathcal{X}, \mathcal{Y}) \in \mathcal{D}$ **do**
5:         $\mathbf{S} \leftarrow \emptyset$               ▷ Initialize empty ensemble list
         **// Ensemble generation during in-sample prediction. Typically, set $M = 2$**
6:         **for** $m = 1$ **to** $M$ **do**
7:             $\widehat{\mathcal{Y}}^{(m)} \leftarrow \text{Model}(\mathcal{X}; \Theta)$         ▷ Forward pass through GCEN/STEN/MVEN
8:             $\mathbf{S} \leftarrow \mathbf{S} \cup \{\widehat{\mathcal{Y}}^{(m)}\}$     ▷ Append to ensemble list; each forward pass produces a different forecast trajectory
9:         **end for**
10:         Form the forecast ensemble tensor by stacking the elements of $\mathbf{S}$ along the first dimension: $\widehat{\mathcal{Y}}_{ensemble} \in \mathbb{R}^{M \times B \times q \times N \times D}$
         **// Loss computation and backpropagation**
11:         loss $\leftarrow \mathcal{L}(\mathcal{Y}, \widehat{\mathcal{Y}}_{ensemble})$         ▷ Probabilistic loss over ensemble
12:         $\Theta \leftarrow \Theta - \varepsilon \cdot \nabla_{\Theta}(\text{loss})$         ▷ Update weights via Adam/SGD
13:     **end for**
14:     Record loss per epoch for convergence monitoring;
15: **end for**
16: **return** $\Theta$

---

sharpness (second) term is the average distance between prediction pairs. This is subtracted within the loss function. Maximizing this distance during training encourages predictive diversity and prevents mode collapse, where the model might otherwise converge on identical values. This forces the model to generate a spread of plausible futures that accurately reflect underlying uncertainty. The energy score loss inherently balances exploitation - anchoring the predictive distribution to the ground truth via the accuracy term, with exploration - effectively penalizing mode collapse to ensure the ensemble adequately spans the underlying stochasticity.

## D Global Temporal Properties of the Datasets

We present the tables summarizing the global temporal properties for the datasets used in our experimental setup in Tables 3-8. Our characterization included standard descriptive statistics (mean, standard deviation, skewness, and kurtosis) for each nodal time series. We also assessed three global characteristics for each node: stationarity, long-range dependence, and nonlinearity. Stationarity, a key assumption for classical forecasting models like ARMA, was evaluated using the Augmented Dickey-Fuller (ADF) test (Dickey & Fuller, 1979) at a significance level of 0.05; a time series is considered non-stationary for a p-value $> 0.05$. Long-range dependence, which reflects the self-similarity of a time series and is crucial for probabilistic modeling, was quantified using the Hurst exponent. Values below 0.5 indicate mean-reverting (anti-persistent) behavior, 0.5 suggests a random walk (Brownian motion), and values above 0.5 indicate a trending (persistent) series. Finally, to assess the presence of nonlinear dynamics at each node, Teräsvirta's neural network test (Teräsvirta et al., 1993) at a significance level of 0.05 was employed. Under this framework, the null hypothesis postulates linearity; therefore, a $p$-value $< 0.05$ signifies a statistically significant departure from linearity, confirming the nonlinear nature of the time series.

The analysis revealed distinct behaviors across the datasets. The tuberculosis time series for both Japan and China were found to be predominantly non-stationary while uniformly exhibiting long-range dependence. Specifically, all prefectures in Japan (except Yamagata, Kyoto, and Okinawa) and all provinces in China (except Jiangxi and Hainan) were non-stationary. Japan's TB data typically exhibits a decreasing trend (and hence, linearity in many nodes), reflecting Japan's effective and aggressive combat against the disease[5]. In contrast, the majority of nodes in the remaining datasets were stationary. All states in the USA ILI dataset were stationary, with the exception of New Jersey, and most exhibited long-range dependence. Similarly, all provinces in Belgium's COVID-19 data and all counties in Hungary's chickenpox data were stationary and showed strong long-range dependence, with only a few locations in Belgium (Brabant Wallon, Luxembourg, and Namur) identified as mean-reverting series.

## E Baseline Models

In this section, we briefly describe the baseline models used to compare the model performances on real datasets.

1. **LSTM:** The Long Short-Term Memory (LSTM) network (Hochreiter & Schmidhuber, 1997) is a specialized architecture of RNNs designed to overcome the vanishing gradient problem, thereby enabling the modeling of long-range temporal dependencies. The memory cell state in LSTM is regulated by three distinct gating mechanisms: the forget gate, which determines what information to discard from the previous state; the input gate, which controls the integration of new information; and the output gate, which modulates the information passed to the next hidden state. Although LSTMs are popular for modeling long-range dependencies, Zhao et al. (2020) show that they are short memory processes under mild assumptions.

2. **NHiTS:** The Neural Hierarchical Interpolation for Time Series (NHiTS) (Challu et al., 2023) is an advanced neural forecasting architecture designed to address the challenges of long-horizon forecasting by significantly reducing computational complexity and mitigating the effects of high-frequency noise. Building upon the basis-expansion principles of NBeats (Oreshkin et al., 2019), NHiTS introduces a

---

[5]https://www.mofa.go.jp/announce/announce/2008/7/1182066_1030.html

Table 3: Global characteristics of Japan's TB dataset.

| Node No. | Node Name | Mean | STD | Range | Skewness | Kurtosis | ADF p-value | Hurst Exponent | Teräsvirta p-value |
|---|---|---|---|---|---|---|---|---|---|
| 1 | Hokkaido | 71 | 25.74 | [26, 163] | 0.8 | 0 | 0.24 | 0.81 | 0.00 |
| 2 | Aomori | 24.05 | 9.98 | [8, 61] | 0.87 | 0.56 | 0.45 | 0.76 | 0.59 |
| 3 | Iwate | 17.5 | 7.86 | [5, 48] | 1.1 | 1.51 | 0.32 | 0.71 | 0.09 |
| 4 | Miyagi | 27.59 | 11.55 | [6, 71] | 0.92 | 0.6 | 0.5 | 0.73 | 0.01 |
| 5 | Akita | 14.52 | 6.91 | [2, 38] | 0.82 | 0.47 | 0.27 | 0.71 | 0.63 |
| 6 | Yamagata | 13.43 | 5.99 | [3, 36] | 1.31 | 2.08 | 0.01 | 0.72 | 0.49 |
| 7 | Fukushima | 26.02 | 11.3 | [7, 75] | 1.11 | 1.65 | 0.71 | 0.73 | 0.02 |
| 8 | Ibaragi | 43.38 | 13.17 | [22, 84] | 0.68 | -0.19 | 0.77 | 0.71 | 0.39 |
| 9 | Tochigi | 27.51 | 9.59 | [9, 59] | 0.77 | 0.21 | 0.07 | 0.67 | 0.07 |
| 10 | Gunnma | 25.23 | 10.08 | [10, 70] | 1.18 | 1.99 | 0.59 | 0.7 | 0.24 |
| 11 | Saitama | 111.5 | 27.39 | [53, 232] | 0.91 | 1.35 | 0.81 | 0.74 | 0.13 |
| 12 | Chiba | 106.15 | 30.01 | [53, 190] | 0.64 | -0.15 | 0.48 | 0.78 | 0.09 |
| 13 | Tokyo | 289.13 | 63.62 | [165, 544] | 0.5 | 0.51 | 0.95 | 0.77 | 0.44 |
| 14 | Kanagawa | 150.27 | 37.59 | [89, 306] | 0.94 | 1.11 | 0.71 | 0.76 | 0.50 |
| 15 | Niigata | 31.31 | 11.97 | [11, 65] | 0.66 | -0.18 | 0.69 | 0.72 | 0.09 |
| 16 | Toyoama | 17.44 | 7.54 | [5, 48] | 0.9 | 0.77 | 0.5 | 0.7 | 0.01 |
| 17 | Ishikawa | 18.12 | 7.35 | [4, 47] | 1.01 | 1.31 | 0.63 | 0.69 | 0.13 |
| 18 | Fukui | 11.81 | 4.76 | [2, 29] | 0.69 | 0.63 | 0.87 | 0.64 | 0.24 |
| 19 | Yamanashi | 9.9 | 4.38 | [2, 28] | 0.76 | 0.85 | 0.05 | 0.7 | 0.71 |
| 20 | Nagano | 20.34 | 6.33 | [7, 42] | 0.68 | 1.05 | 0.23 | 0.66 | 0.44 |
| 21 | Gifu | 42.79 | 14.53 | [18, 107] | 1.05 | 1.65 | 0.65 | 0.73 | 0.10 |
| 22 | Shizuoka | 61.15 | 17.83 | [30, 130] | 0.45 | 0 | 0.55 | 0.72 | 0.11 |
| 23 | Aichi | 151.2 | 37.73 | [76, 280] | 0.64 | 0.21 | 0.67 | 0.82 | 0.00 |
| 24 | Mie | 30.3 | 10.87 | [8, 68] | 0.7 | 0.23 | 0.69 | 0.69 | 0.01 |
| 25 | Shiga | 20.75 | 7.46 | [6, 49] | 0.79 | 0.85 | 0.58 | 0.71 | 0.01 |
| 26 | Kyoto | 59.57 | 28.43 | [20, 178] | 1.98 | 4.58 | 0.03 | 0.76 | 0.22 |
| 27 | Osaka | 280.49 | 113.18 | [119, 623] | 0.94 | -0.11 | 0.4 | 0.87 | 0.01 |
| 28 | Hyogo | 127.13 | 48.09 | [57, 320] | 1.1 | 0.63 | 0.42 | 0.8 | 0.00 |
| 29 | Nara | 27.9 | 9.69 | [9, 58] | 0.68 | -0.11 | 0.58 | 0.73 | 0.04 |
| 30 | Wakayama | 22.69 | 9.41 | [4, 57] | 1.03 | 0.93 | 0.56 | 0.73 | 0.00 |
| 31 | Tottori | 9.38 | 4.35 | [1, 27] | 0.91 | 1.32 | 0.23 | 0.61 | 0.23 |
| 32 | Shimane | 11.66 | 4.28 | [3, 27] | 0.56 | 0.36 | 0.42 | 0.66 | 0.64 |
| 33 | Okayama | 30.54 | 10.85 | [11, 63] | 0.75 | 0.34 | 0.7 | 0.72 | 0.26 |
| 34 | Hiroshima | 44.77 | 14.48 | [17, 91] | 0.77 | 0.03 | 0.7 | 0.76 | 0.62 |
| 35 | Yamaguchi | 27.09 | 12.09 | [9, 71] | 1.05 | 0.83 | 0.46 | 0.78 | 0.26 |
| 36 | Tokushima | 16.94 | 7.43 | [5, 42] | 1.08 | 1.06 | 0.46 | 0.75 | 0.03 |
| 37 | Kagawa | 19.31 | 7.87 | [4, 52] | 0.84 | 0.91 | 0.44 | 0.72 | 0.02 |
| 38 | Ehime | 23.14 | 8.83 | [8, 49] | 0.83 | 0.42 | 0.4 | 0.75 | 0.34 |
| 39 | Kochi | 14.84 | 6.82 | [1, 40] | 0.82 | 0.34 | 0.34 | 0.7 | 0.13 |
| 40 | Fukuoka | 97.16 | 30.8 | [38, 201] | 0.81 | 0.24 | 0.67 | 0.81 | 0.10 |
| 41 | Saga | 15.06 | 5.29 | [2, 33] | 0.36 | 0.2 | 0.37 | 0.62 | 0.08 |
| 42 | Nagasaki | 31.32 | 10.01 | [11, 63] | 0.63 | 0.02 | 0.64 | 0.69 | 0.65 |
| 43 | Kumamoto | 30.69 | 8.46 | [14, 60] | 0.64 | 0.3 | 0.76 | 0.64 | 0.08 |
| 44 | Oita | 24.4 | 9.64 | [9, 69] | 1.5 | 3.08 | 0.22 | 0.72 | 0.14 |
| 45 | Miyazaki | 19.09 | 8.2 | [4, 50] | 1.05 | 0.91 | 0.21 | 0.66 | 6.54E-07 |
| 46 | Kagoshima | 34.04 | 11.7 | [10, 65] | 0.58 | -0.26 | 0.37 | 0.74 | 0.45 |
| 47 | Okinawa | 24.51 | 6.86 | [9, 45] | 0.35 | -0.28 | 0 | 0.63 | 0.31 |

hierarchical multi-rate sampling approach that employs distinct interpolation layers at various scales. This architecture decomposes the signal by processing different frequency components across its blocks, utilizing subsampling and subsequent interpolation to reconstruct the forecast.

Table 4: Global characteristics of China's TB dataset.

| Node No. | Node Name | Mean | STD | Range | Skewness | Kurtosis | ADF p-value | Hurst Exponent | Teräsvirta p-value |
|---|---|---|---|---|---|---|---|---|---|
| 1 | Beijing | 576.68 | 70.15 | [397, 706] | -0.29 | -0.61 | 0.62 | 0.56 | 0.00 |
| 2 | Tianjin | 266 | 38.01 | [171, 344] | -0.4 | 0.1 | 0.63 | 0.58 | 0.59 |
| 3 | Hebei | 2731.63 | 341.66 | [1755, 3369] | -0.42 | -0.06 | 0.99 | 0.6 | 0.00 |
| 4 | Shanxi | 1218.1 | 193.28 | [854, 1625] | 0 | -0.63 | 0.78 | 0.64 | 0.00 |
| 5 | Inner Mongolia | 1040.45 | 168.35 | [733, 1433] | 0.47 | -0.4 | 0.97 | 0.76 | 0.00 |
| 6 | Liaoning | 1986.65 | 258.02 | [1357, 2584] | 0.12 | -0.41 | 0.63 | 0.69 | 0.00 |
| 7 | Jilin | 1209.37 | 231.38 | [737, 1732] | 0.3 | -0.49 | 0.68 | 0.65 | 0.00 |
| 8 | Heilongjiang | 2507.3 | 488.15 | [1489, 3598] | 0.05 | -0.62 | 1 | 0.56 | 0.00 |
| 9 | Shanghai | 547.65 | 65.84 | [388, 657] | -0.3 | -0.8 | 0.66 | 0.55 | 0.02 |
| 10 | Jiangsu | 2418.22 | 371.02 | [1609, 3193] | -0.18 | -0.43 | 0.96 | 0.57 | 0.00 |
| 11 | Zhejiang | 2281.55 | 301.3 | [1731, 2825] | -0.11 | -1.11 | 0.49 | 0.57 | 0.00 |
| 12 | Anhui | 2893.8 | 353.49 | [2094, 3721] | 0.16 | -0.5 | 0.98 | 0.63 | 0.00 |
| 13 | Fujian | 1425.73 | 177.96 | [1046, 1807] | -0.14 | -0.62 | 0.82 | 0.63 | 0.00 |
| 14 | Jiangxi | 3101.47 | 2882.15 | [1976, 24882] | 7.37 | 53.22 | 0 | 0.55 | 0.51 |
| 15 | Shandong | 2588.25 | 459.76 | [1670, 4059] | 0.73 | 0.95 | 0.75 | 0.57 | 0.00 |
| 16 | Henan | 4780.77 | 522.39 | [3242, 5769] | -0.34 | 0.25 | 0.84 | 0.62 | 0.00 |
| 17 | Hubei | 3549.92 | 497.49 | [2518, 4550] | -0.14 | -0.7 | 1 | 0.6 | 0.00 |
| 18 | Hunan | 4507.5 | 566.54 | [3443, 5801] | 0.22 | -0.77 | 0.69 | 0.63 | 0.00 |
| 19 | Guangdong | 6556.75 | 795.79 | [4484, 7976] | -0.44 | -0.54 | 0.97 | 0.61 | 0.00 |
| 20 | Guangxi | 3626.18 | 556.43 | [2623, 5086] | 0.35 | -0.37 | 0.82 | 0.59 | 0.00 |
| 21 | Hainan | 693.97 | 84.28 | [485, 857] | -0.34 | -0.42 | 0 | 0.58 | 0.30 |
| 22 | Chongqing | 1889.9 | 289.03 | [1328, 2645] | 0.32 | 0.03 | 0.81 | 0.63 | 0.00 |
| 23 | Sichuan | 4412.45 | 571.64 | [3307, 6044] | 0.36 | -0.02 | 1 | 0.6 | 0.00 |
| 24 | Guizhou | 3708.45 | 545.25 | [2530, 5251] | 0.32 | 0.43 | 1 | 0.55 | 0.00 |
| 25 | Yunnan | 2237.87 | 297.6 | [1616, 3004] | 0.29 | 0.57 | 0.99 | 0.59 | 0.00 |
| 26 | Tibet | 413.55 | 85.75 | [176, 593] | -0.44 | 0.3 | 0.98 | 0.67 | 0.00 |
| 27 | Shaanxi | 1821.4 | 242.03 | [1462, 2855] | 1.38 | 3.68 | 0.72 | 0.62 | 0.06 |
| 28 | Gansu | 1181.68 | 282.9 | [622, 2039] | 0.37 | 0.24 | 1 | 0.74 | 0.00 |
| 29 | Qinghai | 616.07 | 116.13 | [375, 964] | 0.35 | -0.01 | 0.1 | 0.71 | 0.06 |
| 30 | Ningxia | 236.67 | 49.81 | [144, 394] | 0.74 | 0.97 | 0.62 | 0.73 | 0.04 |
| 31 | Xinjiang | 4153.5 | 1370.1 | [2212, 8628] | 1.46 | 1.87 | 1 | 0.69 | 0.43 |

3. **Transformers:** The Transformer architecture, originally introduced by Vaswani et al. (2017) in the context of natural language processing, has revolutionized time series forecasting by replacing recurrent structures with self-attention mechanisms. Unlike sequential models, Transformers process entire sequences in parallel, utilizing multi-head attention to capture dependencies between time steps regardless of their temporal distance. This allows the model to assign dynamic weights to past observations based on their relevance to the current forecast horizon. To preserve temporal order, the architecture incorporates positional encodings into the input embeddings.

4. **TCN:** The Temporal Convolutional Network (TCN) (Bai, 2018) is a specialized 1D convolutional architecture designed for sequence modeling that serves as an alternative to RNNs. TCNs are defined by two primary characteristics: the use of causal convolutions, which ensure no information leakage from the future to the past, and dilated convolutions, which allow the network's receptive field to grow exponentially with its depth. This enables the model to capture long-range temporal dependencies with significantly fewer layers than standard convolutions. Moreover, the architecture incorporates residual blocks to facilitate the training of deeper networks and mitigate the vanishing gradient problem. TCNs benefit from massive parallelism during training, as convolutions can be computed across the entire sequence simultaneously, making them highly efficient for large-scale temporal forecasting tasks.

5. **STARMA:** The Space-Time Autoregressive Moving Average (STARMA) (Pfeifer & Deutrch, 1980) model is a specialized class of spatiotemporal models designed to capture systematic dependencies in data observed across both space and time. It extends the univariate Box-Jenkins ARIMA framework by representing a single variable as a linear combination of its past observations and errors, lagged in both dimensions.

Table 5: Global characteristics of USA's ILI dataset.

| Node No. | Node Name | Mean | STD | Range | Skewness | Kurtosis | ADF p-value | Hurst Exponent | Teräsvirta p-value |
|---|---|---|---|---|---|---|---|---|---|
| 1 | Alabama | 336.61 | 347.18 | [13, 1938] | 2.13 | 4.93 | 0 | 0.76 | 0.01 |
| 2 | Alaska | 18.16 | 18.87 | [0, 116] | 2.11 | 5.28 | 0 | 0.72 | 0.04 |
| 3 | Arizona | 304.68 | 225.1 | [56, 1297] | 1.6 | 2.93 | 0 | 0.77 | 0.02 |
| 4 | Arkansas | 65.26 | 87.15 | [0, 525] | 2.13 | 4.83 | 0 | 0.72 | 2.52E-09 |
| 5 | California | 833.96 | 461.23 | [91, 2452] | 1.07 | 0.8 | 0 | 0.72 | 0.20 |
| 6 | Colorado | 123.79 | 145.14 | [0, 891] | 2.29 | 5.86 | 0 | 0.78 | 0.02 |
| 7 | Connecticut | 42.34 | 53.39 | [0, 272] | 1.85 | 3.59 | 0 | 0 | 0.29 |
| 8 | Delaware | 15.71 | 27.57 | [0, 157] | 2.69 | 7.77 | 0 | 0 | 0.33 |
| 9 | Florida | 284.18 | 194.97 | [9, 1165] | 1.29 | 2.34 | 0 | 0.77 | 0.46 |
| 10 | Georgia | 557.41 | 468.69 | [1, 2489] | 1.5 | 2.17 | 0.05 | 0.82 | 0.02 |
| 11 | Hawaii | 87.91 | 100.43 | [2, 603] | 2.05 | 4.77 | 0 | 0.88 | 0.00 |
| 12 | Idaho | 54.03 | 59.9 | [0, 301] | 1.73 | 3.1 | 0 | 0 | 0.42 |
| 13 | Illinois | 901.93 | 511.59 | [244, 3569] | 1.5 | 3.69 | 0 | 0.71 | 0.00 |
| 14 | Indiana | 109.56 | 135.33 | [5, 902] | 3.12 | 11.77 | 0 | 0.74 | 0.11 |
| 15 | Iowa | 33.11 | 45.32 | [0, 302] | 2.41 | 7.89 | 0 | 0 | 0.08 |
| 16 | Kansas | 94.88 | 117.49 | [0, 555] | 1.87 | 2.97 | 0 | 0.74 | 0.01 |
| 17 | Kentucky | 96.84 | 133.63 | [0, 746] | 2.21 | 5.45 | 0 | 0.73 | 0.00 |
| 18 | Louisiana | 810.6 | 620.94 | [35, 3473] | 1.61 | 2.94 | 0 | 0.8 | 0.01 |
| 19 | Maine | 45.82 | 32.63 | [2, 213] | 2.12 | 6 | 0 | 0.79 | 0.00 |
| 20 | Maryland | 125.27 | 77.21 | [4, 467] | 1.4 | 2.52 | 0 | 0.74 | 0.00 |
| 21 | Massachusetts | 263.47 | 152.1 | [22, 961] | 1.49 | 2.77 | 0 | 0.77 | 1.69E-05 |
| 22 | Michigan | 173.43 | 142.71 | [10, 720] | 1.48 | 2.07 | 0 | 0.78 | 0.10 |
| 23 | Minnesota | 61.06 | 47.31 | [2, 302] | 1.55 | 3.2 | 0 | 0.76 | 0.69 |
| 24 | Mississippi | 559.02 | 354.32 | [104, 2359] | 1.76 | 3.92 | 0 | 0.79 | 0.00 |
| 25 | Missouri | 106.36 | 111.77 | [0, 528] | 1.37 | 1.49 | 0 | 0.72 | 0.05 |
| 26 | Montana | 6.69 | 10.78 | [0, 71] | 2.67 | 8.43 | 0 | 0 | 0.12 |
| 27 | Nebraska | 46.29 | 63.62 | [1, 454] | 3.18 | 13.1 | 0 | 0.79 | 0.79 |
| 28 | Nevada | 160.62 | 163.92 | [6, 784] | 1.78 | 2.84 | 0 | 0.79 | 0.67 |
| 29 | New Hampshire | 13.37 | 18.4 | [0, 149] | 3.18 | 13.92 | 0 | 0.75 | 0.00 |
| 30 | New Jersey | 138.89 | 136.78 | [1, 894] | 2.1 | 6.9 | 0.1 | 0.86 | 0.07 |
| 31 | New Mexico | 190.13 | 140.28 | [8, 699] | 1.55 | 2.04 | 0 | 0.72 | 0.05 |
| 32 | New York | 163.67 | 164.81 | [0, 744] | 1.41 | 1.36 | 0 | 0.82 | 0.01 |
| 33 | North Carolina | 194.97 | 230.75 | [0, 1406] | 2.72 | 9.11 | 0 | 0.73 | 0.03 |
| 34 | North Dakota | 17.51 | 24.58 | [0, 170] | 2.58 | 9.21 | 0 | 0.79 | 2.29E-06 |
| 35 | Ohio | 116.13 | 120.59 | [1, 835] | 2.27 | 6.64 | 0 | 0.73 | 1.21E-07 |
| 36 | Oklahoma | 88.88 | 96.08 | [0, 511] | 1.45 | 1.86 | 0 | 0.72 | 2.45E-05 |
| 37 | Oregon | 31.74 | 35.21 | [0, 219] | 1.96 | 4.67 | 0 | 0.74 | 0.00 |
| 38 | Pennsylvania | 208.09 | 158.48 | [40, 893] | 1.85 | 3.68 | 0 | 0.77 | 4.89E-06 |
| 39 | Rhode Island | 28.74 | 43.1 | [0, 286] | 2.87 | 9.79 | 0 | 0.73 | 1.70E-05 |
| 40 | South Carolina | 49.02 | 80.72 | [0, 726] | 4.28 | 25.24 | 0 | 0.73 | 0.01 |
| 41 | South Dakota | 65.56 | 53.84 | [0, 275] | 1.21 | 1.38 | 0 | 0.77 | 0.01 |
| 42 | Tennessee | 123.9 | 147.2 | [0, 841] | 2.08 | 5.02 | 0 | 0.75 | 0.01 |
| 43 | Texas | 1068.6 | 756.48 | [189, 4821] | 1.97 | 4.51 | 0 | 0.78 | 5.17E-07 |
| 44 | Utah | 303.95 | 233.37 | [0, 1282] | 1.47 | 2.6 | 0 | 0.8 | 0.21 |
| 45 | Vermont | 48.3 | 41.75 | [0, 259] | 1.96 | 4.99 | 0 | 0.79 | 0.07 |
| 46 | Virginia | 1428.67 | 1159.25 | [155, 9716] | 2.97 | 13.43 | 0 | 0.71 | 4.51E-06 |
| 47 | Washington | 38.9 | 38.63 | [0, 231] | 1.76 | 3.57 | 0 | 0.75 | 0.18 |
| 48 | West Virginia | 165.21 | 190.98 | [5, 1116] | 2.49 | 7.19 | 0 | 0.75 | 0.10 |
| 49 | Wisconsin | 58.05 | 62.59 | [0, 428] | 1.67 | 3.96 | 0 | 0.81 | 0.00 |
| 50 | Wyoming | 40.34 | 52.5 | [0, 336] | 2.41 | 6.97 | 0 | 0.76 | 1.08E-08 |

STARMA incorporates $N \times N$ spatial weight matrices, which define the hierarchical relationship between different locations (such as countries or states) to account for spatial correlation and heterogeneity.

6. **GSTAR:** The Generalized Space-Time Autoregressive (GSTAR) model (Ruchjana et al., 2012) is a modeling framework designed to capture spatiotemporal dependencies in multivariate time series. As an

Table 6: Global characteristics of Belgium's COVID-19 dataset.

| Node No. | Node Name | Mean | STD | Range | Skewness | Kurtosis | ADF p-value | Hurst Exponent | Teräsvirta p-value |
|---|---|---|---|---|---|---|---|---|---|
| 1 | Antwerpen | 877.6 | 1391.49 | [9, 11873] | 3.67 | 17.68 | 0.01 | 0.86 | 0.19 |
| 2 | BrabantWallon | 191.16 | 334.6 | [0, 3004] | 3.93 | 19.73 | 0.01 | 0 | 0.78 |
| 3 | Brussels | 615.33 | 925.16 | [0, 6839] | 3.88 | 17.54 | 0 | 0.87 | 0.81 |
| 4 | Hainaut | 581.7 | 1040.44 | [0, 8706] | 4.09 | 20.69 | 0 | 0.85 | 0.14 |
| 5 | Limburg | 395.23 | 689.45 | [0, 5711] | 3.78 | 18.05 | 0.03 | 0.86 | 0.04 |
| 6 | Liege | 451.17 | 846.35 | [0, 6996] | 3.68 | 16.24 | 0 | 0.85 | 0.00 |
| 7 | Luxembourg | 118.84 | 235.89 | [0, 1958] | 4.42 | 24.11 | 0 | 0 | 0.44 |
| 8 | Namur | 219.06 | 388.4 | [0, 3198] | 3.97 | 19.92 | 0 | 0 | 0.02 |
| 9 | OostVlaanderen | 744.99 | 1222.12 | [0, 11526] | 3.94 | 20.83 | 0.01 | 0.86 | 0.34 |
| 10 | VlaamsBrabant | 515.82 | 847.77 | [0, 6931] | 3.53 | 16.31 | 0.01 | 0.86 | 0.72 |
| 11 | WestVlaanderen | 564.27 | 993.44 | [0, 8858] | 4.22 | 23.7 | 0 | 0.86 | 0.21 |

Table 7: Global characteristics of Colombia's dengue dataset.

| Node No. | Node Name | Mean | STD | Range | Skewness | Kurtosis | ADF p-value | Hurst Exponent | Teräsvirta p-value |
|---|---|---|---|---|---|---|---|---|---|
| 1 | Amazonas | 6.05 | 11.32 | [0, 126] | 5.86 | 45.88 | 0 | 0 | 0.02 |
| 2 | Antioquia | 119.1 | 193.49 | [1, 1335] | 3.51 | 13.37 | 0 | 0 | 0.12 |
| 3 | Arauca | 16.94 | 20.47 | [1, 186] | 3.49 | 15.98 | 0 | 0.85 | 0.00 |
| 4 | Atlantico | 80.23 | 89.61 | [2, 674] | 2.75 | 10.03 | 0 | 0.85 | 0.02 |
| 5 | Bogota | 4.49 | 2.79 | [0, 13] | -0.22 | -1.45 | 0.49 | 0 | 0.00 |
| 6 | Bolivar | 52.43 | 66.43 | [1, 520] | 3.16 | 14.09 | 0.07 | 0.89 | 0.55 |
| 7 | Boyaca | 11.44 | 13.32 | [1, 103] | 2.55 | 8.18 | 0.01 | 0.9 | 7.31E-13 |
| 8 | Caldas | 9.33 | 11.21 | [1, 93] | 2.79 | 11.63 | 0 | 0 | 7.25E-07 |
| 9 | Caqueta | 14.33 | 17.52 | [1, 121] | 3.04 | 11.56 | 0 | 0.83 | 0.93 |
| 10 | Casanare | 40.85 | 41.79 | [2, 308] | 2.64 | 9.49 | 0 | 0.82 | 0.01 |
| 11 | Cauca | 11.27 | 11.6 | [1, 96] | 3.06 | 13.01 | 0 | 0.86 | 2.01E-05 |
| 12 | Cesar | 51.74 | 47.37 | [1, 297] | 1.64 | 3.14 | 0 | 0.91 | 6.71E-05 |
| 13 | Choco | 5.7 | 6.64 | [0, 77] | 3.4 | 20.7 | 0 | 0 | 0.00 |
| 14 | Cordoba | 37.35 | 38.23 | [1, 166] | 1.46 | 1.22 | 0 | 0.9 | 8.12E-08 |
| 15 | Cundinamarca | 44.83 | 42.43 | [1, 286] | 1.73 | 3.31 | 0 | 0.89 | 1.31E-07 |
| 16 | Guainia | 2.63 | 3.24 | [0, 22] | 2.84 | 8.8 | 0 | 0 | 0.00 |
| 17 | Guajira | 19.32 | 20.02 | [1, 148] | 2.16 | 6.61 | 0.48 | 0 | 0.14 |
| 18 | Guaviare | 6.96 | 7.75 | [1, 53] | 2.22 | 5.52 | 0 | 0 | 7.69E-08 |
| 19 | Huila | 81.48 | 72.77 | [6, 412] | 1.74 | 3.2 | 0 | 0.89 | 0.01 |
| 20 | Magdalena | 26.05 | 26.05 | [1, 176] | 1.95 | 5.13 | 0 | 0.88 | 0.18 |
| 21 | Meta | 95.46 | 91.67 | [10, 639] | 2.69 | 10.17 | 0 | 0.9 | 0.86 |
| 22 | Narino | 6.68 | 5.8 | [0, 41] | 1.42 | 3.06 | 0.03 | 0 | 1.50E-09 |
| 23 | Norte Santander | 84.52 | 56.88 | [5, 300] | 1.08 | 0.92 | 0.01 | 0.9 | 0.06 |
| 24 | Putumayo | 19.34 | 17.94 | [1, 131] | 2.09 | 5.86 | 0 | 0.84 | 0.09 |
| 25 | Quindio | 34.77 | 62.56 | [1, 598] | 5.55 | 38.19 | 0 | 0.91 | 0.33 |
| 26 | Risaralda | 26.42 | 73.56 | [1, 684] | 5.68 | 35.44 | 0 | 0.9 | 0.00 |
| 27 | San andres | 1.95 | 2.24 | [0, 30] | 5.45 | 44.18 | 0 | 0 | 0.00 |
| 28 | Santander | 142.65 | 130.42 | [8, 756] | 1.65 | 3.33 | 0.01 | 0.91 | 0.56 |
| 29 | Sucre | 42.74 | 42.8 | [1, 251] | 2.1 | 4.86 | 0 | 0.88 | 0.00 |
| 30 | Tolima | 112.87 | 100.95 | [14, 518] | 1.56 | 1.87 | 0.01 | 0.9 | 3.25E-05 |
| 31 | Valle | 210.16 | 235.2 | [1, 1460] | 2.17 | 5.64 | 0 | 0.87 | 0.00 |
| 32 | Vaupes | 1.38 | 1.66 | [0, 19] | 6.47 | 52.64 | 0.01 | 0 | 0.00 |
| 33 | Vichada | 3.46 | 4.43 | [0, 43] | 4.14 | 23.26 | 0 | 0 | 0.44 |

extension of the standard STAR model, GSTAR relaxes the assumption of spatial homogeneity by allowing the autoregressive parameters to vary across different locations. This flexibility makes it particularly effective for modeling phenomena where spatial units exhibit heterogeneous characteristics. The model

Table 8: Global characteristics of Hungary's chickenpox dataset.

| Node No. | Node Name | Mean | STD | Range | Skewness | Kurtosis | ADF p-value | Hurst Exponent | Teräsvirta p-value |
|---|---|---|---|---|---|---|---|---|---|
| 1 | Bács | 37.17 | 36.84 | [0, 274] | 1.79 | 5.35 | 0 | 0.76 | 4.28E-13 |
| 2 | Baranya | 34.2 | 32.57 | [0, 194] | 1.37 | 2.25 | 0 | 0.77 | 6.86E-13 |
| 3 | Békés | 28.91 | 37.62 | [0, 271] | 2.48 | 8.12 | 0 | 0.83 | 0.95 |
| 4 | Borsod | 57.08 | 50.73 | [0, 355] | 1.39 | 3.08 | 0 | 0.74 | 4.81E-12 |
| 5 | Budapest | 101.25 | 76.35 | [0, 479] | 0.95 | 1.4 | 0 | 0.74 | 2.07E-14 |
| 6 | Csongrád | 31.49 | 33.79 | [0, 199] | 1.66 | 3.11 | 0 | 0.75 | 0.19 |
| 7 | Fejér | 33.27 | 31.4 | [0, 164] | 1.11 | 0.71 | 0 | 0.82 | 6.36E-11 |
| 8 | Győr | 41.44 | 36.01 | [0, 181] | 1.02 | 0.74 | 0 | 0.78 | 1.04E-08 |
| 9 | Hajdú | 47.1 | 44.61 | [0, 262] | 1.4 | 2.31 | 0 | 0.77 | 3.12E-08 |
| 10 | Heves | 29.69 | 31.86 | [0, 210] | 1.94 | 4.88 | 0 | 0.81 | 0.03 |
| 11 | Jasz | 40.87 | 37.28 | [0, 224] | 1.12 | 1.13 | 0 | 0.8 | 5.26E-07 |
| 12 | Komárom | 25.64 | 24.47 | [0, 160] | 1.4 | 2.75 | 0 | 0.8 | 1.78E-08 |
| 13 | Nógrád | 21.85 | 22.03 | [0, 112] | 1.4 | 2 | 0 | 0.83 | 2.87E-07 |
| 14 | Pest | 86.1 | 66.77 | [0, 431] | 0.87 | 1.13 | 0 | 0.76 | 3.33E-16 |
| 15 | Somogy | 27.61 | 26.72 | [0, 155] | 1.41 | 2.23 | 0 | 0.74 | 6.19E-06 |
| 16 | Szabolcs | 29.85 | 31.81 | [0, 203] | 1.89 | 5.26 | 0 | 0.74 | 3.84E-07 |
| 17 | Tolna | 20.35 | 23.27 | [0, 131] | 1.71 | 2.96 | 0 | 0.82 | 0.01 |
| 18 | Vas | 22.47 | 25.01 | [0, 141] | 1.67 | 3.12 | 0 | 0.82 | 0.00 |
| 19 | Veszprém | 40.64 | 40.7 | [0, 230] | 1.57 | 3.03 | 0 | 0.81 | 0.01 |
| 20 | Zala | 19.87 | 22 | [0, 216] | 2.37 | 12.34 | 0 | 0.79 | 2.41E-13 |

utilizes location-specific weights (represented through weights matrices $W$) to account for the proximity and influence of neighboring sites, combining temporal lags and spatial lags into a unified linear system.

7. **STGCN:** The Spatio-Temporal Graph Convolutional Network (STGCN), proposed by Yu et al. (2018), is a novel deep learning framework designed to address the challenges of time series prediction in the traffic domain by modeling multi-scale traffic networks. Unlike traditional approaches that rely on regular convolutional or recurrent units, STGCN is built with a complete convolutional structure on graphs. The architecture consists of several spatio-temporal convolutional blocks, each featuring a sandwich structure where two gated temporal convolution layers flank a spatial graph convolution layer.

8. **DeepAR:** DeepAR (Salinas et al., 2020) is a probabilistic forecasting methodology that utilizes RNNs to learn a global model from a large collection of related time series. Unlike classical methods that fit parameters to individual series, DeepAR shares information across the dataset, allowing it to capture complex seasonal patterns and provide forecasts for items with little historical data. The architecture employs an encoder-decoder framework where the network's output parametrizes a chosen likelihood function, such as Gaussian for real-valued data or negative binomial for count data.

9. **Probabilistic iTransformer:** The Inverted Transformer (iTransformer) (Liu et al., 2023; Alexandrov et al., 2020) architecture applies Transformers to multivariate time series forecasting by inverting the standard data modality. Unlike vanilla Transformers that embed multiple variates of a single timestamp into temporal tokens, iTransformer embeds the entire historical sequence of each individual variate into its own variate token. This inverted approach allows the self-attention mechanism to explicitly capture multivariate correlations between variate tokens, while the shared feed-forward network (FFN) learns nonlinear, global temporal representations for each series independently.

10. **GpGp:** The Fast Gaussian Process (GpGp) method (Guinness, 2018), represents a sophisticated refinement of Vecchia's approximation (Vecchia, 1988), specifically optimized for the analysis of large-scale spatiotemporal datasets. This approach streamlines the computational complexity of traditional Gaussian Process regression by introducing a grouping mechanism for ordered sequences, which allows for highly efficient likelihood evaluation and parameter estimation. A defining advantage of the GpGp framework

is its interpolative nature; as a continuous spatial predictor, it can generalize beyond the observed data points to forecast values for specific coordinates (longitude and latitude) that were not originally present in the training set.

11. **DiffSTG:** DiffSTG (Wen et al., 2023) is a non-autoregressive probabilistic forecasting framework that generalizes denoising diffusion probabilistic models (DDPMs) to spatio-temporal graphs. It utilizes a dedicated denoising network, UGnet, which integrates a U-Net-based structure for multi-scale temporal modeling with GNNs to capture complex spatial correlations. By implementing a conditional reverse diffusion process, the model learns to reconstruct future graph signals from Gaussian noise, conditioned on historical observations and the underlying graph structure.

12. **Spatiotemporal Echo-State Networks (STESNs):** STESN (Huang et al., 2022) captures nonlinear spatiotemporal dynamics by mapping inputs to high-dimensional reservoir states. These states evolve via the fixed nonlinear dynamics, where the weight matrices are sparse, random, and fixed. The output is formulated with quadratic interactions to model complex physical processes. Unlike standard RNNs, only the output matrices are trained using a closed-form ridge regression estimator. To ensure computational tractability over large domains, the model is applied to a reduced set of $n^*$ spatial knots and then reconstructed via kriging. Uncertainty quantification is done via empirical calibration by estimating specific quantiles.

### E.1 Implementation Details

We implemented the NHiTS, Transformers, TCN, and DeepAR models using the Darts library (Herzen et al., 2022) in Python. The STARMA, GSTAR, and GpGp were built using the `starma`, `gstar`, and `GpGp` packages respectively, in the R statistical software. For STGCN, we utilized the PyTorch implementation in PyTorch Geometric Temporal[6] (Rozemberczki et al., 2021a). The probabilistic iTransformer was implemented using GluonTS (Alexandrov et al., 2020) with a PyTorch backend, and the default implementations were used for DiffSTG[7] and STESN[8]. The multivariate LSTM and the proposed MVEN, GCEN, and STEN models were implemented using the PyTorch framework in Python, offering seamless training and inference on GPUs.

The proposed and baseline deep learning models were trained on an NVIDIA P100 GPU with 16 GB RAM. For GSTAR, STARMA, and GpGp, we utilized an Intel(R) Xeon(R) CPU with 4 cores and 30 GB RAM. The default STESN implementation is based on NumPy and SciPy, and does not natively support training and inference on GPUs, so we used an Intel(R) Xeon(R) CPU with 20 cores and 126 GB RAM. The time taken by the models for both training and inference (in the evaluation setup of 50 independent runs) is presented in Table 9. Time consumption mainly depends on the training device (whether GPU or CPU), and hyperparameter configuration of the models, such as the number of hidden layers and size of the embedding dimension. The STESN model was the most time-consuming as compared to others, in our evaluation setup. The predictive uncertainty profile was assessed directly from the raw STESN forecast ensembles, bypassing the empirical calibration procedure suggested in the original framework. Furthermore, as the target forecast domain was restricted to the spatial points within the established training grid, all nodes were used in fitting the ESN. The spatial kriging step, typically utilized for interpolation at unobserved locations, was omitted. For DiffSTG, we utilized the same adjacency matrices that were used in training GCEN.

Hyperparameters for the proposed models were optimized over 100 trials using the Optuna framework (Akiba et al., 2019) by minimizing the CRPS on the corresponding holdout validation sets. The validation set length for hyperparameter tuning was adjusted according to the forecast term and temporal granularity of the dataset. Specifically, the validation lengths for short, medium, and long-term forecast horizons were set to 60, 90, and 120 days for the daily data; 9, 13, and 26 weeks for the weekly data; and 12, 24, and 36 months for the monthly data, respectively. The hyperparameter search space encompassed the output dimensionality of the spatial module, bounded between 1 and 10; the hidden state dimensionality of the LSTM module, constrained between 2 and 100; the depth of the LSTM network, ranging from 1 to 5 layers; the LSTM dropout rate, sampled continuously from $[0.0, 0.5]$; the learning rate, sampled from $[10^{-5}, 0.1]$; and the decay parameter

---

[6]https://github.com/benedekrozemberczki/pytorch_geometric_temporal

[7]https://github.com/wenhaomin/DiffSTG

[8]https://github.com/hhuang90/KSA-wind-forecast

Table 9: Total time consumed (in seconds) to train the models and evaluate them across 50 independent evaluation runs. The deterministic models (LSTM-STGCN) are evaluated in a single run only.

| Dataset | Horizon | LSTM | NHiTS | Transformers | TCN | GSTAR | STARMA | STGCN | DeepAR | Prob-iTransformer | GpGp | DiffSTG | STESN | MVEN | GCEN | STEN |
|---|---|---|---|---|---|---|---|---|---|---|---|---|---|---|---|---|
| Japan TB | 6-month | 4.59 | 6.64 | 11.4 | 3.39 | 2.46 | 0.16 | 300 | 6.53 | 111.82 | 1019.35 | 5266.52 | 9026.05 | 15.77 | 15.4 | 6.62 |
| | 12-month | 1.68 | 6.08 | 8.93 | 3.01 | 2.44 | 0.15 | 673 | 7.18 | 116.97 | 1015.88 | 4776.41 | 9224.47 | 26.69 | 12.4 | 11.27 |
| | 24-month | 1.99 | 5.19 | 3.25 | 1.17 | 2.46 | 0.15 | 836 | 8.6 | 123.44 | 783.37 | 4814.27 | 461.9 | 7.96 | 18.3 | 38.88 |
| China TB | 6-month | 0.61 | 2.41 | 3.72 | 1.14 | 0.35 | 0.069 | 56 | 4.77 | 110.7 | 150.61 | 4740.22 | 26.14 | 2.77 | 5.54 | 6.18 |
| | 12-month | 0.42 | 1.42 | 3.16 | 1.2 | 0.35 | 0.07 | 50 | 5.25 | 114.01 | 148.7 | 4819.17 | 27.56 | 3.36 | 4.75 | 5.48 |
| USA ILI | 4-week | 8.17 | 9.54 | 14.3 | 4.68 | 16.61 | 0.21 | 1293 | 8.31 | 117.83 | 1972.37 | 5569.36 | 9689.09 | 20.94 | 37.07 | 51.22 |
| | 9-week | 5.97 | 8.91 | 14.4 | 4.66 | 15.31 | 0.21 | 1244 | 8.8 | 120.27 | 2030.99 | 5615.99 | 9694.36 | 13.31 | 26.9 | 11.57 |
| | 13-week | 22.7 | 8.8 | 14.5 | 4.63 | 15.37 | 0.21 | 1205 | 9.37 | 122.87 | 2105.11 | 5770.92 | 9744.84 | 13.46 | 12.25 | 21.81 |
| Belgium COVID-19 | 30-day | 7.69 | 19.2 | 27.8 | 11.4 | 2.78 | 0.04 | 7155 | 16.33 | 113.57 | 713.31 | 4930.86 | 10778.94 | 32 | 81 | 10.2 |
| | 60-day | 5.01 | 16.2 | 25.6 | 9.4 | 5.4 | 0.04 | 9468 | 17.43 | 117.28 | 692.75 | 4919.1 | 10869.83 | 15.1 | 150 | 18 |
| | 90-day | 41.8 | 14.2 | 24.5 | 8.4 | 8.7 | 0.044 | 10838 | 19.4 | 120.44 | 695.07 | 5037.91 | 10744.37 | 8.12 | 29.2 | 32.3 |
| Colombia Dengue | 4-week | 25 | 23.8 | 37.4 | 11.6 | 12.94 | 0.24 | 3473 | 15.24 | 127.55 | 3010.38 | 4975.49 | 11304.18 | 16.71 | 24.44 | 93.96 |
| | 9-week | 7.65 | 23.4 | 37.4 | 11.5 | 12.37 | 0.244 | 3354 | 15.5 | 124.95 | 3079.72 | 4897.05 | 11391.19 | 113.91 | 41.99 | 31.61 |
| | 13-week | 23 | 23.5 | 33.4 | 11.2 | 12.36 | 0.24 | 3320 | 15.82 | 126.8 | 2986.14 | 4953.34 | 11317.98 | 60.02 | 14.22 | 106.49 |
| Hungary Chickenpox | 4-week | 8.79 | 15.1 | 21 | 7.27 | 2.08 | 0.14 | 2143 | 12.63 | 125.09 | 1021.59 | 4648.87 | 10090.61 | 14.07 | 32.92 | 9.87 |
| | 9-week | 4.05 | 13.6 | 22.4 | 7.1 | 2.22 | 0.126 | 2048 | 10.86 | 108.2 | 1001.36 | 4614.07 | 10035.31 | 75.40 | 23.42 | 51.04 |
| | 13-week | 9.66 | 13.6 | 22.4 | 7.11 | 2.22 | 0.122 | 2049 | 11.22 | 110.74 | 985.69 | 4769.56 | 10022.03 | 10.60 | 13.25 | 13.12 |

$\alpha$ and maximum spatial lags $L$ for STEN (see Sec. 4.2) chosen from $\{1, 2, 3\}$. Following optimization, the best-performing hyperparameters were used to train the final models on the combined training and validation data prior to evaluation. The networks were trained via backpropagation using the energy score loss and the Adam optimizer (Kingma & Ba, 2014). With the exception of STGCN, all architectures were trained for 100 epochs. The STGCN was trained for 50 epochs, due to its substantial time consumption.

To ensure a rigorous and transparent empirical evaluation, we adopted a standardized hyperparameter configuration protocol across all baseline models, thereby eliminating potential optimization bias (i.e., inadvertently under-tuning baselines to favor our proposed models). For the temporal models implemented via Darts framework (NHiTS, Transformers, TCN, and DeepAR), we utilized the native, library-specified default configurations[9]. For Prob-iTransformer and the spatiotemporal architectures (STGCN (Yu et al., 2018), DiffSTG (Wen et al., 2023), and STESN (Huang et al., 2022)), we strictly adhered to the hyperparameter settings established in their respective foundational papers, ensuring that each baseline was evaluated under its intended, expert-optimized conditions. Conversely, because a standard, universally accepted default configuration does not exist for the vanilla LSTM baseline, its hyperparameters were independently optimized using the validation sets, with the same search space as MVEN. Finally, to maintain strict experimental control and ensure a fair comparison, the input and output sequence windows were kept uniform across all evaluated architectures, including our proposed models.

## F Evaluation Metrics

This section provides a comprehensive overview of the performance measures employed to evaluate the proposed models. Our evaluation framework is designed to be nearly exhaustive, as it accounts for both point forecast accuracy and probabilistic calibration. Let $\left\{\widehat{Y}_t\right\}_{t=T+1}^{T+q}$ denote the predicted values for $\{Y_t\}_{t=T+1}^{T+q}$, $q$ be the forecast horizon, and $t = \{1, \ldots, T\}$ be the time indices for the observed (historical) data $\{Y_t\}_{t=1}^{T}$.

1. **SMAPE:** The Symmetric Mean Absolute Percentage Error (SMAPE) adjusts for the asymmetry between over-forecasts and under-forecasts and is bounded between 0% and 200%.

$$SMAPE = \frac{100\%}{q} \sum_{t=T+1}^{T+q} \frac{|\widehat{Y}_t - Y_t|}{(|Y_t| + |\widehat{Y}_t|)/2}.$$

2. **MAE:** Mean Absolute Error (MAE) is the average of the absolute differences between the forecasted values ($\widehat{Y}_t$) and the actual observations ($Y_t$).

$$MAE = \frac{1}{q} \sum_{t=T+1}^{T+q} \left|\widehat{Y}_t - Y_t\right|.$$

---

[9]https://github.com/unit8co/darts/tree/master/darts/models/forecasting

3. **RMSE:** Root Mean Squared Error (RMSE) is the square root of the average of squared differences between prediction and observation.

$$RMSE = \sqrt{\frac{1}{q} \sum_{t=T+1}^{T+q} \left(\widehat{Y}_t - Y_t\right)^2}.$$

4. **MASE:** Mean Absolute Scaled Error (MASE) scales the MAE by the mean absolute error of a naive forecast produced on the training data.

$$MASE = \frac{\frac{1}{q} \sum_{t=T+1}^{T+q} \left|\widehat{Y}_t - Y_t\right|}{\frac{1}{T-1} \sum_{t=2}^{T} |Y_t - Y_{t-1}|}.$$

5. **RMSSE:** Similar to MASE, the Root Mean Squared Scaled Error (RMSSE) scales the MSE of the forecast by the MSE of a naive baseline, subsequently taking the square root.

$$RMSSE = \sqrt{\frac{\frac{1}{q} \sum_{t=T+1}^{T+q} \left(\widehat{Y}_t - Y_t\right)^2}{\frac{1}{T-1} \sum_{t=2}^{T} (Y_t - Y_{t-1})^2}}.$$

6. **Pinball Loss:** The Pinball loss (Gneiting et al., 2023) is used to evaluate the accuracy of a quantile forecast. For a target quantile $\tau \in [0, 1]$, the loss is defined as:

$$L_\tau(\mathbf{Y}, \widehat{\mathbf{Y}}) = \frac{1}{q} \sum_{t=T+1}^{T+q} \max\left\{\tau\left(Y_t - \widehat{Y}_t\right), \ (1-\tau)\left(Y_t - \widehat{Y}_t\right)\right\}.$$

7. **$\rho$-Risk:** $\rho$-risk is a normalized version of the Pinball loss, often calculated across a range of quantiles to provide a summary of the distribution's coverage (Seeger et al., 2016; Salinas et al., 2020).

$$\rho\text{-risk} = \frac{2 \sum_{t=T+1}^{T+q} \max\left\{\tau\left(Y_t - \widehat{Y}_t\right), \ (1-\tau)\left(Y_t - \widehat{Y}_t\right)\right\}}{\sum_{t=T+1}^{T+q} Y_t}$$

8. **CRPS:** The Continuous Ranked Probability Score (CRPS) (Gneiting & Raftery, 2007) generalizes the MAE to the case of probabilistic forecasts. It measures the difference between the predicted cumulative distribution function and the empirical CDF of the observation.

$$CRPS(F, y) = \frac{1}{q} \sum_{t=T+1}^{T+q} \int_{\mathbb{R}} \left(F_t(y) - \mathbb{I}_{y \geq Y_t}\right)^2 \ dy$$

where, $F_t$ denotes the predicted probability distribution at time $t$.

9. **Winkler Score:** The Winkler score (Winkler et al., 1996) is a strictly proper scoring rule used to assess the quality of predictive confidence intervals by jointly evaluating their width and coverage. For a nominal confidence level $1 - \alpha$, given a prediction interval $[\ell_{\alpha,t}, u_{\alpha,t}]$ at time $t$ and the observed value $y_t$, the Winkler score $W_{\alpha,t}$ is mathematically defined as:

$$W_{\alpha,t} = \begin{cases} (u_{\alpha,t} - \ell_{\alpha,t}) + \frac{2}{\alpha}(\ell_{\alpha,t} - y_t), & \text{if } y_t < \ell_{\alpha,t} \\ (u_{\alpha,t} - \ell_{\alpha,t}), & \text{if } \ell_{\alpha,t} \leq y_t \leq u_{\alpha,t} \\ (u_{\alpha,t} - \ell_{\alpha,t}) + \frac{2}{\alpha}(y_t - u_{\alpha,t}), & \text{if } y_t > u_{\alpha,t} \end{cases}$$

This formulation assigns the Winkler score as the length of the interval when the true value lies inside it, rewarding narrow intervals. However, when the true value falls outside the interval, the score imposes an additional penalty proportional to the distance the observation lies beyond the interval bounds, scaled by $2/\alpha$. This penalization emphasizes the importance of coverage, discouraging intervals that fail to contain the true values. Consequently, lower Winkler scores indicate better prediction intervals that are simultaneously sharp and well-calibrated.

10. **Probability Integral Transform (PIT):** For every spatiotemporal observation $y_{true}$ in the test set, we compute a PIT value relative to the model's predicted forecast ensemble. Given an ensemble of $M$ generated trajectories, the PIT value is calculated using a rank-based formulation defined as:

$$PIT = \frac{1}{M+1} \sum_{i=1}^{M} \mathbb{I}(\widehat{y}^{(i)} < y_{true})$$

where $\mathbb{I}(\cdot)$ is the indicator function and $\widehat{y}^{(i)}$ represents the $i$-th ensemble member. This formulation interprets the PIT value as the normalized rank of the ground truth relative to the ensemble distribution.

11. **Empirical Coverage:** Empirical coverage is the proportion of times the true values fall within the predicted confidence intervals:

$$\text{Coverage} = \frac{1}{n} \sum_{i=1}^{n} \mathbb{I}(y_i \in CI_i)$$

where, $\mathbb{I}$ denotes the indicator function, $y_i$ is the true value, and $CI_i$ is the confidence interval for prediction $i$. For a 95% confidence interval, the best coverage score is 0.95. This metric is often misleading, because a high coverage score, potentially reaching 1.00, can be artificially attained by generating excessively wide intervals (see Table 10). Fig. 17 illustrates this, where GpGp produces extremely wide intervals and attains 100% coverage on the Belgium dataset. Such results lack sharpness, rendering them impractical for precision-dependent applications such as policy formulation or strategic decision-making. Hence, the intervals are evaluated using the Winkler score in the main text, which simultaneously penalizes both lack of coverage and excessive interval width.

## G   Tables and Figures

Table 10: Empirical coverage achieved on the test set by the 95% PIs generated by the probabilistic models. The mean (standard deviation) is reported across 50 runs of forecast ensemble generation and evaluation, as in the results in Tables 11-13.

| Dataset | Horizon | DeepAR | Prob-iTransformer | GpGp | DiffSTG | STESN | MVEN | GCEN | STEN |
|---|---|---|---|---|---|---|---|---|---|
| Japan TB | 6-month | 0.94 (0.01) | 0.91 (0.01) | 0.98 (0.00) | 0.98 (0.01) | 0.25 (0.01) | 0.41 (0.01) | 0.63 (0.01) | 0.59 (0.01) |
| | 12-month | 0.92 (0.01) | 0.89 (0.01) | 0.99 (0.00) | 1.00 (0.00) | 0.30 (0.01) | 0.51 (0.01) | 0.52 (0.01) | 0.57 (0.01) |
| | 24-month | 0.89 (0.01) | 0.76 (0.01) | 0.99 (0.00) | 1.00 (0.00) | 0.19 (0.00) | 0.36 (0.01) | 0.47 (0.00) | 0.40 (0.01) |
| China TB | 6-month | 0.72 (0.01) | 0.39 (0.02) | 0.96 (0.01) | 0.99 (0.00) | 0.09 (0.05) | 0.46 (0.01) | 0.44 (0.01) | 0.34 (0.01) |
| | 12-month | 0.78 (0.01) | 0.34 (0.01) | 0.96 (0.00) | 0.99 (0.00) | 0.20 (0.02) | 0.45 (0.01) | 0.35 (0.01) | 0.36 (0.01) |
| USA ILI | 4-week | 0.85 (0.01) | 0.74 (0.02) | 0.78 (0.01) | 0.81 (0.01) | 0.23 (0.01) | 0.84 (0.01) | 0.69 (0.01) | 0.77 (0.01) |
| | 9-week | 0.67 (0.03) | 0.86 (0.01) | 0.88 (0.00) | 0.92 (0.00) | 0.24 (0.01) | 0.82 (0.01) | 0.75 (0.01) | 0.66 (0.01) |
| | 13-week | 0.48 (0.01) | 0.85 (0.00) | 0.92 (0.00) | 0.92 (0.00) | 0.24 (0.01) | 0.80 (0.01) | 0.65 (0.01) | 0.61 (0.01) |
| Belgium COVID-19 | 30-day | 0.93 (0.01) | 0.98 (0.01) | 1.00 (0.00) | 0.75 (0.03) | 0.10 (0.01) | 0.79 (0.01) | 0.86 (0.01) | 0.53 (0.01) |
| | 60-day | 0.80 (0.02) | 0.99 (0.00) | 1.00 (0.00) | 0.99 (0.01) | 0.10 (0.01) | 0.60 (0.01) | 0.63 (0.01) | 0.22 (0.01) |
| | 90-day | 0.97 (0.00) | 0.98 (0.00) | 1.00 (0.00) | 1.00 (0.00) | 0.08 (0.00) | 0.78 (0.01) | 0.50 (0.01) | 0.85 (0.01) |
| Colombia Dengue | 4-week | 0.75 (0.02) | 0.93 (0.01) | 0.94 (0.01) | 0.75 (0.02) | 0.38 (0.01) | 0.85 (0.01) | 0.77 (0.01) | 0.82 (0.01) |
| | 9-week | 0.83 (0.01) | 0.95 (0.01) | 0.95 (0.00) | 0.83 (0.02) | 0.35 (0.01) | 0.90 (0.01) | 0.70 (0.02) | 0.80 (0.01) |
| | 13-week | 0.80 (0.01) | 0.97 (0.00) | 0.95 (0.00) | 0.83 (0.02) | 0.37 (0.01) | 0.85 (0.01) | 0.74 (0.01) | 0.75 (0.01) |
| Hungary Chickenpox | 4-week | 0.87 (0.02) | 0.84 (0.02) | 0.94 (0.01) | 0.88 (0.02) | 0.13 (0.01) | 0.69 (0.03) | 0.44 (0.02) | 0.32 (0.01) |
| | 9-week | 0.95 (0.01) | 0.95 (0.01) | 0.96 (0.01) | 0.94 (0.01) | 0.16 (0.01) | 0.74 (0.02) | 0.53 (0.01) | 0.46 (0.02) |
| | 13-week | 0.93 (0.01) | 0.89 (0.01) | 0.97 (0.00) | 0.98 (0.01) | 0.16 (0.01) | 0.76 (0.01) | 0.51 (0.01) | 0.57 (0.01) |

Table 11: Mean forecast performance (along with standard deviations) of the proposed and benchmark models on all six datasets for the short-term forecast horizons (30-day/4-week/6-month). The **best** and **second best** results are highlighted.

| Dataset | Metric | Temporal Models | | | | Spatiotemporal Models | | | Probabilistic Models (Temporal & Spatiotemporal) | | | | | ST-Engression Models (Proposed) | | |
|---|---|---|---|---|---|---|---|---|---|---|---|---|---|---|---|---|
| | | LSTM | NHiTS | Transformers | TCN | GSTAR | STARMA | STGCN | DeepAR | Prob-iTransformer | GpGp | DiffSTG | STESN | MVEN | GCEN | STEN |
| Japan TB | SMAPE | **22.04** | 22.58 | 22.79 | 30.26 | 23.24 | 27.7 | 41.16 | 25.57 (0.42) | 35.32 (0.26) | 98.26 (3.55) | 64.37 (0.99) | 23.42 (0.09) | 23.99 (0.13) | **21.30 (0.14)** | 23.10 (0.14) |
| | MAE | 6.09 | 6.09 | 5.8 | 9.08 | 5.67 | 6.46 | 14.99 | 7.48 (0.20) | 10.86 (0.15) | 42.81 (3.43) | 19.95 (0.38) | **5.61 (0.03)** | 6.82 (0.05) | 5.29 (0.05) | 6.46 (0.07) |
| | RMSE | 11.18 | **7.04** | 6.78 | 10.19 | 8.99 | 11.11 | 26.03 | 13.24 (0.37) | 20.67 (0.47) | 52.12 (3.02) | 26.38 (0.35) | 9.18 (0.04) | 12.39 (0.12) | 9.39 (0.12) | 11.78 (0.16) |
| | MASE | 0.71 | **0.63** | **0.63** | 0.93 | 0.66 | 0.75 | 1.74 | 0.87 (0.02) | 1.26 (0.02) | 4.98 (0.40) | 2.32 (0.04) | 0.65 (0.00) | 0.79 (0.01) | **0.62 (0.01)** | 0.75 (0.01) |
| | RMSSE | 0.84 | 0.58 | 0.58 | 0.84 | **0.67** | 0.83 | 1.95 | 0.99 (0.03) | 1.55 (0.03) | 3.90 (0.23) | 1.98 (0.03) | 0.69 (0.00) | 0.93 (0.01) | 0.70 (0.01) | 0.88 (0.01) |
| | Pinball-80 | 1.82 | 2.15 | 2.3 | 2.58 | 2.88 | 2.7 | 3.09 | 2.82 (0.05) | 3.75 (0.12) | 20.44 (0.86) | 9.31 (0.13) | 2.14 (0.01) | 1.99 (0.02) | **1.90 (0.02)** | 2.05 (0.02) |
| | Pinball-95 | 1.21 | 1.71 | 2 | 1.6 | 2.9 | 2.43 | 0.89 | 1.04 (0.01) | 1.28 (0.05) | 8.18 (0.25) | 3.82 (0.11) | 1.56 (0.01) | **0.80 (0.01)** | 0.91 (0.02) | 0.85 (0.01) |
| | ρ-0.5 Risk | 0.19 | 0.19 | 0.18 | 0.28 | 0.17 | 0.2 | 0.46 | 0.23 (0.01) | 0.33 (0.00) | 1.32 (0.11) | 0.61 (0.01) | **0.17 (0.00)** | 0.21 (0.00) | **0.16 (0.00)** | 0.20 (0.00) |
| | ρ-0.9 Risk | **0.09** | 0.11 | 0.13 | 0.12 | 0.18 | 0.15 | 0.1 | 0.11 (0.00) | 0.13 (0.01) | 0.81 (0.03) | 0.38 (0.01) | **0.09 (0.00)** | **0.08 (0.00)** | 0.08 (0.00) | 0.08 (0.00) |
| | CRPS | 6.09 | 6.09 | 5.8 | 9.08 | 5.67 | 6.46 | 14.99 | 5.31 (0.11) | 7.80 (0.08) | 30.81 (1.48) | 14.25 (0.21) | **5.12 (0.02)** | 5.97 (0.03) | 4.23 (0.03) | 5.14 (0.04) |
| | Winkler Score | - | - | - | - | - | - | - | **48.41 (1.98)** | 70.02 (1.62) | 308.61 (7.42) | 132.89 (3.61) | 164.18 (1.34) | 172.63 (0.89) | 79.05 (1.79) | 95.13 (2.34) |
| China TB | SMAPE | 14.63 | **14.55** | 19.06 | 27.81 | 21.6 | 20.86 | 21.09 | 19.73 (0.21) | 123.04 (0.52) | 63.65 (1.12) | 65.03 (0.87) | 18.93 (0.04) | 15.88 (0.10) | **12.59 (0.08)** | 14.73 (0.12) |
| | MAE | 360.46 | 436.29 | 475.54 | 578.56 | 490.91 | **292.29** | 606.41 | 494.89 (6.21) | 1756.88 (2.76) | 1217.47 (21.71) | 1291.81 (22.18) | 468.06 (0.89) | 391.07 (3.10) | 295.58 (2.07) | 342.05 (2.57) |
| | RMSE | 816.36 | 531.64 | **516.28** | 618.39 | 728.55 | 427.14 | 1401.16 | 871.08 (7.57) | 2365.15 (2.76) | 1677.44 (24.17) | 1682.44 (25.83) | 803.85 (0.74) | 807.22 (5.84) | 677.79 (2.65) | 794.22 (2.19) |
| | MASE | 1.44 | 1.2 | 1.66 | 2.18 | 1.96 | 1.17 | 2.42 | 1.98 (0.02) | 7.03 (0.01) | 4.87 (0.09) | 5.17 (0.09) | 1.87 (0.00) | 1.56 (0.01) | **1.18 (0.01)** | 1.37 (0.01) |
| | RMSSE | 0.95 | 0.9 | 1.28 | 1.66 | 0.85 | **0.5** | 1.63 | 1.02 (0.01) | 2.76 (0.00) | 1.96 (0.03) | 1.96 (0.03) | 0.94 (0.00) | 0.94 (0.01) | 0.79 (0.00) | 0.93 (0.00) |
| | Pinball-80 | 164.61 | 163.04 | 168.98 | 293.54 | 150.69 | **95.97** | 194.49 | 212.44 (1.78) | 1214.66 (2.85) | 530.33 (6.98) | 631.04 (8.18) | 163.84 (0.22) | 172.31 (1.02) | 144.95 (0.44) | 167.67 (0.77) |
| | Pinball-95 | 156.8 | 135.48 | 134.59 | 295.67 | **103.31** | 70.89 | 140.14 | 124.89 (2.13) | 1128.40 (11.84) | 243.03 (6.07) | 262.48 (5.42) | 120.92 (1.48) | 125.57 (1.19) | 125.05 (0.83) | 143.11 (0.96) |
| | ρ-0.5 Risk | 0.18 | 0.21 | 0.23 | 0.28 | 0.24 | **0.14** | 0.29 | 0.24 (0.00) | 0.85 (0.00) | 0.59 (0.01) | 0.63 (0.01) | 0.18 (0.10) | 0.19 (0.00) | 0.14 (0.00) | 0.17 (0.00) |
| | ρ-0.9 Risk | 0.15 | 0.14 | 0.14 | 0.29 | 0.12 | 0.08 | 0.15 | 0.16 (0.00) | 1.19 (0.01) | 0.36 (0.01) | 0.42 (0.01) | **0.10 (0.06)** | 0.14 (0.00) | 0.13 (0.00) | 0.15 (0.00) |
| | CRPS | 360.46 | 436.29 | 475.54 | 578.56 | 490.91 | 292.29 | 606.41 | 402.43 (2.45) | 1560.92 (3.68) | 925.29 (9.76) | 1023.12 (8.67) | 448.60 (2.30) | 335.04 (2.21) | **259.17 (1.28)** | 303.31 (1.22) |
| | Winkler Score | - | - | - | - | - | - | - | **6767.63 (174.41)** | 41308.55 (724.21) | 10336.87 (364.22) | 11141.40 (189.03) | 15678.32 (459.52) | 8369.24 (123.60) | 7600.06 (67.49) | 8890.56 (55.77) |
| USA ILI | SMAPE | 31.33 | 39.86 | 53.91 | 57.62 | 41.28 | 36.06 | 50.65 | 45.90 (0.57) | 81.83 (0.48) | 103.79 (1.23) | 85.17 (1.25) | **28.71 (0.13)** | 37.17 (0.45) | 31.15 (0.24) | 32.49 (0.29) |
| | MAE | 89.1 | 118.65 | 126.97 | 177.96 | 128.32 | 110.43 | 151.24 | 120.80 (2.15) | 292.77 (0.76) | 410.74 (3.19) | 268.10 (1.99) | 69.28 (0.32) | 119.33 (2.10) | **78.06 (1.15)** | 90.69 (1.34) |
| | RMSE | 187.27 | **128.46** | 141.12 | 185.68 | 238.04 | 225.65 | 303.97 | 267.24 (4.06) | 611.56 (1.00) | 657.67 (3.57) | 517.34 (3.50) | 125.66 (0.89) | 245.61 (6.20) | 147.83 (2.86) | 183.59 (4.34) |
| | MASE | 2.38 | 2.87 | 3.29 | 4.12 | 3.43 | 2.95 | 4.04 | 3.23 (0.06) | 7.83 (0.02) | 10.98 (0.09) | 7.17 (0.05) | **1.85 (0.01)** | 3.19 (0.06) | 2.09 (0.03) | 2.43 (0.04) |
| | RMSSE | 1.82 | 1.9 | 2.15 | 2.6 | 2.31 | 2.19 | 2.95 | 2.59 (0.04) | 5.94 (0.01) | 6.39 (0.03) | 5.02 (0.03) | **1.22 (0.01)** | 2.38 (0.06) | 1.44 (0.03) | 1.78 (0.04) |
| | Pinball-80 | 58.75 | 91.34 | 75.89 | 135.8 | 101.02 | 83.95 | 104.77 | 60.36 (0.76) | 187.13 (0.83) | 200.18 (1.27) | 155.42 (1.74) | 42.52 (0.38) | 52.45 (1.91) | **38.26 (0.71)** | 44.76 (1.37) |
| | Pinball-95 | 65.85 | 107.35 | 82.09 | 159.21 | 119.45 | 98.32 | 119.34 | 42.22 (1.12) | 163.26 (2.40) | 137.28 (1.78) | 122.78 (2.66) | 42.25 (0.57) | 30.40 (1.67) | 27.68 (0.42) | 29.15 (1.35) |
| | ρ-0.5 Risk | 0.23 | 0.31 | 0.33 | 0.46 | 0.33 | 0.29 | 0.39 | 0.31 (0.01) | 0.76 (0.00) | 1.07 (0.01) | 0.70 (0.01) | **0.18 (0.00)** | 0.31 (0.01) | 0.20 (0.00) | 0.24 (0.00) |
| | ρ-0.9 Risk | 0.33 | 0.53 | 0.42 | 0.79 | 0.59 | 0.49 | 0.6 | 0.27 (0.00) | 0.94 (0.01) | 0.86 (0.01) | 0.74 (0.01) | 0.22 (0.00) | 0.21 (0.01) | **0.17 (0.00)** | 0.19 (0.01) |
| | CRPS | 89.1 | 118.65 | 126.97 | 177.96 | 128.32 | 110.43 | 151.24 | 95.32 (1.19) | 251.45 (0.84) | 336.26 (1.84) | 218.96 (1.81) | **63.05 (0.34)** | 87.82 (1.46) | 61.95 (0.61) | 69.86 (1.00) |
| | Winkler Score | - | - | - | - | - | - | - | 1691.99 (43.58) | 6160.70 (154.79) | 6830.05 (97.82) | 4684.26 (127.85) | 2011.11 (26.89) | 1211.44 (50.65) | **1187.96 (21.32)** | 1122.44 (53.26) |
| Belgium COVID-19 | SMAPE | 70.71 | 72.37 | 97.49 | 82.72 | 109.5 | 97.49 | 105.22 | 73.27 (0.64) | 78.65 (0.72) | 126.89 (25.12) | 86.70 (1.36) | 62.40 (0.06) | **61.60 (0.58)** | 46.99 (1.08) | 68.02 (0.20) |
| | MAE | **81.52** | 90.76 | 126.95 | 177.6 | 141.8 | 132.48 | 343.76 | 117.87 (2.54) | 119.80 (1.14) | 232.83 (71.70) | 157.74 (5.41) | 87.14 (0.12) | 86.08 (1.13) | 55.13 (0.86) | 103.29 (0.54) |
| | RMSE | **107.67** | 115.59 | 151.97 | 203.19 | 207 | 189.31 | 421.59 | 156.11 (2.83) | 172.89 (1.22) | 293.61 (79.12) | 174.38 (5.29) | 134.65 (0.12) | 114.83 (1.56) | 82.02 (1.44) | 140.29 (0.68) |
| | MASE | **0.42** | 0.49 | 0.66 | 0.89 | 0.72 | 0.68 | 1.76 | 0.60 (0.01) | 0.61 (0.01) | 1.19 (0.37) | 0.81 (0.03) | 0.44 (0.00) | 0.44 (0.01) | 0.28 (0.00) | 0.53 (0.00) |
| | RMSSE | **0.22** | 0.27 | 0.35 | 0.45 | 0.42 | 0.39 | 0.86 | 0.32 (0.01) | 0.35 (0.00) | 0.60 (0.16) | 0.36 (0.01) | 0.27 (0.00) | 0.22 (0.01) | 0.17 (0.00) | 0.29 (0.00) |
| | Pinball-80 | 48.11 | 35.59 | 80.64 | 49.25 | 93.15 | 98.06 | 69.72 | 47.62 (1.18) | 51.68 (0.73) | 287.23 (31.08) | 59.58 (1.49) | 46.05 (0.10) | **30.11 (0.71)** | 22.20 (0.6) | 38.30 (0.24) |
| | Pinball-95 | 51.79 | 30.7 | 89.22 | 29.47 | 104.28 | 113.96 | 18.64 | 20.47 (0.58) | 24.25 (1.08) | 141.83 (10.06) | 23.16 (0.55) | 45.80 (0.18) | **11.35 (0.70)** | 10.17 (0.32) | 20.93 (0.47) |
| | ρ-0.5 Risk | **0.45** | 0.51 | 0.72 | 1 | 0.8 | 0.75 | 1.94 | 0.67 (0.01) | 0.67 (0.01) | 1.29 (0.38) | 0.90 (0.03) | 0.49 (0.00) | 0.49 (0.01) | 0.31 (0.00) | 0.58 (0.00) |
| | ρ-0.9 Risk | 0.55 | 0.36 | 0.97 | 0.41 | 1.13 | 1.23 | 0.4 | 0.36 (0.01) | 0.41 (0.01) | 2.49 (0.19) | 0.43 (0.01) | 0.52 (0.00) | **0.18 (0.00)** | 0.18 (0.00) | 0.32 (0.00) |
| | CRPS | 81.52 | 90.76 | 126.95 | 177.6 | 141.8 | 132.48 | 343.76 | 83.58 (1.52) | 90.34 (0.75) | 440.50 (25.57) | 111.87 (4.05) | 84.51 (0.14) | **60.71 (0.79)** | 42.38 (0.59) | 81.44 (0.44) |
| | Winkler Score | - | - | - | - | - | - | - | 761.72 (22.68) | 1227.73 (39.83) | 6796.43 (347.11) | 867.90 (52.97) | 3160.21 (14.32) | **599.41 (20.14)** | 561.29 (14.58) | 1499.49 (37.99) |
| Colombia Dengue | SMAPE | 32.7 | 48.98 | 50.37 | 73.73 | 35.89 | **30.97** | 82.81 | 65.61 (0.97) | 35.22 (0.57) | 105.96 (3.00) | 61.42 (0.67) | 51.02 (0.75) | 43.32 (1.46) | 29.24 (1.17) | 36.20 (0.99) |
| | MAE | 13.42 | 16.23 | 13.29 | 43.36 | 12.3 | **7.99** | 44.55 | 26.43 (0.29) | 23.25 (0.20) | 48.33 (0.85) | 24.05 (0.22) | 13.41 (0.08) | 12.22 (0.33) | 8.31 (0.15) | 8.24 (0.20) |
| | RMSE | 30.24 | 17.94 | 15.05 | 44.49 | 24.94 | **12.73** | 96.36 | 55.67 (0.85) | 73.55 (0.48) | 97.92 (1.04) | 62.72 (0.75) | 21.74 (0.11) | 23.28 (0.29) | 13.14 (0.21) | 13.22 (0.32) |
| | MASE | 1.82 | 2.08 | 2.02 | 4.69 | 1.67 | **1.08** | 6.04 | 3.58 (0.04) | 3.15 (0.03) | 6.55 (0.12) | 3.26 (0.03) | 1.82 (0.01) | 1.66 (0.04) | 1.12 (0.02) | 1.12 (0.03) |
| | RMSSE | 1.98 | 1.4 | 1.31 | 2.88 | 1.64 | **0.84** | 6.32 | 3.65 (0.06) | 4.83 (0.03) | 6.43 (0.07) | 4.12 (0.05) | 1.43 (0.01) | 1.53 (0.02) | 0.86 (0.01) | 0.86 (0.02) |
| | Pinball-80 | 8.23 | 10.21 | 7.44 | 20.78 | 6.63 | 4.58 | 25.72 | 12.15 (0.20) | 12.28 (0.17) | 26.96 (0.38) | 12.97 (0.18) | 5.96 (0.06) | 6.22 (0.17) | **3.73 (0.08)** | 3.72 (0.09) |
| | Pinball-95 | 9 | 11.26 | 7.83 | 20.33 | 6.87 | 4.88 | 27.45 | 8.20 (0.24) | 9.35 (0.45) | 16.95 (0.32) | 10.26 (0.27) | 4.87 (0.08) | 5.29 (0.08) | **2.35 (0.09)** | 2.12 (0.08) |
| | ρ-0.5 Risk | 0.26 | 0.32 | 0.26 | 0.84 | 0.24 | **0.16** | 0.87 | 0.51 (0.01) | 0.45 (0.00) | 0.94 (0.02) | 0.46 (0.00) | 0.26 (0.00) | 0.24 (0.01) | 0.16 (0.00) | 0.16 (0.00) |
| | ρ-0.9 Risk | 0.34 | 0.42 | 0.3 | 0.8 | 0.26 | 0.19 | 1.04 | 0.39 (0.01) | 0.41 (0.01) | 0.83 (0.01) | 0.44 (0.01) | 0.20 (0.00) | 0.22 (0.00) | **0.12 (0.00)** | 0.11 (0.00) |
| | CRPS | 13.42 | 16.23 | 13.29 | 43.36 | 12.3 | 7.99 | 44.55 | 20.27 (0.24) | 18.73 (0.19) | 41.88 (0.41) | 19.53 (0.19) | 11.94 (0.06) | 10.16 (0.13) | **6.49 (0.08)** | 6.42 (0.11) |
| | Winkler Score | - | - | - | - | - | - | - | 342.37 (14.29) | 368.89 (34.12) | 717.57 (14.68) | 414.97 (11.95) | 361.49 (3.79) | 230.14 (2.46) | **112.29 (3.87)** | 95.18 (4.23) |
| Hungary Chickenpox | SMAPE | 84.27 | 78.75 | 81.2 | 78.95 | 94.54 | 79.29 | 86.42 | 84.79 (0.70) | 85.22 (0.67) | 86.41 (1.67) | 85.81 (1.09) | 82.65 (0.21) | **76.78 (0.53)** | 80.24 (0.48) | 78.19 (0.24) |
| | MAE | 23.65 | 22.17 | 24.05 | 22.59 | 22.37 | 20.92 | 27.21 | 25.46 (0.32) | 21.99 (0.24) | 24.36 (0.48) | 25.01 (0.44) | 23.69 (0.08) | **20.74 (0.16)** | 20.54 (0.10) | 21.74 (0.08) |
| | RMSE | 38.34 | **26.89** | 27.75 | 27.1 | 40 | 37.8 | 39.68 | 36.06 (0.47) | 38.26 (0.54) | 44.82 (0.63) | 41.31 (0.55) | 36.34 (0.10) | 35.27 (0.20) | 36.19 (0.09) | 35.52 (0.10) |
| | MASE | 1.23 | 1.07 | 1.17 | **1.08** | 1.16 | 1.09 | 1.42 | 1.33 (0.02) | 1.14 (0.01) | 1.27 (0.03) | 1.30 (0.02) | 1.23 (0.00) | 1.08 (0.01) | **1.07 (0.01)** | 1.13 (0.00) |
| | RMSSE | 1.2 | 0.84 | 0.87 | 0.84 | 1.26 | 1.19 | 1.25 | 1.13 (0.01) | 1.20 (0.02) | 1.41 (0.02) | 1.30 (0.02) | 1.14 (0.00) | 1.11 (0.01) | 1.14 (0.00) | 1.12 (0.00) |
| | Pinball-80 | 12.16 | 10.76 | 10.25 | 10.99 | 14.88 | 12.5 | 11.28 | **9.58 (0.14)** | 11.25 (0.15) | 12.82 (0.21) | 12.25 (0.24) | 10.70 (0.03) | 9.40 (0.09) | 10.52 (0.06) | 10.60 (0.05) |
| | Pinball-95 | 12.32 | 10.6 | 9.36 | 10.84 | 16.72 | 13.52 | 10.11 | **3.98 (0.21)** | 6.81 (0.48) | 7.82 (0.19) | 7.29 (0.28) | 9.54 (0.04) | 6.22 (0.13) | 8.58 (0.16) | 8.97 (0.10) |
| | ρ-0.5 Risk | 0.76 | 0.71 | 0.77 | 0.72 | 0.72 | 0.67 | 0.87 | 0.82 (0.01) | 0.70 (0.01) | 0.77 (0.02) | 0.80 (0.01) | 0.76 (0.00) | 0.67 (0.01) | **0.66 (0.00)** | 0.70 (0.00) |
| | ρ-0.9 Risk | 0.79 | 0.68 | 0.62 | 0.7 | 1.03 | 0.85 | 0.67 | **0.41 (0.01)** | 0.57 (0.02) | 0.64 (0.01) | 0.60 (0.02) | 0.63 (0.00) | 0.49 (0.01) | 0.61 (0.01) | 0.63 (0.00) |
| | CRPS | 23.65 | 22.17 | 24.05 | 22.59 | 22.37 | 20.92 | 27.21 | 18.02 (0.20) | **16.95 (0.21)** | 19.72 (0.22) | 18.90 (0.29) | 22.50 (0.06) | 16.27 (0.12) | 17.32 (0.09) | 19.13 (0.05) |
| | Winkler Score | - | - | - | - | - | - | - | **147.45 (9.92)** | 235.52 (21.95) | 324.32 (7.10) | 262.94 (12.45) | 798.71 (4.09) | 290.08 (7.47) | 420.70 (8.76) | 548.79 (6.64) |

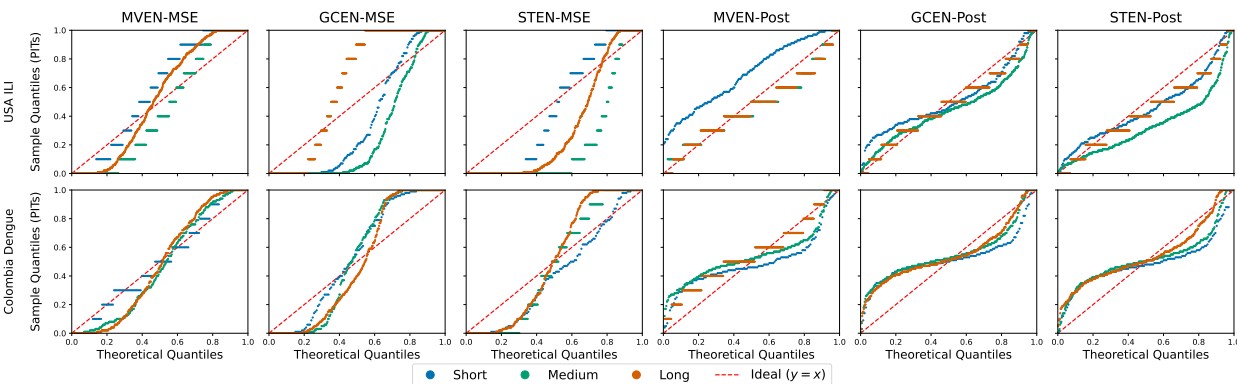

Fig. 14: PIT Q-Q plots for the modified models considered in the ablation study on USA ILI and Colombia dengue datasets.

Table 12: Mean forecast performance (along with standard deviations) of the proposed and benchmark models on all six datasets for the medium-term forecast horizons (60-day/9-week/12-month). The **best** and **second best** results are highlighted.

| Dataset | Metric | Temporal Models | | | | Spatiotemporal Models | | | DeepAR | Probabilistic Models (Temporal & Spatiotemporal) | | | | ST-Engression Models (Proposed) | | |
|---|---|---|---|---|---|---|---|---|---|---|---|---|---|---|---|---|
| | | LSTM | NHiTS | Transformers | TCN | GSTAR | STARMA | STGCN | | Prob-iTransformer | GpGp | DiffSTG | STESN | MVEN | GCEN | STEN |
| Japan TB | SMAPE | 23.21 | 23.53 | 24 | 37.48 | 25.06 | 25.39 | 44.43 | 30.16 (0.29) | 37.54 (0.17) | 105.99 (4.92) | 87.55 (0.99) | 23.32 (0.09) | 23.79 (0.11) | **22.41 (0.11)** | 22.58 (0.09) |
| | MAE | 5.79 | 5.85 | 6.03 | 10.72 | 5.64 | 5.78 | 16.21 | 9.11 (0.14) | 11.35 (0.11) | 52.60 (7.44) | 32.55 (0.57) | **5.48 (0.01)** | 6.12 (0.03) | 5.42 (0.02) | 5.58 (0.02) |
| | RMSE | 9.35 | 6.98 | 7.19 | 12.03 | 8.49 | 8.76 | 27.94 | 15.23 (0.25) | 19.42 (0.24) | 63.08 (7.95) | 40.37 (0.67) | 8.29 (0.04) | 10.20 (0.06) | 8.99 (0.04) | 9.29 (0.06) |
| | MASE | 0.67 | 0.64 | 0.65 | 1.11 | 0.65 | 0.67 | 1.87 | 1.05 (0.02) | 1.31 (0.01) | 6.08 (0.86) | 3.76 (0.07) | **0.63 (0.01)** | 0.71 (0.00) | 0.63 (0.00) | 0.64 (0.00) |
| | RMSSE | 0.7 | 0.6 | 0.62 | 0.98 | 0.63 | 0.65 | 2.08 | 1.13 (0.02) | 1.44 (0.02) | 4.69 (0.59) | 3.00 (0.05) | **0.62 (0.00)** | 0.76 (0.00) | 0.67 (0.00) | 0.69 (0.00) |
| | Pinball-80 | 2 | 2.14 | 2.1 | 3.4 | 2.17 | 3.22 | 3.38 | 3.24 (0.03) | 4.16 (0.06) | 27.72 (1.97) | 19.98 (0.16) | 2.01 (0.01) | **1.90 (0.01)** | 1.97 (0.01) | 1.93 (0.00) |
| | Pinball-95 | 1.55 | 1.75 | 1.64 | 2.42 | 1.85 | 3.38 | 1.01 | 1.16 (0.01) | 1.42 (0.05) | 11.10 (0.62) | 8.40 (0.05) | 1.37 (0.01) | **0.82 (0.01)** | 1.07 (0.01) | 0.92 (0.01) |
| | ρ-0.5 Risk | 0.18 | 0.18 | 0.19 | 0.33 | 0.17 | 0.18 | 0.5 | 0.28 (0.00) | 0.35 (0.00) | 1.65 (0.23) | 1.03 (0.02) | **0.17 (0.00)** | 0.19 (0.00) | 0.17 (0.00) | 0.17 (0.00) |
| | ρ-0.9 Risk | 0.11 | 0.12 | 0.11 | 0.17 | 0.12 | 0.21 | 0.11 | 0.12 (0.00) | 0.15 (0.00) | 1.12 (0.07) | 0.84 (0.01) | **0.09 (0.00)** | 0.08 (0.00) | 0.09 (0.00) | 0.08 (0.00) |
| | CRPS | 5.79 | 5.85 | 6.03 | 10.72 | 5.64 | 5.78 | 16.21 | 6.35 (0.10) | 8.07 (0.05) | 39.05 (3.07) | 27.88 (0.20) | 4.78 (0.01) | 5.10 (0.02) | **4.44 (0.01)** | 4.46 (0.02) |
| | Winkler Score | - | - | - | - | - | - | - | **53.84 (1.40)** | 68.41 (1.54) | 431.31 (18.03) | 344.98 (1.90) | 141.29 (0.94) | 119.69 (1.18) | 97.62 (1.23) | 86.96 (1.11) |
| China TB | SMAPE | 14.96 | **12.91** | 15.29 | 31.01 | 12.3 | 29.03 | 19.39 | 17.11 (0.14) | 122.25 (0.24) | 64.06 (0.94) | 65.28 (0.66) | 15.35 (0.02) | 15.51 (0.11) | 16.72 (0.07) | 17.04 (0.07) |
| | MAE | 350.56 | **328.93** | 368.59 | 544.52 | 320.86 | 601.67 | 486.08 | 405.15 (4.61) | 1884.40 (1.36) | 1269.76 (17.41) | 1370.15 (16.83) | 363.11 (0.63) | 359.42 (1.98) | 415.46 (1.95) | 404.03 (1.84) |
| | RMSE | 728.07 | 440.73 | 450.99 | 625.19 | 951.75 | 963.32 | 1226.49 | 718.91 (5.83) | 2470.48 (1.30) | 1705.62 (25.34) | 1728.10 (17.30) | 666.15 (0.61) | 744.89 (4.21) | 932.84 (5.01) | 812.76 (5.40) |
| | MASE | 1.49 | 1.21 | 1.42 | 2.19 | **1.36** | 2.56 | 2.06 | 1.72 (0.02) | 8.00 (0.01) | 5.39 (0.07) | 5.82 (0.07) | 1.54 (0.00) | 1.53 (0.01) | 1.76 (0.01) | 1.72 (0.01) |
| | RMSSE | 0.82 | 1.05 | 1.22 | 1.74 | 1.07 | 1.08 | 1.38 | **0.81 (0.01)** | 2.78 (0.00) | 1.92 (0.03) | 1.94 (0.02) | **0.75 (0.00)** | 0.81 (0.02) | 1.05 (0.01) | 0.91 (0.01) |
| | Pinball-80 | 168.19 | 151.57 | 161.44 | 351.68 | 139.57 | 437.17 | 206.25 | 177.31 (1.70) | 1339.22 (1.72) | 517.96 (5.73) | 622.62 (7.29) | 158.26 (0.31) | 157.36 (1.26) | 167.72 (0.68) | 162.49 (1.11) |
| | Pinball-95 | 164.64 | 145.13 | 150.02 | 391.38 | 129.13 | 505.34 | 187.85 | 99.45 (1.58) | 1301.16 (7.28) | 220.75 (4.28) | 256.89 (3.50) | 135.32 (1.06) | 112.32 (1.64) | 123.15 (0.84) | 112.83 (1.23) |
| | ρ-0.5 Risk | 0.16 | **0.15** | 0.17 | 0.25 | 0.14 | 0.27 | 0.22 | 0.18 (0.00) | 0.85 (0.00) | 0.57 (0.01) | 0.62 (0.01) | 0.16 (0.00) | **0.15 (0.01)** | 0.19 (0.00) | 0.18 (0.00) |
| | ρ-0.9 Risk | 0.15 | **0.13** | 0.14 | 0.34 | 0.12 | 0.44 | 0.18 | **0.13 (0.01)** | 1.24 (0.00) | 0.32 (0.00) | 0.38 (0.00) | **0.13 (0.00)** | **0.12 (0.00)** | 0.13 (0.00) | **0.12 (0.00)** |
| | CRPS | 350.56 | 328.93 | 368.59 | 544.52 | 320.86 | 601.67 | 486.08 | 328.87 (2.12) | 1704.94 (2.42) | 955.18 (10.12) | 1062.12 (7.25) | 343.40 (1.13) | 304.96 (1.82) | 361.44 (1.71) | 347.77 (2.47) |
| | Winkler Score | - | - | - | - | - | - | - | 5154.01 (86.01) | 48839.51 (542.86) | 9957.39 (193.71) | 11358.48 (112.90) | 11485.91 (175.47) | 7294.79 (70.88) | 9992.48 (157.22) | 8795.67 (202.00) |
| USA ILI | SMAPE | **38.34** | 54.78 | 50.95 | 58.12 | 63.11 | 44.55 | 60.53 | 75.56 (1.77) | 66.40 (0.43) | 105.39 (1.57) | 96.98 (1.40) | 38.6 (0.09) | **38.47 (0.41)** | 39.50 (0.27) | 38.90 (0.28) |
| | MAE | **73.44** | 124.71 | 116.66 | 153.91 | 142.27 | 119.11 | 128.19 | 168.66 (2.93) | 228.69 (0.53) | 320.74 (4.45) | 262.60 (2.89) | 74.26 (0.32) | 97.08 (1.78) | 75.29 (0.98) | 72.27 (1.62) |
| | RMSE | 142.72 | 137.44 | 131.24 | 170.97 | 260.38 | 266.33 | 255.1 | 348.57 (5.33) | 511.10 (0.76) | 528.00 (3.44) | 495.49 (3.09) | 143.65 (0.84) | 222.87 (4.85) | 135.37 (2.96) | 131.13 (4.77) |
| | MASE | **1.96** | 3.24 | 2.97 | 3.39 | 3.8 | 3.18 | 3.42 | 4.51 (0.08) | 6.11 (0.01) | 8.57 (0.12) | 7.02 (0.08) | 1.98 (0.01) | 2.59 (0.05) | 2.01 (0.03) | 1.93 (0.04) |
| | RMSSE | 1.38 | 2.12 | 2.03 | 2.25 | 2.51 | 2.57 | 2.46 | 3.37 (0.05) | 4.93 (0.01) | 5.10 (0.03) | 4.78 (0.03) | 1.39 (0.01) | 2.15 (0.05) | **1.31 (0.03)** | 1.27 (0.05) |
| | Pinball-80 | 39.26 | 61.72 | 54.71 | 112.39 | 112.83 | 90.31 | 80.83 | 97.27 (3.17) | 144.17 (0.54) | 172.47 (1.44) | 152.81 (1.31) | 34.11 (0.22) | 47.02 (0.78) | **31.62 (0.44)** | 30.64 (0.35) |
| | Pinball-95 | 40.53 | 61.41 | 52.9 | 130.11 | 133.67 | 105.69 | 89.2 | 73.74 (5.51) | 123.24 (1.50) | 99.21 (1.40) | 91.30 (1.83) | 28.59 (0.38) | 35.85 (0.86) | **18.09 (0.34)** | 18.00 (0.21) |
| | ρ-0.5 Risk | **0.23** | 0.39 | 0.37 | 0.49 | 0.45 | 0.38 | 0.4 | 0.53 (0.01) | 0.72 (0.00) | 1.02 (0.01) | 0.83 (0.01) | **0.24 (0.01)** | 0.31 (0.01) | 0.24 (0.00) | 0.23 (0.01) |
| | ρ-0.9 Risk | 0.25 | 0.39 | 0.34 | 0.78 | 0.8 | 0.63 | 0.55 | 0.55 (0.03) | 0.87 (0.01) | 0.84 (0.01) | 0.77 (0.01) | **0.19 (0.00)** | 0.26 (0.01) | 0.15 (0.00) | 0.15 (0.00) |
| | CRPS | 73.44 | 124.71 | 116.66 | 153.91 | 142.27 | 119.11 | 128.19 | 136.72 (3.38) | 197.32 (0.57) | 270.35 (1.82) | 226.45 (1.39) | 67.18 (0.34) | 75.62 (0.93) | **56.67 (0.59)** | 55.72 (0.75) |
| | Winkler Score | - | - | - | - | - | - | - | 2720.32 (257.44) | 4689.32 (90.29) | 4439.78 (60.03) | 3568.76 (72.75) | 2105.99 (24.57) | 1432.37 (32.92) | 794.90 (20.31) | 879.44 (15.03) |
| Belgium COVID-19 | SMAPE | **64.04** | 75.63 | 106.21 | 123.67 | 123.5 | 135.81 | 121.83 | 84.02 (0.89) | 75.77 (0.32) | 145.07 (21.67) | 120.62 (0.97) | 65.99 (0.06) | 67.43 (0.67) | **57.10 (0.42)** | 70.99 (0.09) |
| | MAE | 69.87 | 103.86 | 111.21 | 418.62 | 291.7 | 123.68 | 366.12 | 94.18 (0.63) | 91.53 (0.52) | 262.35 (61.50) | 302.94 (5.99) | **69.07 (0.07)** | 77.55 (0.47) | 58.29 (0.42) | 87.54 (0.13) |
| | RMSE | **100.37** | 121.33 | 145.81 | 439.21 | 419.51 | 186.89 | 424.76 | 141.92 (1.25) | 133.87 (0.71) | 326.99 (69.11) | 327.39 (6.13) | 110.47 (0.10) | 111.76 (0.62) | **85.91 (0.55)** | 121.89 (0.11) |
| | MASE | 0.35 | 0.49 | 0.57 | 1.94 | 1.45 | 0.61 | 1.81 | 0.47 (0.00) | 0.45 (0.00) | 1.30 (0.30) | 1.50 (0.03) | **0.34 (0.00)** | 0.34 (0.02) | 0.29 (0.00) | 0.43 (0.00) |
| | RMSSE | **0.2** | 0.26 | 0.33 | 0.92 | 0.84 | 0.37 | 0.85 | 0.28 (0.00) | 0.27 (0.00) | 0.65 (0.14) | 0.65 (0.01) | 0.22 (0.00) | 0.22 (0.00) | **0.17 (0.00)** | 0.24 (0.00) |
| | Pinball-80 | 27.02 | 24.25 | 71.26 | 84.04 | 87.61 | 96.08 | 74.05 | 44.83 (0.88) | 39.70 (0.39) | 354.73 (34.89) | 124.04 (1.53) | 36.10 (0.10) | 29.79 (0.28) | **23.97 (0.18)** | 37.72 (0.10) |
| | Pinball-95 | 23.06 | 10.41 | 79.09 | 21.41 | 58.49 | 113.21 | 19.55 | 25.60 (1.68) | 19.58 (0.47) | 174.00 (11.56) | 48.16 (0.54) | 35.62 (0.18) | 16.14 (0.25) | **14.43 (0.27)** | 30.74 (0.22) |
| | ρ-0.5 Risk | 0.5 | 0.75 | 0.8 | 3.03 | 2.11 | 0.89 | 2.65 | 0.68 (0.00) | 0.66 (0.00) | 1.85 (0.42) | 2.22 (0.04) | **0.50 (0.00)** | 0.56 (0.00) | 0.42 (0.00) | 0.63 (0.00) |
| | ρ-0.9 Risk | 0.35 | 0.22 | 1.11 | 0.61 | 0.99 | 1.5 | 0.45 | 0.41 (0.01) | 0.43 (0.01) | 3.93 (0.29) | 1.16 (0.01) | 0.52 (0.00) | 0.32 (0.00) | **0.27 (0.00)** | 0.49 (0.00) |
| | CRPS | 69.87 | 103.86 | 111.21 | 418.62 | 291.7 | 123.68 | 366.12 | 70.40 (0.89) | 69.08 (0.34) | 533.06 (23.97) | 201.96 (3.61) | 66.80 (0.09) | 61.99 (0.33) | **48.13 (0.25)** | 81.24 (0.15) |
| | Winkler Score | - | - | - | - | - | - | - | 913.66 (77.80) | 993.36 (21.10) | 8312.55 (369.09) | 1362.17 (15.46) | 2481.60 (9.53) | 1229.14 (18.58) | 979.99 (26.68) | 2708.76 (17.32) |
| Colombia Dengue | SMAPE | 37.06 | 54.03 | 74.49 | 75.94 | 44.15 | 37.83 | 76.53 | 56.30 (0.68) | 38.03 (0.51) | 105.81 (2.21) | 65.62 (0.87) | 54.14 (0.82) | 39.14 (1.56) | **37.01 (1.01)** | 39.07 (1.32) |
| | MAE | 13.1 | 26.35 | 26.53 | 42.55 | 17.87 | **14.03** | 40.69 | 24.94 (0.28) | 24.20 (0.16) | 45.31 (0.63) | 27.77 (0.25) | 16.36 (0.10) | 16.35 (0.27) | 15.86 (0.18) | 15.79 (0.16) |
| | RMSE | 31.16 | 28.18 | 28.8 | 43.84 | 46.27 | 33.12 | 83.37 | 58.37 (0.63) | 74.15 (0.43) | 92.72 (0.73) | 73.00 (0.46) | 29.37 (0.20) | 43.26 (0.78) | 43.08 (0.79) | 43.56 (0.56) |
| | MASE | 1.78 | 3.05 | 3.24 | 4.51 | 2.43 | **1.9** | 5.52 | 3.38 (0.04) | 3.28 (0.02) | 6.15 (0.09) | 3.77 (0.03) | 2.22 (0.01) | 2.22 (0.04) | 2.15 (0.02) | 2.14 (0.02) |
| | RMSSE | 2.05 | 1.99 | 2.15 | 2.81 | 3.04 | 2.17 | 5.47 | 3.83 (0.04) | 4.87 (0.03) | 6.09 (0.05) | 4.79 (0.03) | **1.93 (0.01)** | 2.84 (0.05) | 2.83 (0.05) | 2.86 (0.04) |
| | Pinball-80 | 9.32 | 16.23 | 19.61 | 22.58 | 12.21 | 8.87 | 22.18 | 12.42 (0.19) | 12.43 (0.15) | 24.79 (0.21) | 16.30 (0.19) | **7.91 (0.07)** | 7.89 (0.23) | 8.33 (0.24) | 8.33 (0.13) |
| | Pinball-95 | 10.7 | 17.76 | 22.78 | 23.24 | 13.85 | 9.8 | 23.1 | 8.70 (0.25) | 9.21 (0.41) | 16.15 (0.20) | 13.62 (0.29) | 6.62 (0.09) | **5.39 (0.26)** | 6.22 (0.28) | 6.85 (0.23) |
| | ρ-0.5 Risk | 0.26 | 0.52 | 0.52 | 0.84 | 0.35 | **0.28** | 0.8 | 0.49 (0.01) | 0.47 (0.00) | 0.89 (0.01) | 0.55 (0.00) | 0.32 (0.00) | 0.32 (0.01) | 0.31 (0.00) | 0.31 (0.00) |
| | ρ-0.9 Risk | 0.4 | 0.68 | 0.86 | 0.91 | 0.53 | 0.37 | 0.9 | 0.41 (0.01) | 0.43 (0.00) | 0.79 (0.01) | 0.59 (0.01) | **0.26 (0.01)** | 0.26 (0.01) | 0.28 (0.01) | 0.30 (0.01) |
| | CRPS | 13.1 | 26.35 | 26.53 | 42.55 | 17.87 | 14.03 | 40.69 | 19.27 (0.21) | 19.56 (0.15) | 40.23 (0.29) | 23.02 (0.22) | 14.50 (0.08) | **12.69 (0.20)** | 12.80 (0.18) | 12.95 (0.16) |
| | Winkler Score | - | - | - | - | - | - | - | 339.66 (11.59) | 360.26 (28.63) | 695.43 (9.77) | 533.03 (11.79) | | 218.35 (10.10) | **263.45 (12.12)** | 292.44 (11.94) |
| Hungary Chickenpox | SMAPE | 75.93 | 90.57 | 86.15 | 89.03 | 140.47 | 85.86 | 96.71 | 85.88 (0.72) | 83.32 (0.41) | 89.73 (2.57) | 93.26 (0.89) | 85.67 (0.19) | 76.01 (0.81) | **73.58 (0.39)** | 76.55 (0.34) |
| | MAE | 16.56 | 21.44 | 20.21 | 22.2 | 20.67 | 16.74 | 26.19 | 20.80 (0.41) | 18.28 (0.18) | 18.61 (0.29) | 23.17 (0.42) | 19.64 (0.09) | 16.41 (0.13) | **15.32 (0.09)** | 16.11 (0.10) |
| | RMSE | 26.76 | 26.61 | 24.57 | 26.22 | 37.29 | 34.56 | 34.61 | 29.66 (0.49) | 28.75 (0.36) | 35.76 (0.56) | 35.54 (0.34) | 28.95 (0.07) | 27.15 (0.14) | 26.60 (0.10) | 27.31 (0.10) |
| | MASE | 0.86 | 1.09 | 1.03 | 1.11 | 1.07 | 0.87 | 1.36 | 1.08 (0.02) | 0.95 (0.01) | 0.97 (0.02) | 1.20 (0.02) | 1.02 (0.00) | 0.85 (0.01) | **0.79 (0.00)** | 0.84 (0.01) |
| | RMSSE | 0.84 | 0.87 | 0.8 | 0.84 | 1.17 | 1.08 | 1.08 | 0.93 (0.02) | 0.90 (0.01) | 1.12 (0.02) | 1.11 (0.01) | 0.91 (0.00) | 0.85 (0.00) | **0.83 (0.00)** | 0.86 (0.00) |
| | Pinball-80 | 7.13 | 8.11 | 7.71 | 7.88 | 15.55 | 12.27 | 8.26 | 8.13 (0.10) | 7.64 (0.11) | 10.25 (0.17) | 11.26 (0.14) | 7.72 (0.02) | 6.96 (0.05) | **6.89 (0.05)** | 7.22 (0.04) |
| | Pinball-95 | 6.55 | 6.81 | 6.51 | 6.27 | 18.15 | 14.22 | 5.85 | **3.19 (0.10)** | 3.74 (0.20) | 5.68 (0.14) | 5.70 (0.14) | 6.01 (0.03) | 4.22 (0.07) | 4.83 (0.11) | 5.27 (0.11) |
| | ρ-0.5 Risk | 0.7 | 0.9 | 0.85 | 0.93 | 0.87 | 0.7 | 1.1 | 0.88 (0.02) | 0.77 (0.01) | 0.77 (0.01) | 0.97 (0.02) | 0.83 (0.00) | 0.69 (0.01) | **0.65 (0.00)** | 0.68 (0.00) |
| | ρ-0.9 Risk | 0.57 | 0.61 | 0.58 | 0.57 | 1.46 | 1.14 | 0.56 | **0.45 (0.01)** | 0.45 (0.01) | 0.65 (0.01) | 0.67 (0.01) | 0.54 (0.00) | **0.45 (0.00)** | 0.49 (0.01) | 0.52 (0.01) |
| | CRPS | 16.56 | 21.44 | 20.21 | 22.2 | 20.67 | 16.74 | 26.19 | 14.38 (0.26) | 13.11 (0.13) | 16.63 (0.20) | 16.99 (0.24) | 18.21 (0.08) | **12.88 (0.08)** | 12.90 (0.06) | 13.71 (0.08) |
| | Winkler Score | - | - | - | - | - | - | - | **104.46 (5.64)** | 133.58 (11.57) | 245.64 (6.43) | 192.39 (6.75) | 605.78 (4.42) | 219.00 (7.10) | 318.03 (5.71) | 353.77 (6.82) |

Table 13: Mean forecast performance (along with standard deviations) of the proposed and benchmark models on all six datasets for the long-term forecast horizons (90-day/13-week/24-month). The **best** and **second best** results are highlighted.

| Dataset | Metric | LSTM | NHiTS | Transformers | TCN | GSTAR | STARMA | STGCN | DeepAR | Prob-iTransformer | GpGp | DiffSTG | STESN | MVEN | GCEN | STEN |
|---|---|---|---|---|---|---|---|---|---|---|---|---|---|---|---|---|
| | | | | Temporal Models | | | Spatiotemporal Models | | | | Probabilistic Models (Temporal & Spatiotemporal) | | | | ST-Engression Models (Proposed) | | |
| Japan TB | SMAPE | 25.37 | 23.88 | **19.06** | 27.81 | 26.14 | 26.33 | 39.24 | 30.79 (0.24) | 47.01 (0.12) | 125.67 (5.23) | 92.02 (0.69) | 23.16 (0.03) | 23.24 (0.06) | **22.38 (0.06)** | 22.45 (0.04) |
| | MAE | 7.54 | 6.72 | 475.54 | 578.56 | 6.58 | 6.07 | 14.54 | 9.91 (0.10) | 15.40 (0.08) | 106.12 (14.70) | 36.54 (0.39) | 6.09 (0.01) | 6.37 (0.03) | **5.99 (0.02)** | **6.00 (0.01)** |
| | RMSE | 13.04 | **8.06** | 516.28 | 618.39 | 10.31 | **9.21** | 27.57 | 16.51 (0.18) | 24.11 (0.18) | 127.24 (16.56) | 45.95 (0.40) | 10.31 (0.02) | 11.01 (0.07) | 10.51 (0.04) | 10.45 (0.03) |
| | MASE | 0.86 | **0.68** | 1.66 | 2.18 | 0.75 | **0.69** | 1.66 | 1.13 (0.01) | 1.76 (0.01) | 12.12 (1.68) | 4.17 (0.04) | 0.70 (0.00) | 0.74 (0.00) | **0.69 (0.01)** | 0.70 (0.00) |
| | RMSSE | 0.96 | **0.65** | 1.28 | 1.66 | 0.76 | **0.68** | 2.03 | 1.21 (0.01) | 1.77 (0.01) | 9.35 (1.22) | 3.38 (0.03) | 0.76 (0.00) | 0.82 (0.01) | 0.79 (0.00) | 0.78 (0.00) |
| | Pinball-80 | **1.94** | **1.99** | 168.98 | 293.54 | 2.26 | 3.38 | 3.38 | 3.44 (0.03) | 5.55 (0.04) | 48.96 (3.26) | 22.86 (0.12) | 2.11 (0.01) | **1.94 (0.01)** | 2.05 (0.01) | 2.11 (0.01) |
| | Pinball-95 | **1.03** | 1.31 | 134.59 | 295.67 | 1.74 | 3.55 | 1.43 | 1.21 (0.01) | 1.86 (0.05) | 18.74 (1.08) | 9.45 (0.05) | 1.48 (0.01) | **0.92 (0.01)** | 1.06 (0.01) | 1.08 (0.02) |
| | ρ-0.5 Risk | 0.22 | 0.2 | 0.19 | 0.36 | 0.2 | **0.18** | 0.43 | 0.30 (0.00) | 0.46 (0.00) | 3.21 (0.44) | 1.11 (0.01) | **0.18 (0.00)** | **0.19 (0.00)** | **0.18 (0.00)** | **0.18 (0.00)** |
| | ρ-0.9 Risk | **0.08** | **0.09** | 0.14 | 0.1 | 0.11 | 0.21 | 0.12 | 0.13 (0.00) | 0.20 (0.00) | 1.87 (0.11) | 0.93 (0.00) | **0.09 (0.00)** | 0.08 (0.00) | **0.08 (0.01)** | **0.09 (0.00)** |
| | CRPS | 7.54 | 6.72 | 475.54 | 578.56 | 6.58 | 6.07 | 14.54 | 6.97 (0.07) | 11.15 (0.05) | 70.39 (6.79) | 32.04 (0.13) | 5.78 (0.01) | **5.69 (0.02)** | **5.08 (0.01)** | 5.36 (0.01) |
| | Winkler Score | - | - | - | - | - | - | - | **61.03 (1.28)** | 103.48 (1.42) | 650.09 (41.34) | 395.52 (1.72) | 205.30 (0.87) | 169.59 (1.18) | 129.10 (0.61) | 157.69 (0.93) |
| USA ILI | SMAPE | **41.72** | 50.84 | 53.97 | 60.72 | 104.89 | 52.87 | 52.22 | 86.24 (1.40) | 70.79 (0.28) | 106.61 (1.56) | 95.28 (1.09) | 43.33 (0.18) | 48.66 (0.49) | 43.92 (0.22) | **41.56 (0.30)** |
| | MAE | 73.2 | 102.91 | 104.94 | 146.24 | 179.33 | 136.7 | 115.36 | 173.18 (1.62) | 208.71 (0.34) | 283.65 (5.44) | 237.80 (2.72) | 89.42 (0.47) | 110.15 (1.27) | **70.39 (0.70)** | **65.88 (0.90)** |
| | RMSE | 149.72 | **118.53** | 121.45 | 160.45 | 322.39 | 303.15 | 249.15 | 361.28 (2.54) | 466.74 (0.55) | 473.18 (3.14) | 435.99 (2.37) | 186.83 (0.99) | 250.69 (3.42) | 124.26 (1.80) | **119.07 (1.70)** |
| | MASE | 1.95 | 2.69 | 2.57 | 3.44 | 4.77 | 3.64 | 3.07 | 4.61 (0.04) | 5.55 (0.01) | 7.54 (0.14) | 6.32 (0.07) | 2.38 (0.01) | 2.93 (0.03) | **1.87 (0.02)** | **1.75 (0.02)** |
| | RMSSE | 1.44 | 1.84 | 1.78 | 2.25 | 3.1 | 2.91 | 2.39 | 3.47 (0.02) | 4.48 (0.01) | 4.54 (0.03) | 4.19 (0.02) | 1.79 (0.01) | 2.41 (0.03) | **1.19 (0.02)** | **1.14 (0.02)** |
| | Pinball-80 | 42.27 | 56.45 | 74.36 | 77.85 | 143.22 | 106.23 | 76.91 | 113.00 (1.67) | 131.98 (0.41) | 161.14 (1.60) | 135.38 (1.07) | 40.23 (0.24) | 54.30 (1.21) | **29.62 (0.34)** | **29.33 (0.34)** |
| | Pinball-95 | 45.1 | 58.94 | 85.3 | 80.21 | 170 | 125.17 | 86.53 | 105.51 (2.92) | 112.02 (1.15) | 85.77 (0.85) | 82.06 (1.17) | 33.95 (0.41) | 38.19 (1.37) | **17.44 (0.38)** | **18.96 (0.20)** |
| | ρ-0.5 Risk | 0.26 | 0.36 | 0.37 | 0.51 | 0.63 | 0.48 | 0.4 | 0.60 (0.01) | 0.73 (0.00) | 1.00 (0.02) | 0.83 (0.01) | 0.31 (0.00) | 0.38 (0.00) | **0.25 (0.00)** | **0.23 (0.00)** |
| | ρ-0.9 Risk | 0.31 | 0.41 | 0.57 | 0.55 | 1.12 | 0.83 | 0.58 | 0.78 (0.02) | 0.88 (0.01) | 0.85 (0.01) | 0.76 (0.01) | **0.25 (0.00)** | 0.32 (0.01) | **0.16 (0.00)** | **0.16 (0.00)** |
| | CRPS | 73.2 | 102.91 | 104.94 | 146.24 | 179.34 | 136.7 | 115.36 | 151.28 (1.97) | 179.31 (0.44) | 245.18 (1.61) | 193.91 (1.30) | 81.79 (0.48) | 84.32 (1.07) | **55.76 (0.47)** | **53.17 (0.50)** |
| | Winkler Score | - | - | - | - | - | - | - | 4138.05 (162.95) | 4212.21 (76.86) | 3596.55 (47.90) | 3088.02 (51.80) | 2636.71 (36.95) | 1504.60 (47.28) | **957.69 (18.21)** | 1039.84 (16.75) |
| Belgium COVID-19 | SMAPE | 101.51 | 90.45 | 119.99 | 127.66 | 96.48 | 163.31 | 121.48 | 82.84 (2.14) | 77.40 (0.27) | 151.75 (16.27) | 122.36 (0.69) | **67.17 (0.06)** | 69.29 (0.47) | 69.72 (0.23) | 81.37 (0.37) |
| | MAE | 242.15 | 165.5 | 195.33 | 457.57 | 190.85 | 140.64 | 374.97 | 134.57 (9.08) | 102.53 (0.45) | 283.38 (68.24) | 351.98 (5.25) | **78.10 (0.15)** | **90.09 (0.88)** | 90.13 (0.48) | 97.53 (0.50) |
| | RMSE | 304.01 | 184.3 | 232.09 | 473.9 | 310.3 | 208.02 | 430.97 | 174.57 (10.21) | 156.61 (0.66) | 355.68 (73.93) | 385.31 (5.34) | **122.58 (0.12)** | 124.67 (1.14) | **120.75 (0.59)** | 144.28 (0.48) |
| | MASE | 1.18 | 0.78 | 0.95 | 2.13 | 0.93 | 0.68 | 1.82 | 0.65 (0.04) | 0.50 (0.00) | 1.38 (0.33) | 1.71 (0.03) | **0.38 (0.00)** | **0.44 (0.00)** | **0.44 (0.00)** | 0.47 (0.00) |
| | RMSSE | 0.6 | 0.39 | 0.5 | 0.98 | 0.61 | 0.41 | 0.85 | 0.34 (0.02) | 0.31 (0.00) | 0.70 (0.15) | 0.76 (0.01) | **0.24 (0.00)** | **0.24 (0.00)** | **0.24 (0.00)** | 0.28 (0.00) |
| | Pinball-80 | 51.44 | 38.71 | 90.45 | 91.94 | 50.08 | 112.27 | 75.92 | 86.29 (10.85) | 45.71 (0.31) | 347.10 (30.83) | 175.37 (1.30) | 38.19 (0.08) | **31.39 (0.36)** | **30.76 (0.10)** | 40.45 (0.84) |
| | Pinball-95 | 16.63 | 16.69 | 86.84 | 23.51 | 27.41 | 133.25 | 20.13 | 45.78 (2.93) | 23.00 (0.46) | 173.37 (9.59) | 71.45 (0.43) | 36.51 (0.17) | **12.30 (0.36)** | 16.62 (0.16) | **12.68 (0.69)** |
| | ρ-0.5 Risk | 1.6 | 1.09 | 1.29 | 3.03 | 1.26 | 0.93 | 2.48 | 0.90 (0.06) | 0.68 (0.00) | 1.81 (0.42) | 2.37 (0.03) | **0.52 (0.00)** | **0.60 (0.01)** | **0.60 (0.00)** | 0.65 (0.00) |
| | ρ-0.9 Risk | 0.37 | 0.32 | 1.16 | 0.61 | 0.46 | 1.67 | 0.51 | 0.91 (0.08) | 0.44 (0.00) | 3.56 (0.22) | 1.55 (0.01) | **0.49 (0.00)** | **0.27 (0.00)** | 0.29 (0.00) | 0.30 (0.02) |
| | CRPS | 242.15 | 165.5 | 195.33 | 457.57 | 190.85 | 140.64 | 374.97 | 77.24 (0.33) | 106.24 (23.61) | 546.12 (23.61) | 243.25 (2.53) | 75.75 (0.17) | **65.78 (0.62)** | 72.34 (0.47) | 71.36 (0.69) |
| | Winkler Score | - | - | - | - | - | - | - | 1324.40 (60.61) | 1106.11 (23.78) | 8465.70 (313.41) | 2400.48 (13.87) | 2830.95 (12.33) | **713.64 (23.65)** | 1361.05 (21.84) | **638.55 (29.30)** |
| Colombia Dengue | SMAPE | **40.68** | 61.2 | 68.89 | 78.16 | 44.21 | 44.17 | 77.49 | 59.61 (0.88) | 42.11 (0.55) | 107.44 (2.65) | 62.59 (0.71) | 55.56 (0.39) | **41.78 (1.15)** | 42.06 (0.69) | 47.54 (0.84) |
| | MAE | 18.07 | 27.31 | 22.97 | 44.02 | 19.15 | **16.59** | 39.27 | 23.53 (0.20) | 23.01 (0.20) | 42.81 (0.42) | 25.02 (0.23) | 18.04 (0.10) | **18.02 (0.28)** | 18.48 (0.22) | 20.17 (0.36) |
| | RMSE | 48.76 | **29.87** | **26.35** | 45.97 | 47.17 | 44.46 | 78.05 | 54.64 (0.63) | 68.18 (0.37) | 85.45 (0.71) | 65.34 (0.46) | 34.19 (0.21) | 51.89 (0.48) | 49.43 (0.35) | 55.49 (1.09) |
| | MASE | 2.45 | 3.3 | 2.92 | 4.59 | 2.6 | **2.25** | 5.33 | 3.19 (0.03) | 3.12 (0.03) | 5.81 (0.06) | 3.39 (0.03) | 2.46 (0.01) | **2.45 (0.04)** | 2.50 (0.03) | 2.74 (0.05) |
| | RMSSE | 3.2 | **2.13** | 2.92 | 2.91 | 3.1 | **2.92** | 5.12 | 3.59 (0.04) | 4.47 (0.02) | 5.61 (0.05) | 4.29 (0.03) | 2.24 (0.01) | 3.41 (0.03) | 3.24 (0.02) | 3.64 (0.07) |
| | Pinball-80 | 12.65 | 13.35 | 14.25 | 21.73 | 11.59 | 11.74 | 21.42 | 11.76 (0.20) | 12.01 (0.13) | 22.89 (0.22) | 14.46 (0.16) | **9.45 (0.08)** | 10.82 (0.18) | 11.15 (0.11) | 11.78 (0.14) |
| | Pinball-95 | 14.45 | 13.2 | 15.64 | 21.59 | 12.59 | 13.46 | 22.31 | **8.22 (0.25)** | 8.24 (0.30) | 14.41 (0.20) | 11.86 (0.22) | 8.51 (0.12) | **8.20 (0.30)** | 9.93 (0.13) | 11.15 (0.17) |
| | ρ-0.5 Risk | **0.38** | 0.57 | 0.48 | 0.91 | 0.4 | **0.34** | 0.81 | 0.49 (0.00) | 0.47 (0.00) | 0.88 (0.01) | 0.51 (0.00) | **0.38 (0.01)** | 0.39 (0.01) | **0.38 (0.00)** | 0.42 (0.01) |
| | ρ-0.9 Risk | 0.57 | 0.55 | 0.63 | 0.9 | 0.51 | 0.53 | 0.91 | 0.41 (0.01) | 0.42 (0.01) | 0.76 (0.01) | 0.54 (0.01) | **0.36 (0.00)** | 0.41 (0.01) | 0.44 (0.00) | 0.48 (0.01) |
| | CRPS | 18.07 | 27.31 | 22.97 | 44.02 | 19.15 | 16.59 | 39.27 | 18.49 (0.19) | 18.91 (0.12) | 38.18 (0.29) | 20.65 (0.18) | 16.12 (0.11) | **15.62 (0.20)** | **15.88 (0.12)** | 17.22 (0.18) |
| | Winkler Score | - | - | - | - | - | - | - | **308.33 (11.06)** | 332.72 (18.32) | | 463.00 (9.97) | 491.81 (6.12) | 346.43 (12.66) | 412.48 (5.51) | 459.42 (6.81) |
| Hungary Chickenpox | SMAPE | 86.76 | 92.3 | 98.06 | 104.74 | 136.13 | 107.16 | 103.33 | 93.02 (0.63) | 84.54 (0.48) | 99.63 (2.94) | 107.29 (0.88) | 96.69 (0.46) | 83.83 (1.40) | **83.40 (0.55)** | **82.74 (0.74)** |
| | MAE | **12.97** | 16.25 | 20.15 | 24.95 | 35.23 | 32.08 | 31.19 | 17.63 (0.36) | 13.48 (0.08) | 16.60 (0.30) | 26.07 (0.37) | 18.97 (0.06) | 13.73 (0.13) | 13.90 (0.10) | **13.39 (0.09)** |
| | RMSE | **23.03** | **20.89** | 24.19 | 28.11 | 35.23 | 32.08 | 31.19 | 25.99 (0.42) | 25.46 (0.19) | 31.13 (0.41) | 34.76 (0.36) | 26.78 (0.05) | 23.29 (0.11) | 23.43 (0.12) | 23.74 (0.07) |
| | MASE | **0.67** | 0.82 | 1.02 | 1.23 | 1.08 | 0.81 | 1.23 | 0.91 (0.02) | 0.70 (0.00) | 0.86 (0.02) | 1.35 (0.02) | 0.98 (0.00) | 0.71 (0.01) | 0.72 (0.01) | **0.69 (0.00)** |
| | RMSSE | 0.73 | **0.67** | 0.78 | 0.9 | 1.1 | 1 | 0.97 | 0.81 (0.01) | 0.79 (0.01) | 0.97 (0.01) | 1.09 (0.01) | 0.84 (0.00) | **0.72 (0.00)** | **0.72 (0.00)** | 0.74 (0.00) |
| | Pinball-80 | 6.78 | 6.62 | 6.33 | 7.34 | 12.45 | 12.18 | 6.8 | 7.14 (0.12) | 6.29 (0.06) | 9.43 (0.20) | 12.53 (0.13) | 6.68 (0.02) | **5.70 (0.04)** | **5.93 (0.03)** | 6.13 (0.04) |
| | Pinball-95 | 6.92 | 5.87 | 4.45 | 4.77 | 13.44 | 14.34 | 4.24 | **2.93 (0.10)** | 3.66 (0.04) | 5.03 (0.10) | 5.67 (0.08) | 4.59 (0.03) | **3.32 (0.05)** | 3.90 (0.08) | 4.34 (0.07) |
| | ρ-0.5 Risk | **0.66** | 0.83 | 1.03 | 1.28 | 1.07 | 0.81 | 1.22 | 0.91 (0.02) | **0.69 (0.00)** | 0.84 (0.02) | 1.34 (0.02) | 0.97 (0.00) | 0.71 (0.01) | 0.71 (0.01) | **0.69 (0.00)** |
| | ρ-0.9 Risk | 0.69 | 0.63 | 0.52 | 0.58 | 1.34 | 1.4 | 0.52 | 0.49 (0.01) | **0.48 (0.01)** | 0.71 (0.01) | 0.88 (0.01) | 0.50 (0.01) | **0.44 (0.00)** | 0.50 (0.01) | 0.53 (0.01) |
| | CRPS | 12.97 | 16.25 | 20.15 | 24.95 | 20.9 | 15.71 | 23.85 | 12.22 (0.24) | **10.05 (0.06)** | 15.23 (0.16) | 18.67 (0.22) | 17.34 (0.07) | **10.72 (0.09)** | 11.53 (0.06) | 11.27 (0.06) |
| | Winkler Score | - | - | - | - | - | - | - | **94.17 (4.15)** | 134.88 (6.71) | 220.04 (4.34) | 182.18 (3.51) | 555.90 (4.28) | 184.34 (3.81) | 259.69 (4.83) | 275.69 (4.62) |

Table 14: study for pre-ANM vs. post-ANM and ES vs. MSE loss: results for short-term horizon. The mean and standard deviations for each metric are computed over 50 independent evaluation runs. The **best** results are highlighted.

| Dataset | Metric | **MVEN** | MVEN-Post | MVEN-MSE | **GCEN** | GCEN-Post | GCEN-MSE | **STEN** | STEN-Post | STEN-MSE |
|---|---|---|---|---|---|---|---|---|---|---|
| Japan TB | SMAPE | 23.99 (0.13) | 26.99 (0.36) | 27.45 (0.08) | **21.30 (0.14)** | 24.90 (0.41) | 22.83 (0.09) | 23.10 (0.14) | 23.50 (0.38) | 22.40 (0.08) |
| | MAE | 6.82 (0.05) | 8.04 (0.22) | 8.59 (0.03) | **5.29 (0.05)** | 7.53 (0.17) | 6.47 (0.02) | 6.46 (0.07) | 6.61 (0.15) | 6.17 (0.02) |
| | RMSE | 12.39 (0.12) | 14.23 (0.58) | 15.28 (0.06) | **9.39 (0.12)** | 13.91 (0.42) | 11.83 (0.04) | 11.78 (0.16) | 12.03 (0.37) | 11.31 (0.04) |
| | MASE | 0.79 (0.01) | 0.94 (0.03) | 1.00 (0.00) | **0.62 (0.01)** | 0.88 (0.02) | 0.75 (0.00) | 0.75 (0.01) | 0.77 (0.02) | 0.72 (0.00) |
| | RMSSE | 0.93 (0.01) | 1.07 (0.04) | 1.14 (0.00) | **0.70 (0.01)** | 1.04 (0.03) | 0.89 (0.00) | 0.88 (0.01) | 0.90 (0.03) | 0.85 (0.00) |
| | Pinball-80 | 1.99 (0.02) | 4.15 (0.04) | 2.13 (0.01) | **1.90 (0.00)** | 3.99 (0.05) | 1.91 (0.01) | 2.05 (0.02) | 3.70 (0.05) | 1.91 (0.01) |
| | Rho-0.5 | 0.21 (0.00) | 0.25 (0.01) | 0.26 (0.00) | **0.16 (0.00)** | 0.23 (0.01) | 0.20 (0.00) | 0.20 (0.00) | 0.21 (0.00) | 0.19 (0.00) |
| | Rho-0.9 | **0.08 (0.00)** | 0.17 (0.00) | **0.08 (0.00)** | **0.08 (0.00)** | 0.16 (0.00) | **0.08 (0.00)** | **0.08 (0.00)** | 0.15 (0.00) | **0.08 (0.00)** |
| | CRPS | 5.97 (0.03) | 6.08 (0.08) | 8.24 (0.02) | **4.23 (0.03)** | 5.90 (0.07) | 6.11 (0.01) | 5.14 (0.04) | 5.53 (0.06) | 5.72 (0.02) |
| China TB | SMAPE | 15.88 (0.10) | 16.27 (0.20) | 16.64 (0.10) | **12.59 (0.08)** | 13.42 (0.20) | 13.19 (0.09) | 14.73 (0.12) | 16.76 (0.20) | 17.63 (0.08) |
| | MAE | 391.07 (3.10) | 393.65 (5.67) | 407.38 (3.05) | **295.58 (2.07)** | 342.68 (6.19) | 318.13 (2.66) | 342.05 (2.57) | 413.91 (5.20) | 430.51 (1.63) |
| | RMSE | 807.22 (5.84) | 832.30 (14.57) | 818.06 (5.23) | **677.79 (2.65)** | 797.65 (15.09) | 725.23 (4.00) | 794.24 (2.19) | 833.39 (7.15) | 820.57 (1.47) |
| | MASE | 1.56 (0.01) | 1.57 (0.02) | 1.63 (0.01) | **1.18 (0.01)** | 1.37 (0.02) | 1.27 (0.01) | 1.37 (0.01) | 1.66 (0.02) | 1.72 (0.01) |
| | RMSSE | 0.94 (0.01) | 0.97 (0.02) | 0.95 (0.01) | **0.79 (0.00)** | 0.93 (0.02) | 0.85 (0.00) | 0.93 (0.00) | 0.97 (0.01) | 0.96 (0.00) |
| | Pinball-80 | 172.31 (1.02) | 181.69 (1.77) | 171.61 (0.93) | **144.95 (0.44)** | 170.62 (2.05) | 152.68 (0.44) | 167.67 (0.77) | 191.83 (1.79) | 172.07 (0.79) |
| | Rho-0.5 | 0.19 (0.00) | 0.19 (0.00) | 0.20 (0.00) | **0.14 (0.00)** | 0.17 (0.00) | 0.15 (0.00) | 0.17 (0.00) | 0.20 (0.00) | 0.21 (0.00) |
| | Rho-0.9 | 0.14 (0.00) | 0.14 (0.00) | 0.14 (0.00) | **0.13 (0.00)** | 0.14 (0.00) | 0.14 (0.00) | 0.15 (0.00) | 0.15 (0.00) | 0.15 (0.00) |
| | CRPS | 335.04 (2.21) | 297.68 (3.03) | 362.82 (2.06) | **259.17 (1.28)** | 263.84 (3.35) | 288.31 (1.81) | 303.31 (1.22) | 332.01 (2.62) | 398.51 (1.06) |
| USA ILI | SMAPE | 37.17 (0.45) | 63.95 (1.03) | 38.49 (1.25) | **31.15 (0.24)** | 34.71 (0.71) | 34.15 (0.15) | 32.49 (0.29) | 41.17 (0.68) | 39.84 (1.04) |
| | MAE | 119.33 (2.10) | 199.07 (1.87) | 115.83 (5.28) | **78.06 (1.15)** | 100.54 (2.19) | 93.53 (0.74) | 90.69 (1.34) | 99.64 (1.97) | 97.67 (5.27) |
| | RMSE | 245.61 (6.20) | 398.35 (6.13) | 238.97 (15.54) | **147.83 (2.86)** | 215.18 (6.72) | 175.01 (1.68) | 183.59 (4.34) | 169.53 (5.32) | 187.85 (14.23) |
| | MASE | 3.19 (0.06) | 5.32 (0.05) | 3.10 (0.14) | **2.09 (0.03)** | 2.69 (0.06) | 2.50 (0.02) | 2.43 (0.04) | 2.66 (0.05) | 2.61 (0.14) |
| | RMSSE | 2.38 (0.06) | 3.87 (0.06) | 2.32 (0.15) | **1.44 (0.03)** | 2.09 (0.07) | 1.70 (0.02) | 1.78 (0.04) | 1.65 (0.05) | 1.82 (0.14) |
| | Pinball-80 | 52.45 (1.91) | 83.62 (1.79) | 57.72 (4.27) | **38.26 (0.71)** | 42.04 (1.09) | 39.65 (0.43) | 44.76 (1.37) | 43.85 (0.67) | 44.99 (2.93) |
| | Rho-0.5 | 0.31 (0.01) | 0.51 (0.01) | 0.29 (0.01) | **0.20 (0.00)** | 0.26 (0.01) | 0.24 (0.00) | 0.24 (0.00) | 0.26 (0.01) | 0.25 (0.01) |
| | Rho-0.9 | 0.21 (0.01) | 0.32 (0.01) | 0.26 (0.02) | 0.17 (0.00) | **0.15 (0.00)** | 0.17 (0.00) | 0.19 (0.01) | 0.16 (0.00) | 0.20 (0.01) |
| | CRPS | 87.82 (1.46) | 148.23 (1.31) | 87.75 (4.10) | **61.95 (0.61)** | 78.25 (1.17) | 77.85 (0.76) | 69.86 (1.00) | 76.95 (0.91) | 76.12 (3.79) |
| Belgium COVID-19 | SMAPE | 61.60 (0.58) | 73.64 (1.24) | 74.62 (0.30) | **46.99 (1.08)** | 67.24 (1.21) | 60.16 (0.44) | 68.02 (0.20) | 75.95 (1.12) | 72.92 (0.16) |
| | MAE | 86.08 (1.13) | 121.55 (2.25) | 112.78 (0.46) | **55.13 (0.86)** | 115.13 (2.29) | 87.14 (0.82) | 103.29 (0.54) | 114.26 (2.16) | 126.27 (0.76) |
| | RMSE | 114.83 (1.56) | 172.24 (3.16) | 148.94 (0.69) | **82.02 (1.44)** | 162.41 (3.23) | 120.23 (1.08) | 140.29 (0.68) | 155.64 (2.97) | 170.17 (1.08) |
| | MASE | 0.44 (0.01) | 0.62 (0.01) | 0.58 (0.00) | **0.28 (0.00)** | 0.59 (0.01) | 0.44 (0.00) | 0.53 (0.00) | 0.58 (0.01) | 0.64 (0.00) |
| | RMSSE | 0.22 (0.01) | 0.35 (0.01) | 0.30 (0.00) | **0.17 (0.00)** | 0.33 (0.01) | 0.25 (0.00) | 0.29 (0.00) | 0.32 (0.01) | 0.35 (0.00) |
| | Pinball-80 | 30.11 (0.71) | 66.00 (0.50) | 43.73 (0.66) | **22.20 (0.6)** | 67.30 (0.49) | 30.72 (0.37) | 38.30 (0.24) | 53.67 (0.61) | 43.04 (0.16) |
| | Rho-0.5 | 0.49 (0.01) | 0.70 (0.01) | 0.64 (0.00) | **0.31 (0.00)** | 0.66 (0.01) | 0.49 (0.00) | 0.58 (0.00) | 0.65 (0.01) | 0.71 (0.00) |
| | Rho-0.9 | 0.22 (0.01) | 0.46 (0.00) | 0.34 (0.01) | **0.18 (0.00)** | 0.47 (0.00) | 0.24 (0.00) | 0.32 (0.00) | 0.38 (0.00) | 0.35 (0.00) |
| | CRPS | 60.71 (0.79) | 98.31 (0.84) | 87.79 (0.47) | **42.38 (0.59)** | 95.50 (0.84) | 66.91 (0.62) | 81.44 (0.41) | 92.49 (0.67) | 107.24 (0.66) |
| Colombia Dengue | SMAPE | 43.32 (1.46) | 39.20 (2.11) | 39.05 (2.88) | **29.24 (1.17)** | 39.75 (2.66) | 31.67 (1.60) | 36.20 (0.99) | 38.14 (2.08) | 31.58 (2.05) |
| | MAE | 12.22 (0.33) | 13.20 (0.53) | 12.49 (0.60) | 8.31 (0.15) | 10.40 (0.65) | 9.20 (0.13) | **8.24 (0.20)** | 13.06 (0.66) | 9.11 (0.27) |
| | RMSE | 23.28 (0.29) | 24.96 (1.05) | 24.60 (0.97) | **13.14 (0.21)** | 16.58 (1.02) | 15.08 (0.31) | 13.22 (0.32) | 22.83 (0.97) | 15.24 (0.52) |
| | MASE | 1.66 (0.04) | 1.79 (0.07) | 1.69 (0.08) | **1.12 (0.02)** | 1.41 (0.09) | 1.25 (0.02) | 1.12 (0.03) | 1.77 (0.09) | 1.24 (0.04) |
| | RMSSE | 1.53 (0.02) | 1.64 (0.07) | 1.61 (0.06) | **0.86 (0.01)** | 1.09 (0.07) | 0.99 (0.02) | 0.86 (0.02) | 1.50 (0.06) | 1.00 (0.03) |
| | Pinball-80 | 6.22 (0.17) | 9.74 (0.22) | 6.65 (0.37) | 3.73 (0.08) | 8.67 (0.17) | 4.26 (0.10) | **3.72 (0.09)** | 9.16 (0.18) | 4.00 (0.09) |
| | Rho-0.5 | 0.24 (0.01) | 0.26 (0.01) | 0.24 (0.01) | **0.16 (0.00)** | 0.20 (0.01) | 0.18 (0.00) | **0.16 (0.00)** | 0.26 (0.01) | 0.18 (0.01) |
| | Rho-0.9 | 0.22 (0.00) | 0.26 (0.01) | 0.24 (0.01) | 0.12 (0.00) | 0.25 (0.01) | 0.13 (0.00) | **0.11 (0.00)** | 0.26 (0.01) | **0.11 (0.00)** |
| | CRPS | 10.16 (0.13) | 14.79 (0.22) | 10.34 (0.39) | 6.49 (0.08) | 13.40 (0.18) | 7.29 (0.11) | **6.42 (0.11)** | 14.43 (0.20) | 7.19 (0.13) |
| Hungary Chickenpox | SMAPE | **76.78 (0.53)** | 99.65 (5.26) | 78.11 (0.31) | 80.24 (0.48) | 98.31 (6.12) | 77.39 (0.18) | 78.19 (0.24) | 95.28 (4.43) | 77.46 (0.19) |
| | MAE | 20.74 (0.16) | 23.95 (1.54) | 22.29 (0.15) | **20.54 (0.10)** | 24.11 (1.40) | 21.50 (0.07) | 21.74 (0.08) | 24.95 (1.37) | 20.86 (0.05) |
| | RMSE | 35.27 (0.20) | 38.10 (2.80) | 35.45 (0.11) | 36.19 (0.09) | 38.73 (2.68) | **34.89 (0.06)** | 35.52 (0.10) | 40.37 (2.32) | 34.90 (0.05) |
| | MASE | 1.08 (0.01) | 1.25 (0.08) | 1.16 (0.01) | **1.07 (0.01)** | 1.26 (0.07) | 1.12 (0.00) | 1.13 (0.00) | 1.30 (0.07) | 1.09 (0.00) |
| | RMSSE | 1.11 (0.01) | 1.20 (0.09) | 1.11 (0.00) | 1.14 (0.00) | 1.22 (0.08) | **1.10 (0.00)** | 1.12 (0.00) | 1.27 (0.07) | **1.10 (0.00)** |
| | Pinball-80 | **9.40 (0.09)** | 10.26 (0.60) | 9.46 (0.06) | 10.52 (0.06) | 10.43 (0.67) | 9.94 (0.04) | 10.60 (0.05) | 10.99 (0.52) | 10.73 (0.03) |
| | Rho-0.5 | 0.67 (0.01) | 0.77 (0.05) | 0.72 (0.00) | **0.66 (0.00)** | 0.77 (0.05) | 0.69 (0.00) | 0.70 (0.00) | 0.81 (0.05) | 0.67 (0.00) |
| | Rho-0.9 | 0.49 (0.01) | **0.47 (0.04)** | 0.51 (0.01) | 0.61 (0.01) | 0.48 (0.04) | 0.58 (0.00) | 0.63 (0.00) | 0.51 (0.04) | 0.67 (0.00) |
| | CRPS | **16.27 (0.12)** | 18.22 (0.87) | 18.73 (0.11) | 17.32 (0.09) | 18.28 (0.77) | 19.44 (0.06) | 19.13 (0.05) | 19.49 (0.55) | 19.60 (0.03) |

Table 15: Ablation study (similar to Table 14): results for medium-term horizon. The mean and standard deviations for each metric are computed over 50 independent evaluation runs. The **best** results are highlighted.

| Dataset | Metric | **MVEN** | MVEN-Post | MVEN-MSE | **GCEN** | GCEN-Post | GCEN-MSE | **STEN** | STEN-Post | STEN-MSE |
|---|---|---|---|---|---|---|---|---|---|---|
| Japan TB | SMAPE | 23.79 (0.11) | 24.65 (0.27) | 25.58 (0.07) | **22.41 (0.11)** | 23.60 (0.33) | 22.69 (0.05) | 22.58 (0.09) | 23.43 (0.32) | 22.50 (0.06) |
| | MAE | 6.12 (0.03) | 6.27 (0.10) | 7.13 (0.03) | **5.42 (0.02)** | 5.96 (0.10) | 5.71 (0.01) | 5.58 (0.02) | 5.81 (0.10) | 5.43 (0.01) |
| | RMSE | 10.20 (0.06) | 10.53 (0.26) | 12.16 (0.06) | **8.99 (0.04)** | 9.92 (0.27) | 9.61 (0.02) | 9.29 (0.06) | 9.67 (0.28) | 9.04 (0.02) |
| | MASE | 0.71 (0.00) | 0.73 (0.01) | 0.82 (0.00) | **0.63 (0.00)** | 0.69 (0.01) | 0.66 (0.00) | 0.64 (0.00) | 0.67 (0.01) | 0.63 (0.00) |
| | RMSSE | 0.76 (0.00) | 0.78 (0.02) | 0.90 (0.00) | **0.67 (0.00)** | 0.74 (0.02) | 0.71 (0.00) | 0.69 (0.00) | 0.72 (0.02) | 0.67 (0.00) |
| | Pinball-80 | **1.90 (0.01)** | 3.43 (0.03) | 1.95 (0.01) | 1.97 (0.01) | 3.35 (0.03) | 1.96 (0.01) | 1.93 (0.01) | 3.18 (0.03) | 2.03 (0.01) |
| | Rho-0.5 | 0.19 (0.00) | 0.20 (0.00) | 0.22 (0.00) | **0.17 (0.00)** | 0.19 (0.00) | 0.18 (0.00) | **0.17 (0.00)** | 0.18 (0.00) | **0.17 (0.00)** |
| | Rho-0.9 | **0.08 (0.00)** | 0.15 (0.00) | **0.08 (0.00)** | 0.09 (0.00) | 0.14 (0.00) | 0.10 (0.00) | **0.08 (0.00)** | 0.14 (0.00) | 0.10 (0.00) |
| | CRPS | 5.10 (0.02) | 5.34 (0.04) | 6.58 (0.02) | **4.44 (0.01)** | 5.19 (0.04) | 5.26 (0.01) | 4.46 (0.02) | 5.13 (0.03) | 4.92 (0.01) |
| China TB | SMAPE | 15.51 (0.11) | 18.11 (0.21) | 15.65 (0.30) | 16.72 (0.07) | 17.56 (0.43) | 17.36 (0.04) | 17.04 (0.07) | 18.74 (0.55) | 16.43 (0.01) |
| | MAE | **359.42 (1.98)** | 440.37 (4.68) | 362.04 (6.09) | 415.46 (1.95) | 462.89 (11.57) | 415.74 (0.96) | 404.03 (1.84) | 449.99 (16.18) | 386.11 (0.14) |
| | RMSE | 744.89 (4.21) | 890.74 (14.68) | **725.63 (11.77)** | 932.84 (5.01) | 1139.14 (43.17) | 917.58 (2.46) | 812.76 (5.40) | 963.19 (61.73) | 692.78 (0.17) |
| | MASE | **1.53 (0.01)** | 1.87 (0.02) | 1.54 (0.03) | 1.76 (0.01) | 1.97 (0.05) | 1.77 (0.00) | 1.72 (0.01) | 1.91 (0.07) | 1.64 (0.00) |
| | RMSSE | 0.81 (0.02) | 1.00 (0.02) | 0.82 (0.01) | 1.05 (0.01) | 1.28 (0.05) | 1.03 (0.00) | 0.91 (0.01) | 1.08 (0.07) | **0.78 (0.00)** |
| | Pinball-80 | **157.36 (1.26)** | 179.58 (1.47) | 161.25 (3.17) | 167.72 (0.68) | 190.87 (4.77) | 184.74 (0.59) | 162.49 (1.11) | 186.12 (5.58) | 163.45 (0.06) |
| | Rho-0.5 | **0.15 (0.01)** | 0.20 (0.00) | 0.16 (0.00) | 0.19 (0.00) | 0.21 (0.01) | 0.19 (0.00) | 0.18 (0.00) | 0.20 (0.01) | 0.17 (0.00) |
| | Rho-0.9 | 0.12 (0.00) | **0.12 (0.00)** | 0.13 (0.00) | 0.13 (0.00) | 0.13 (0.00) | 0.15 (0.00) | **0.12 (0.00)** | 0.13 (0.00) | 0.14 (0.00) |
| | CRPS | **304.96 (1.82)** | 316.45 (2.34) | 322.87 (5.01) | 361.44 (1.71) | 367.32 (8.23) | 384.62 (1.05) | 347.77 (2.47) | 349.23 (10.50) | 379.34 (0.12) |
| USA ILI | SMAPE | **38.47 (0.41)** | 50.59 (1.65) | 42.40 (0.97) | 39.50 (0.27) | 47.46 (0.59) | 48.05 (0.13) | 38.90 (0.28) | 52.14 (0.50) | 46.63 (0.13) |
| | MAE | 97.08 (1.78) | 106.72 (3.96) | 96.82 (3.97) | 75.29 (0.98) | 91.78 (1.65) | 95.85 (0.89) | **72.27 (1.62)** | 115.75 (1.49) | 89.67 (0.71) |
| | RMSE | 222.87 (4.85) | 217.65 (10.30) | 211.78 (10.35) | 135.37 (2.96) | 159.11 (3.26) | 164.64 (2.49) | **131.13 (4.77)** | 206.51 (3.96) | 152.45 (2.39) |
| | MASE | 2.59 (0.05) | 2.85 (0.11) | 2.59 (0.11) | 2.01 (0.03) | 2.45 (0.04) | 2.56 (0.02) | **1.93 (0.04)** | 3.09 (0.04) | 2.40 (0.02) |
| | RMSSE | 2.15 (0.05) | 2.10 (0.10) | 2.04 (0.10) | 1.31 (0.03) | 1.54 (0.03) | 1.59 (0.02) | **1.27 (0.05)** | 1.99 (0.04) | 1.47 (0.02) |
| | Pinball-80 | 47.02 (0.78) | 47.57 (1.35) | 50.09 (1.62) | 31.62 (0.44) | 45.87 (0.45) | 33.12 (0.23) | **30.64 (0.35)** | 53.54 (0.44) | 33.17 (0.23) |
| | Rho-0.5 | 0.31 (0.01) | 0.33 (0.01) | 0.30 (0.01) | 0.24 (0.00) | 0.29 (0.01) | 0.30 (0.00) | **0.23 (0.01)** | 0.37 (0.00) | 0.28 (0.00) |
| | Rho-0.9 | 0.26 (0.01) | 0.23 (0.01) | 0.30 (0.01) | **0.15 (0.00)** | 0.21 (0.00) | **0.15 (0.00)** | **0.15 (0.00)** | 0.23 (0.00) | 0.16 (0.00) |
| | CRPS | 75.62 (0.93) | 83.50 (1.75) | 79.21 (2.13) | 56.67 (0.59) | 71.74 (0.58) | 77.49 (0.72) | **55.72 (0.75)** | 84.45 (0.68) | 74.85 (0.53) |
| Belgium COVID-19 | SMAPE | 67.43 (0.67) | 79.80 (1.55) | 73.71 (0.18) | **57.10 (0.42)** | 84.03 (1.12) | 73.87 (0.08) | 70.99 (0.09) | 76.66 (0.73) | 93.34 (0.03) |
| | MAE | 77.55 (0.47) | 79.43 (1.65) | 113.29 (0.42) | **58.29 (0.42)** | 106.16 (1.62) | 106.12 (0.07) | 87.54 (0.13) | 107.29 (1.70) | 191.34 (0.11) |
| | RMSE | 111.76 (0.62) | 111.97 (2.21) | 157.26 (0.38) | **85.91 (0.55)** | 146.31 (2.16) | 155.93 (0.08) | 121.89 (0.11) | 145.75 (2.35) | 246.30 (0.11) |
| | MASE | 0.34 (0.02) | 0.39 (0.01) | 0.56 (0.00) | **0.29 (0.00)** | 0.53 (0.01) | 0.53 (0.00) | 0.43 (0.00) | 0.53 (0.01) | 0.95 (0.00) |
| | RMSSE | 0.22 (0.00) | 0.22 (0.00) | 0.31 (0.00) | **0.17 (0.00)** | 0.29 (0.00) | 0.31 (0.00) | 0.24 (0.00) | 0.29 (0.00) | 0.49 (0.00) |
| | Pinball-80 | 29.79 (0.28) | 50.99 (0.35) | 31.13 (0.15) | **23.97 (0.18)** | 57.48 (0.36) | 31.84 (0.12) | 37.72 (0.10) | 62.24 (0.36) | 41.97 (0.02) |
| | Rho-0.5 | 0.56 (0.00) | 0.57 (0.01) | 0.82 (0.00) | **0.42 (0.00)** | 0.77 (0.01) | 0.77 (0.00) | 0.63 (0.00) | 0.79 (0.01) | 1.38 (0.00) |
| | Rho-0.9 | 0.32 (0.00) | 0.48 (0.00) | 0.28 (0.00) | **0.27 (0.00)** | 0.53 (0.00) | 0.33 (0.00) | 0.49 (0.00) | 0.56 (0.00) | 0.34 (0.00) |
| | CRPS | 61.99 (0.33) | 81.97 (0.47) | 101.25 (0.31) | **48.13 (0.25)** | 90.53 (0.57) | 102.42 (0.12) | 81.24 (0.15) | 90.54 (0.62) | 189.13 (0.10) |
| Colombia Dengue | SMAPE | 39.14 (1.56) | 49.02 (1.62) | 41.18 (0.64) | **37.01 (1.01)** | 52.80 (1.82) | 37.29 (0.70) | 39.07 (1.32) | 45.08 (1.40) | 50.67 (1.58) |
| | MAE | 16.35 (0.27) | 17.73 (0.40) | 18.77 (0.27) | 15.86 (0.18) | 20.33 (0.39) | 16.65 (0.12) | **15.79 (0.16)** | 19.02 (0.39) | 23.06 (0.74) |
| | RMSE | 43.26 (0.78) | 43.60 (0.56) | 47.60 (0.61) | **43.08 (0.79)** | 51.62 (0.56) | 47.33 (0.61) | 43.56 (0.56) | 50.20 (0.55) | 57.80 (1.80) |
| | MASE | 2.22 (0.04) | 2.41 (0.05) | 2.55 (0.04) | 2.15 (0.02) | 2.76 (0.05) | 2.26 (0.02) | **2.14 (0.02)** | 2.58 (0.05) | 3.13 (0.10) |
| | RMSSE | 2.84 (0.05) | 2.86 (0.04) | 3.12 (0.04) | **2.83 (0.05)** | 3.39 (0.04) | 3.11 (0.04) | 2.86 (0.04) | 3.30 (0.04) | 3.79 (0.12) |
| | Pinball-80 | **7.89 (0.23)** | 11.81 (0.15) | 9.91 (0.20) | 8.33 (0.24) | 12.69 (0.16) | 9.94 (0.14) | 8.33 (0.13) | 13.30 (0.15) | 13.42 (0.34) |
| | Rho-0.5 | 0.32 (0.01) | 0.35 (0.01) | 0.37 (0.01) | **0.31 (0.00)** | 0.40 (0.01) | 0.33 (0.00) | 0.31 (0.00) | 0.37 (0.01) | 0.46 (0.02) |
| | Rho-0.9 | **0.26 (0.01)** | 0.38 (0.00) | 0.35 (0.01) | 0.28 (0.01) | 0.42 (0.01) | 0.38 (0.01) | 0.30 (0.01) | 0.43 (0.01) | 0.52 (0.01) |
| | CRPS | **12.69 (0.20)** | 18.69 (0.15) | 15.07 (0.18) | 12.80 (0.18) | 20.53 (0.15) | 14.60 (0.16) | 12.95 (0.16) | 20.07 (0.16) | 20.30 (0.45) |
| Hungary Chickenpox | SMAPE | 76.01 (0.81) | 78.80 (1.37) | 80.95 (0.26) | **73.58 (0.39)** | 78.02 (1.05) | 79.55 (0.19) | 76.55 (0.34) | 78.71 (1.22) | 76.02 (0.75) |
| | MAE | 16.41 (0.13) | 16.98 (0.28) | 17.62 (0.11) | **15.32 (0.09)** | 16.38 (0.28) | 17.25 (0.06) | 16.11 (0.10) | 16.80 (0.29) | 16.00 (0.21) |
| | RMSE | 27.15 (0.14) | 27.04 (0.48) | 27.50 (0.09) | **26.60 (0.10)** | 27.38 (0.47) | 27.14 (0.05) | 27.31 (0.10) | 27.47 (0.49) | 27.53 (0.20) |
| | MASE | 0.85 (0.01) | 0.88 (0.01) | 0.91 (0.01) | **0.79 (0.00)** | 0.85 (0.01) | 0.90 (0.00) | 0.84 (0.01) | 0.87 (0.02) | 0.83 (0.01) |
| | RMSSE | 0.85 (0.00) | 0.85 (0.02) | 0.86 (0.00) | **0.83 (0.00)** | 0.86 (0.01) | 0.85 (0.00) | 0.86 (0.00) | 0.86 (0.02) | 0.86 (0.01) |
| | Pinball-80 | 6.96 (0.05) | 8.91 (0.12) | 7.14 (0.03) | **6.89 (0.05)** | 8.61 (0.12) | 7.01 (0.02) | 7.22 (0.04) | 8.94 (0.12) | 7.73 (0.10) |
| | Rho-0.5 | 0.69 (0.01) | 0.72 (0.01) | 0.74 (0.00) | **0.65 (0.00)** | 0.69 (0.01) | 0.73 (0.00) | 0.68 (0.00) | 0.71 (0.01) | 0.68 (0.01) |
| | Rho-0.9 | **0.45 (0.00)** | 0.52 (0.01) | 0.49 (0.00) | 0.49 (0.01) | 0.52 (0.01) | 0.51 (0.00) | 0.52 (0.01) | 0.53 (0.01) | 0.60 (0.01) |
| | CRPS | **12.88 (0.08)** | 13.76 (0.14) | 14.86 (0.09) | 12.90 (0.06) | 13.69 (0.13) | 15.66 (0.05) | 13.71 (0.08) | 13.72 (0.13) | 14.46 (0.17) |

Table 16: Ablation study (similar to Table 14): results for long-term horizon. The mean and standard deviations for each metric are computed over 50 independent evaluation runs. The **best** results are highlighted.

| Dataset | Metric | **MVEN** | MVEN-Post | MVEN-MSE | **GCEN** | GCEN-Post | GCEN-MSE | **STEN** | STEN-Post | STEN-MSE |
|---|---|---|---|---|---|---|---|---|---|---|
| Japan TB | SMAPE | 23.24 (0.06) | 24.95 (0.14) | 25.96 (0.03) | **22.38 (0.06)** | 23.69 (0.16) | 24.17 (0.03) | 22.45 (0.04) | 24.31 (0.16) | 24.44 (0.02) |
| | MAE | 6.37 (0.03) | 7.22 (0.07) | 7.90 (0.01) | **5.99 (0.02)** | 6.57 (0.07) | 7.13 (0.01) | 6.00 (0.01) | 7.05 (0.07) | 7.24 (0.00) |
| | RMSE | 11.01 (0.07) | 12.67 (0.21) | 13.88 (0.03) | 10.51 (0.04) | 11.52 (0.20) | 12.54 (0.01) | **10.45 (0.03)** | 12.48 (0.20) | 12.85 (0.01) |
| | MASE | 0.74 (0.00) | 0.82 (0.01) | 0.90 (0.00) | **0.69 (0.01)** | 0.75 (0.01) | 0.81 (0.00) | 0.70 (0.00) | 0.81 (0.01) | 0.83 (0.00) |
| | RMSSE | 0.82 (0.01) | 0.93 (0.02) | 1.02 (0.00) | 0.79 (0.00) | 0.85 (0.01) | 0.92 (0.00) | **0.78 (0.00)** | 0.92 (0.01) | 0.94 (0.00) |
| | Pinball-80 | **1.94 (0.01)** | 3.82 (0.02) | 2.05 (0.00) | 2.05 (0.01) | 3.59 (0.02) | 1.96 (0.00) | 2.11 (0.01) | 3.74 (0.02) | 2.02 (0.00) |
| | Rho-0.5 | 0.19 (0.00) | 0.22 (0.00) | 0.24 (0.00) | **0.18 (0.00)** | 0.20 (0.00) | 0.21 (0.00) | **0.18 (0.00)** | 0.21 (0.00) | 0.22 (0.00) |
| | Rho-0.9 | **0.08 (0.00)** | 0.15 (0.00) | **0.08 (0.00)** | **0.08 (0.01)** | 0.15 (0.00) | **0.08 (0.00)** | 0.09 (0.00) | 0.15 (0.00) | 0.09 (0.00) |
| | CRPS | 5.69 (0.02) | 5.76 (0.03) | 7.68 (0.01) | **5.08 (0.01)** | 5.49 (0.03) | 6.83 (0.00) | 5.36 (0.01) | 5.70 (0.03) | 7.12 (0.00) |
| USA ILI | SMAPE | 48.66 (0.49) | 55.67 (1.25) | 49.72 (0.34) | 43.92 (0.22) | 60.71 (1.45) | 42.60 (0.29) | **41.56 (0.30)** | 56.57 (1.28) | 45.31 (0.16) |
| | MAE | 110.15 (1.27) | 90.34 (3.25) | 107.97 (1.21) | 70.39 (0.70) | 106.79 (3.28) | 73.74 (1.80) | **65.88 (0.90)** | 99.27 (3.36) | 78.53 (0.98) |
| | RMSE | 250.69 (3.42) | 165.19 (9.67) | 235.13 (3.22) | 124.26 (1.80) | 191.65 (10.39) | 135.36 (5.85) | **119.07 (1.70)** | 180.62 (8.83) | 144.35 (3.05) |
| | MASE | 2.93 (0.03) | 2.40 (0.09) | 2.87 (0.03) | 1.87 (0.02) | 2.84 (0.09) | 1.96 (0.05) | **1.75 (0.02)** | 2.64 (0.09) | 2.09 (0.03) |
| | RMSSE | 2.41 (0.03) | 1.59 (0.09) | 2.26 (0.03) | 1.19 (0.02) | 1.84 (0.10) | 1.30 (0.06) | **1.14 (0.02)** | 1.73 (0.08) | 1.39 (0.03) |
| | Pinball-80 | 54.30 (1.21) | 39.65 (1.08) | 55.31 (0.90) | 29.62 (0.34) | 41.21 (1.08) | 29.58 (0.22) | **29.33 (0.34)** | 43.25 (1.14) | 30.75 (0.25) |
| | Rho-0.5 | 0.38 (0.00) | 0.31 (0.01) | 0.38 (0.00) | 0.25 (0.00) | 0.37 (0.01) | 0.26 (0.01) | **0.23 (0.00)** | 0.35 (0.01) | 0.27 (0.00) |
| | Rho-0.9 | 0.32 (0.01) | 0.20 (0.01) | 0.34 (0.01) | **0.16 (0.00)** | 0.19 (0.01) | **0.16 (0.00)** | **0.16 (0.00)** | 0.21 (0.01) | 0.17 (0.00) |
| | CRPS | 84.32 (1.07) | 72.75 (1.42) | 86.75 (0.89) | 55.76 (0.47) | 83.51 (1.63) | 64.30 (1.23) | **53.17 (0.50)** | 79.75 (1.50) | 63.78 (0.61) |
| Belgium COVID-19 | SMAPE | **69.29 (0.47)** | 80.31 (0.81) | 94.84 (0.25) | 69.72 (0.23) | 87.90 (0.67) | 87.21 (0.08) | 81.37 (0.37) | 94.43 (1.24) | 98.89 (0.10) |
| | MAE | **90.09 (0.88)** | 109.56 (1.25) | 181.33 (1.33) | 90.13 (0.48) | 129.43 (1.33) | 148.00 (0.24) | 97.53 (0.50) | 152.30 (2.95) | 199.69 (0.54) |
| | RMSE | 124.67 (1.14) | 161.90 (1.72) | 221.72 (1.71) | **120.75 (0.59)** | 172.09 (1.64) | 185.59 (0.25) | 144.28 (0.48) | 209.57 (4.60) | 245.84 (0.64) |
| | MASE | **0.44 (0.00)** | 0.53 (0.01) | 0.88 (0.01) | **0.44 (0.00)** | 0.63 (0.01) | 0.72 (0.00) | 0.47 (0.00) | 0.74 (0.01) | 0.97 (0.00) |
| | RMSSE | **0.24 (0.00)** | 0.32 (0.00) | 0.44 (0.00) | **0.24 (0.00)** | 0.34 (0.00) | 0.36 (0.00) | 0.28 (0.00) | 0.41 (0.01) | 0.48 (0.00) |
| | Pinball-80 | 31.39 (0.36) | 60.82 (0.32) | 52.57 (0.65) | **30.76 (0.10)** | 65.90 (0.31) | 37.31 (0.07) | 40.45 (0.84) | 156.10 (0.85) | 51.82 (0.14) |
| | Rho-0.5 | **0.60 (0.01)** | 0.73 (0.00) | 1.20 (0.01) | **0.60 (0.00)** | 0.87 (0.01) | 0.98 (0.00) | 0.65 (0.00) | 1.03 (0.02) | 1.32 (0.00) |
| | Rho-0.9 | **0.27 (0.00)** | 0.51 (0.00) | 0.42 (0.01) | 0.29 (0.00) | 0.54 (0.00) | 0.30 (0.00) | 0.30 (0.02) | 1.51 (0.01) | 0.42 (0.00) |
| | CRPS | **65.78 (0.62)** | 94.73 (0.47) | 152.07 (0.95) | 72.34 (0.47) | 99.71 (0.46) | 137.32 (0.21) | 71.36 (0.69) | 216.12 (0.84) | 178.11 (0.73) |
| Colombia Dengue | SMAPE | **41.78 (1.15)** | 79.01 (2.35) | 45.74 (0.42) | 42.06 (0.69) | 50.59 (1.15) | 44.57 (0.27) | 47.54 (0.84) | 56.00 (0.85) | 45.17 (0.63) |
| | MAE | **18.02 (0.28)** | 26.60 (0.88) | 21.05 (0.25) | 18.48 (0.22) | 21.34 (0.38) | 19.91 (0.13) | 20.17 (0.36) | 24.93 (0.39) | 21.34 (0.12) |
| | RMSE | 51.89 (0.48) | 53.08 (1.71) | 53.32 (0.42) | **49.43 (0.35)** | 54.96 (0.52) | 52.15 (0.20) | 55.49 (1.09) | 65.66 (0.50) | 55.13 (0.23) |
| | MASE | **2.45 (0.04)** | 3.61 (0.12) | 2.86 (0.03) | 2.50 (0.03) | 2.90 (0.05) | 2.70 (0.02) | 2.74 (0.05) | 3.38 (0.05) | 2.90 (0.02) |
| | RMSSE | 3.41 (0.03) | 3.48 (0.11) | 3.50 (0.03) | **3.24 (0.02)** | 3.63 (0.03) | 3.42 (0.01) | 3.64 (0.07) | 4.31 (0.03) | 3.62 (0.02) |
| | Pinball-80 | **10.82 (0.18)** | 11.59 (0.35) | 11.75 (0.14) | 11.15 (0.11) | 13.89 (0.13) | 12.38 (0.06) | 11.78 (0.14) | 16.23 (0.13) | 13.45 (0.06) |
| | Rho-0.5 | 0.39 (0.01) | 0.51 (0.02) | 0.44 (0.01) | **0.38 (0.00)** | 0.44 (0.01) | 0.41 (0.00) | 0.42 (0.01) | 0.52 (0.01) | 0.44 (0.00) |
| | Rho-0.9 | 0.41 (0.01) | **0.38 (0.01)** | 0.45 (0.01) | 0.44 (0.00) | 0.47 (0.01) | 0.51 (0.00) | 0.48 (0.01) | 0.57 (0.01) | 0.57 (0.00) |
| | CRPS | **15.62 (0.20)** | 20.98 (0.43) | 17.71 (0.16) | 15.88 (0.12) | 21.31 (0.13) | 17.71 (0.10) | 17.22 (0.18) | 23.97 (0.13) | 19.45 (0.10) |
| Hungary Chickenpox | SMAPE | 83.83 (1.40) | 89.13 (1.56) | 91.46 (0.22) | 83.40 (0.55) | 88.92 (1.36) | 86.34 (0.28) | **82.74 (0.74)** | 86.65 (1.82) | 87.05 (0.17) |
| | MAE | 13.73 (0.13) | 15.17 (0.27) | 16.55 (0.10) | 13.90 (0.10) | 14.99 (0.28) | 14.78 (0.06) | **13.39 (0.09)** | 13.60 (0.29) | 14.38 (0.03) |
| | RMSE | **23.29 (0.11)** | 25.05 (0.39) | 24.60 (0.08) | 23.43 (0.12) | 23.65 (0.43) | 25.11 (0.05) | 23.74 (0.07) | 24.15 (0.44) | 23.57 (0.02) |
| | MASE | 0.71 (0.01) | 0.78 (0.01) | 0.85 (0.00) | 0.72 (0.01) | 0.77 (0.01) | 0.76 (0.00) | **0.69 (0.00)** | 0.70 (0.01) | 0.74 (0.00) |
| | RMSSE | **0.72 (0.00)** | 0.78 (0.01) | 0.77 (0.00) | **0.72 (0.00)** | 0.74 (0.01) | 0.78 (0.00) | 0.74 (0.00) | 0.75 (0.01) | 0.74 (0.00) |
| | Pinball-80 | **5.70 (0.04)** | 8.22 (0.10) | 5.89 (0.02) | 5.93 (0.03) | 8.18 (0.10) | 6.86 (0.02) | 6.13 (0.04) | 7.89 (0.10) | 6.16 (0.01) |
| | Rho-0.5 | 0.71 (0.01) | 0.79 (0.01) | 0.85 (0.00) | 0.71 (0.01) | 0.78 (0.01) | 0.76 (0.00) | **0.69 (0.00)** | 0.70 (0.01) | 0.74 (0.00) |
| | Rho-0.9 | **0.44 (0.00)** | 0.58 (0.01) | 0.46 (0.00) | 0.50 (0.01) | 0.59 (0.01) | 0.64 (0.00) | 0.53 (0.01) | 0.59 (0.01) | 0.57 (0.00) |
| | CRPS | **10.72 (0.09)** | 13.04 (0.10) | 13.95 (0.08) | 11.53 (0.06) | 12.81 (0.11) | 13.31 (0.04) | 11.27 (0.06) | 12.44 (0.09) | 13.64 (0.02) |

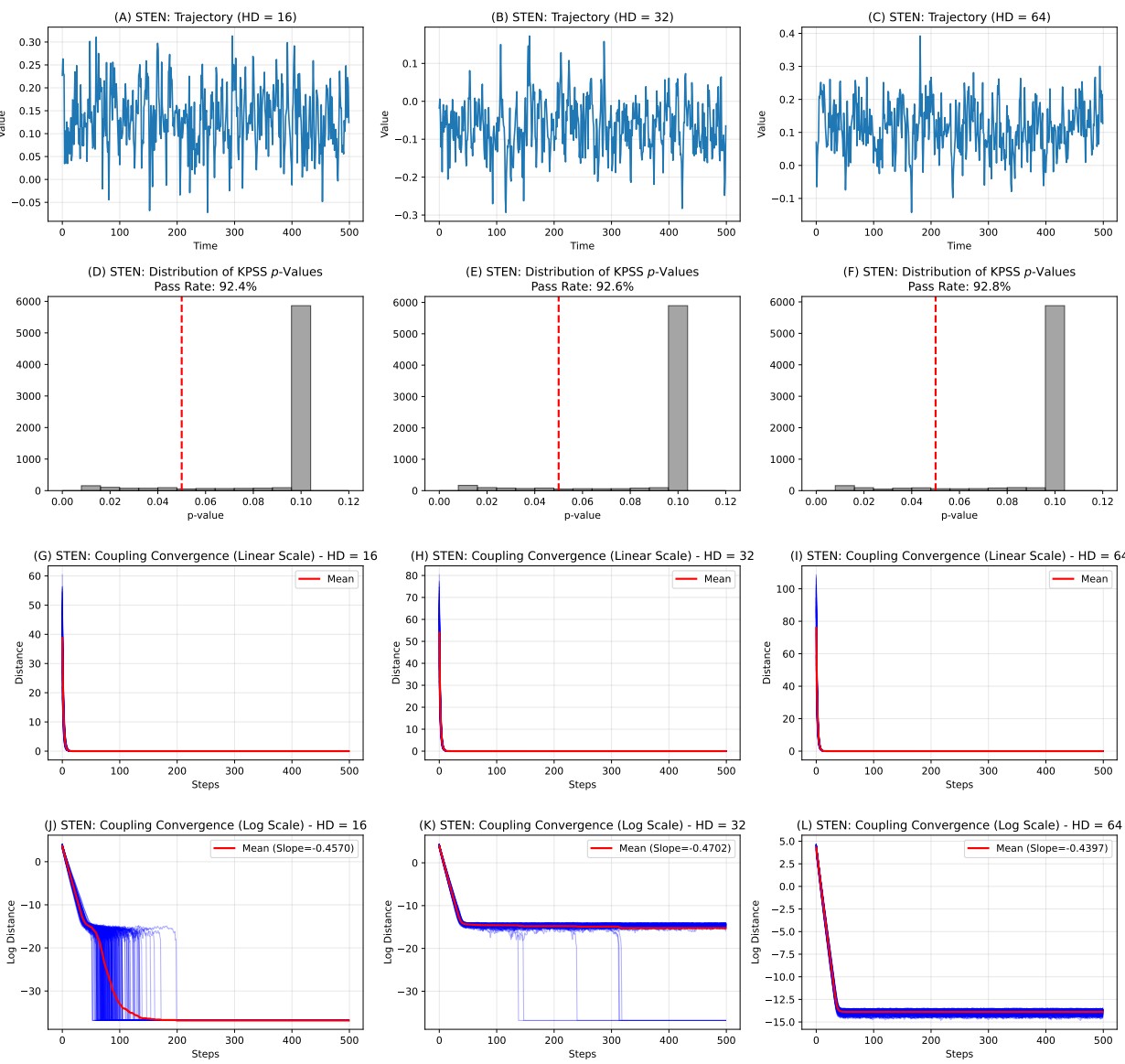

Fig. 15: Simulation results for STEN. For details, refer to Sec. 5.3.

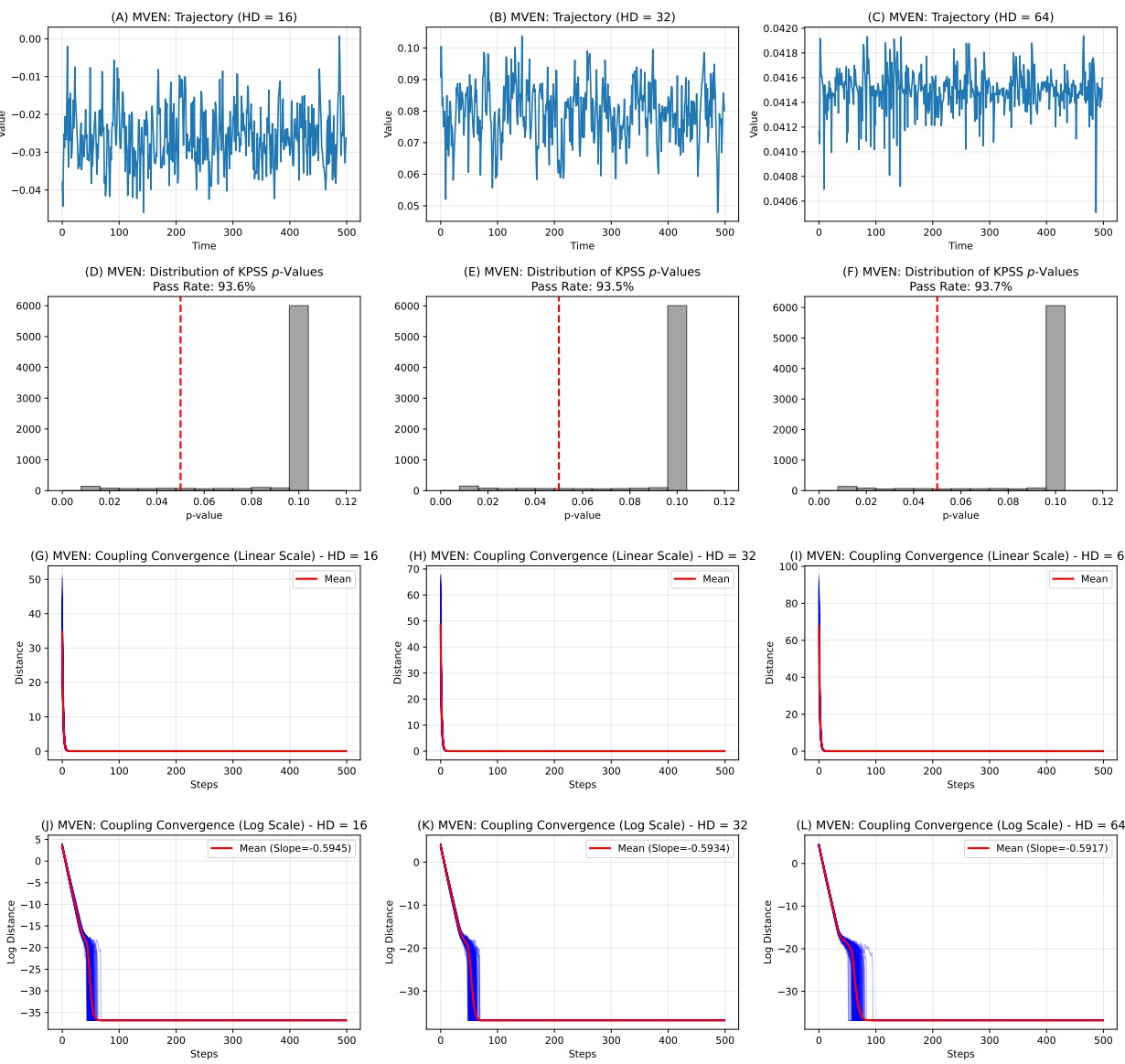

Fig. 16: Simulation results for MVEN. For details, refer to Sec. 5.3.

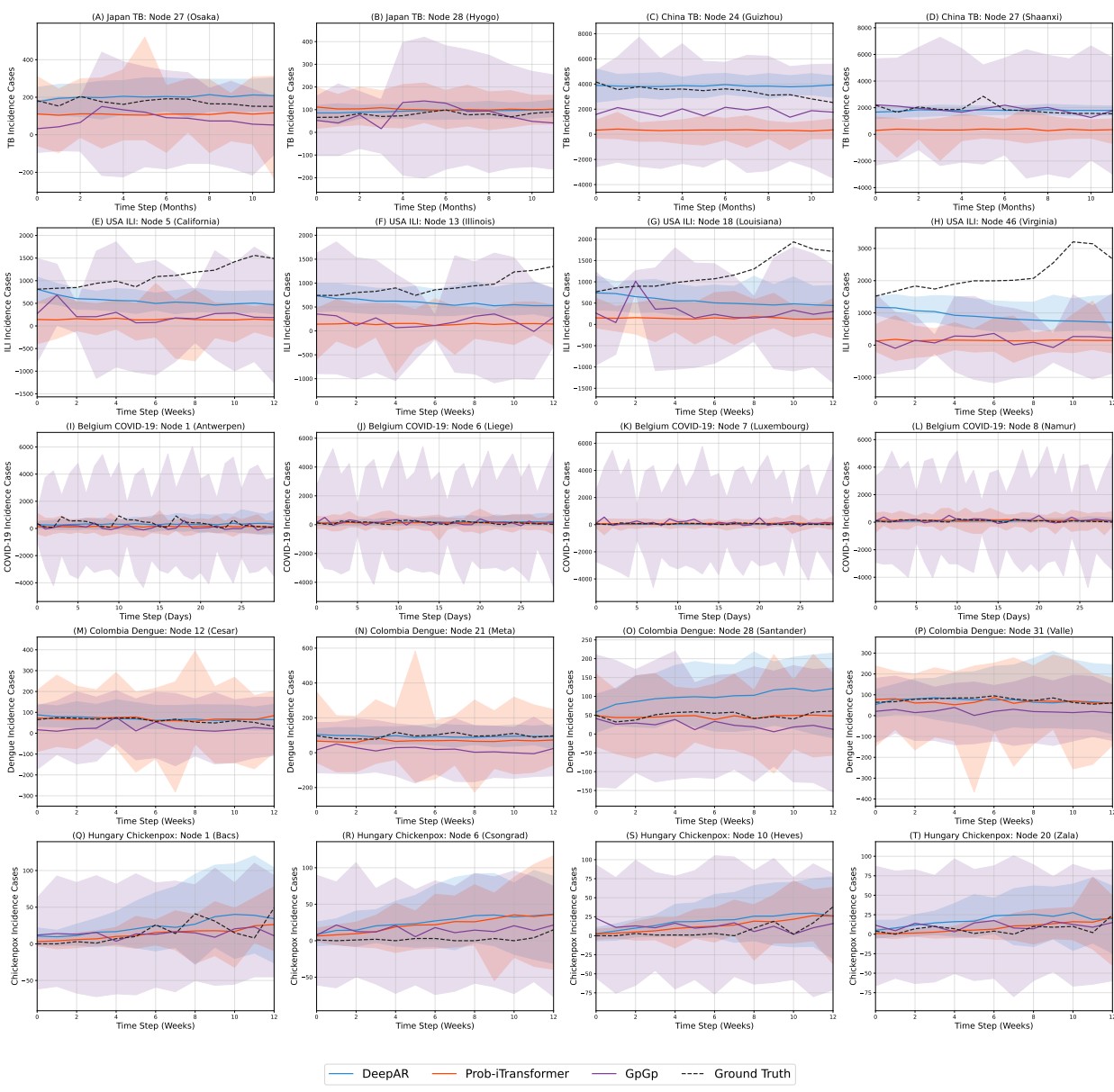

Fig. 17: Forecasts along with 95% PIs, produced by DeepAR, Prob-iTransformer, and GpGp on selected nodes (same as in Fig. 7) of the six datasets.

