# OpenReview forum: "Deep Generative Spatiotemporal Engression for Probabilistic Forecasting of Epidemics"
_TMLR — Accepted by TMLR_

### Review · Reviewer_4wMP · 2026-05-11

**Summary Of Contributions:**

The paper proposes three probabilistic forecasting frameworks for epidemic incidence data: a purely temporal model (MVEN) and two spatiotemporal models (GCEN and STEN). The paper builds on engression (Shen & Meinshausen, 2025) and LSTM-engression (Kraft et al., 2026), extending the framework from regression and temporal forecasting to multivariate and spatiotemporal epidemic forecasting. All three models use pre-additive noise injection and LSTM-based temporal modeling; GCEN and STEN additionally incorporate GCN-based and STAR-based spatial modules, respectively. The claimed contributions are: (i) three lightweight architectures that provide model-intrinsic uncertainty quantification; (ii) a theoretical analysis showing geometric ergodicity and asymptotic stationarity of the closed-loop processes; (iii) empirical evaluation on six epidemiological datasets at different temporal granularities, compared with 12 baselines; (iv) an interpretability analysis based on spatial-lag attribution from STEN's learned weights.

**Strengths**
- Practically useful empirical finding. The result that lightweight architectures can match or exceed heavier probabilistic spatiotemporal baselines (STESN, DiffSTG, GpGp) with substantially lower computational cost (Table 9) is a valuable contribution. This will be of direct interest to practitioners in probabilistic forecasting and applied epidemic modeling.
- Breadth of evaluation. The benchmark covers six datasets spanning airborne and vector-borne diseases, three temporal granularities, three horizons, twelve baselines, and ten metrics. This provides a useful benchmark resource regardless of one's view of the proposed methods.
- Theoretical contribution. The geometric ergodicity and asymptotic stationarity analysis (Theorems 1, 2 and Corollaries 1, 2) is a novel contribution relative to prior engression work (Shen & Meinshausen, 2025; Kraft et al., 2026), and the accompanying simulation in Section 5.3 is a nice empirical complement to the theory.
- Honest reporting of limitations. The authors are transparent about the underrepresentation of extreme peaks (Section 8) and report PIT calibration in Figure 8, which makes the overconfidence behavior visible to readers.
- Interpretability analysis. The STEN spatial-lag attribution (Section 7.2) is a clean and useful diagnostic that distinguishes the contribution of this paper from black-box baselines.

**Audience:**

Yes

**Audience Explanation:**

See Strengths above. The combination of competitive performance, lightweight compute, theoretical guarantees, and a broad benchmark makes the paper useful to both methodological and applied audiences in probabilistic forecasting and epidemic modeling.

**Broader Impact Concerns:**

The paper positions the methods as tools for public-health decision-making, so it would be worth being more explicit in the Broader Impact section about the downstream risks of overconfident prediction intervals. The PIT analysis in Figure 8 already shows that the proposed models tend toward narrow, somewhat overconfident intervals, and Section 6.4 notes this; bringing this observation into the Broader Impact discussion — and pairing it with a brief note that users should avoid interpreting them as fully calibrated uncertainty bounds — would make the section's framing more balanced. This is a small addition rather than a structural change.

**Claims And Evidence:**

Yes

**Claims Explanation:**

The central empirical and theoretical claims of the paper — favorable performance against baselines across six datasets and three horizons, geometric ergodicity of the closed-loop processes, and improved compute efficiency — are well supported by the experiments and proofs provided. The evidence base is broad and the reporting is honest about negative behavior (peak underrepresentation, PIT overconfidence).

My caveats concern the positioning and motivation rather than the experimental results themselves: (i) the description of engression in Section 3 conflates engression with distributional regression as a broader research area, and (ii) the motivating contrast in Section 2 is drawn against a narrow sub-class of post-additive models, while overlooking a property of engression (extrapolability) that may be more directly relevant to the present application. These are framing issues that the authors can address with revision; they do not undermine the core experimental conclusions. See Requested Changes for detail and constructive suggestions.

**Requested Changes:**

*Disclosure: I was not previously familiar with the engression literature before reviewing this paper. The comments below are based on my direct reading of the cited primary sources (Shen & Meinshausen, 2025; Kraft et al., 2026), not on independent expertise in this specific framework. I therefore frame the comments below as suggestions for clearer positioning rather than as factual corrections, and I welcome the authors' response where my reading may be incomplete.*

### Major
**M1. Clarify the description of engression in Section 3.**

The current Section 3 presents engression in a way that may understate its specific contribution relative to the broader distributional regression family. Two phrasings in particular could be clarified:
- On the goal of the method. The paper writes: "The primary goal of engression is to estimate the full conditional distribution of a target variable given a set of covariates… rather than focusing only on a specific functional such as the conditional mean or median." This is a fair description of distributional regression as a research area, but it is shared with mixture density networks (Bishop, 1994), quantile regression networks, normalizing flows, conditional VAEs/GANs, and DeepAR (which the paper uses as a baseline). Shen & Meinshausen (2025) themselves position engression as one member of the distributional regression family (their Section 5.2). It would help the reader if Section 3 first stated the shared goal of distributional regression, and then described what engression specifically contributes within that family.
- On the role of the pre-additive structure. The paper states "the pre-additive structure is what makes engression models generative." I read the authors' intended meaning here as referring to the way pre-additive noise enables input-dependent distributional shapes via non-linear propagation, which is a defensible functional argument. However, as written the sentence can be read as claiming that pre-ANMs are generative while post-ANMs are not, which is not quite accurate: a post-additive model Y = g(X) + η is also generative in the sampling sense. The original engression paper attributes a different property to the pre-additive structure: distributional extrapolability under monotonicity-type assumptions (Theorem 1 of Shen & Meinshausen, 2025; the paper's title is "Extrapolation through the Lens of Distributional Regression"). Rephrasing this sentence to distinguish (a) the source of generativity (latent noise + energy-score loss, shared with other generative models) from (b) what pre-ANM specifically buys (extrapolation guarantees, plus arguably greater distributional flexibility through non-linear propagation of noise) would make the paper's positioning clearer and would also connect directly to the discussion in M2.

A possible revised structure for Section 3: (a) what distributional regression as a goal is, and what existing methods (including DeepAR, MDN, normalizing flows) already achieve; (b) what engression specifically contributes within that family — namely, the energy-score loss formulation and the theoretical extrapolation properties under pre-additive noise. This would also make the paper's positioning relative to its own DeepAR baseline more transparent.

**M2. Revisit the framing in Section 2.**

The current motivation contrasts the proposed pre-additive approach against post-additive models that "presume that the stochasticity of the system is invariant across the state space" with "the shape of the distribution remains constant regardless of the input." This is accurate for the narrowest sub-class — fixed-variance Gaussian noise with a deterministic mean network — but input-dependent post-additive models such as MDNs (Bishop, 1994), heteroscedastic neural networks, and DeepAR (used as a baseline in this paper) do allow rich input-dependent predictive distributions. I would suggest tightening the framing so that the contrast is drawn against the appropriate sub-class, or, alternatively, against post-additive models specifically when paired with an unconditional Gaussian.

As a constructive direction the authors may or may not wish to pursue, I want to flag a potentially stronger motivation that is already available in the original engression paper. Shen & Meinshausen (2025) identify distributional extrapolability under monotonicity-type assumptions as the central theoretical property of engression. Epidemic peak forecasting — which the authors explicitly identify as their main remaining limitation in Section 8 — is structurally close to an extrapolation problem, since peak incidences often exceed values seen in training. The pre-ANM structure is therefore plausibly well-aligned with this application in a way that goes beyond what other distributional-regression methods offer.
I acknowledge that whether to invoke extrapolability as a primary motivation is the authors' call, and that there are perfectly reasonable papers that adopt a method without invoking its strongest theoretical property. As one data point that the authors may wish to consider: Kraft et al. (2026) — cited by the authors as motivation for adopting engression to time series — applied engression to hydrology with a standard temporal split and noted in their Discussion that "our results do not exhibit the stronger generalization for point prediction reported by the original engression study … the temporally distinct splits in training and validation still share relatively similar data distributions, limiting the need for powerful extrapolation capabilities." The present paper's experimental design follows a similar temporal-split pattern, although I note that the epidemiological datasets used here may exhibit more substantial distributional shift between training and test periods (e.g., the Belgium COVID-19 test period falls in a different epidemiological regime, and the Moran's I analysis in Figure 5 shows non-stationary spatial correlation). The authors are better placed than I am to judge whether the test sets here meaningfully probe extrapolation.

If the authors do choose to reframe (parts of) the motivation around extrapolability, Shen & Meinshausen (2025, Section 6.2) provides a fairly direct template: constructing tasks by partitioning on quantiles of the predictor and reporting test error specifically in the out-of-support region. In the epidemic setting an analogous design could train on pre-peak portions of an outbreak and test on the peak window, or hold out an entire wave. This is offered as a suggestion rather than a requirement.

### Minor
**m1. Figure 1B as a motivating figure.**

Figure 1B and the surrounding text in Section 2 have a small internal tension that the authors may wish to address. The first paragraph of Section 2 uses Figure 1B to contrast point-prediction methods (ARIMA, ARNN, LSTM) with the proposed probabilistic method, which is consistent with the figure's contents. The second paragraph then argues that post-additive probabilistic methods impose a fixed-shape uncertainty profile, but Figure 1B does not actually show any post-additive probabilistic baseline — only point predictors and the proposed model. The figure therefore cannot, on its own, demonstrate the limitation being attributed to post-additive probabilistic methods.
A small additional point: in the upper range of y_{t-1} (roughly 15–22) in Figure 1B, the proposed method's probabilistic cloud appears to under-cover the dispersion of the actual data points. I do not want to over-read a single illustrative figure, but if the figure is meant to motivate the choice of pre-additive engression, including at least one input-dependent post-additive probabilistic baseline (heteroscedastic LSTM, MDN-head LSTM, or DeepAR) on the same data would make Figure 1B align more directly with the second paragraph's argument. This is a presentational suggestion only; the core motivation of the paper does not rest on this single figure.

---

> ### Author Response · Authors · 2026-05-22
> **Response to Reviewer 4wMP**
>
> We sincerely thank Reviewer ‘4wMP’ for the invaluable review of our manuscript. Based on their constructive suggestions, we have carefully revised the manuscript as follows (updates are colored in red).
>
> M1. We have restructured Sec. 3 to properly situate engression within the broader distributional regression family.
>
> On the goal of the method: We now explicitly introduce the shared goal of distributional regression in Sec. 3, which is estimating the full conditional distribution $P(Y|X)$, and acknowledge established methods that achieve this, including Mixture Density Networks (MDNs), normalizing flows, DeepAR, and VAEs. We then clearly delineate engression’s specific contribution to this family: the combination of a pre-additive noise structure, the energy score loss formulation, and extrapolability.
>
> On the role of the pre-additive structure: We have revised and clarified that while post-ANMs are fundamentally generative in the sampling sense, the pre-additive structure provides greater distributional flexibility through the non-linear propagation of noise. Sampling from standard point-based post-ANMs would yield a rigid uncertainty profile, as discussed in Sec. 2. We emphasize in Sec. 3 on the use of pre-additive structure for (a) generativity and (b) to achieve highly flexible, non-parametric probabilistic forecasting.
>
> M2. We have updated Sec. 2 to draw a more precise contrast against the appropriate sub-classes of forecasting models:
>
> •	We explicitly state that standard point forecasting frameworks (e.g., AR(p), ARNN) utilize a post-additive noise structure that inherently constrains the predictive distribution, assuming stochasticity remains invariant across the state space.
>
> •	We now acknowledge post-ANMs such as DeepAR that allow for input-dependent predictive distributions. We contrast our approach against these models by highlighting their reliance on rigid parametric likelihood assumptions.
>
> •	We motivate our proposed approach by explaining that a pre-ANM acts as a non-parametric distributional lens, by injecting stochasticity prior to the nonlinear transformation. When coupled with the optimization of a strictly proper scoring rule, we mathematically ensure the predicted distribution converges to the true empirical distribution, offering a distinct advantage over DeepAR or MDNs.
>
> •	Moreover, the uncertainty profile of a post-ANM can also be shaped when trained with energy score loss. We compare this structure with pre-ANM in a new section on ablation study in Sec. 8.
>
>
> m1. We generated the plot by applying DeepAR (a post-additive probabilistic method) to the same data. We found that DeepAR’s prediction intervals fanned out excessively, spanning a massive range. Therefore, to preserve the clarity of Fig. 1, we have included the DeepAR plot along with other post-ANM fits in Fig. 1(B), and the pre-ANM MVEN fit separately in Fig. 1(C). While DeepAR allows for input-dependent variance, its reliance on parametric likelihoods often leads to uncalibrated, exploding uncertainty profiles. Fig. 1(B)-(C) now collectively motivates the use of pre-additive structure and energy score loss for stable probabilistic forecasting.
>
>
> Broader Impact Concern: We added the following line in the revised manuscript: ‘Sometimes, our proposed models produce narrow and somewhat overconfident intervals (Fig. 8). However, this behavior can be readily resolved by applying established recalibration techniques (Rumack et al., 2022), guaranteeing robust uncertainty quantification.’
>
>
> Regarding the direction on extrapolability:
>
> We are grateful for the insightful suggestion regarding extrapolability. While starting our work, we strongly considered framing our motivation around this. However, to the best of our knowledge, there is no well-established definition of ‘spatiotemporal extrapolability’ in the literature. Hence, applying Shen & Meinshausen’s specific definition of extrapolability (under an i.i.d. setup) to complex spatiotemporal domains is not straightforward. Secondly, the pre-additive structure is mathematically linked to extrapolation under monotonicity assumptions. In real-world spatiotemporal epidemic dynamics, monotonicity assumptions are violated; therefore, we do not claim formal spatiotemporal extrapolation guarantees.
>
> While temporal extrapolation structurally mirrors the challenge of extreme events and distribution shifts (like epidemic peaks), sequence-to-sequence architectures struggle to capture these peaks without explicitly integrated extreme-value handling mechanisms. Because our current models still exhibit some limitations in peak capture, we refrain from claiming extrapolability as a primary motivation. Instead, we have added a comprehensive discussion of this exact issue in the Limitations and Future Directions section, explicitly noting that analyzing spatiotemporal extrapolability via engression for extreme epidemic events is a highly promising avenue for future research.

---

### Review · Reviewer_v4zJ · 2026-05-14

**Summary Of Contributions:**

This paper proposes deep generative spatiotemporal engression models for probabilistic epidemic forecasting. The central idea is to use pre-additive noise, energy score training, and neural spatiotemporal architectures to generate forecast ensembles and model-intrinsic prediction intervals. The authors instantiate this idea in three models: MVEN, a multivariate temporal engression model; GCEN, which combines graph convolutional spatial embeddings with an LSTM-based engression forecaster; and STEN, which uses a STAR-inspired spatial layer designed to provide more interpretable spatial lag contributions.

Strengths:
- The problem is well motivated: epidemic forecasting requires uncertainty quantification, especially under low-frequency and scarce-data regimes.
- The proposed models are conceptually simple and computationally lightweight compared with several probabilistic spatiotemporal baselines.
- The experimental evaluation is broad, covering six real-world epidemic datasets, several forecast horizons, and both point and probabilistic metrics.

Weaknesses:
- The methodological contirbution is somewhat incremental: the models are largely combinations of existing components, including engression, LSTM, GCN, and STAR-style spatial aggregation, rather than a fundamentally new forecasting principle.
- The theoretical results are interesting but appear to apply to simplified closed-loop versions of the models under assumptions that may not closely match the actual trained/evaluated architectures.

**Audience:**

Yes

**Audience Explanation:**

The paper should be of interest to parts of the TMLR audience working on probabilistic forecasting, spatiotemporal modeling, uncertainty quantification, graph neural networks, and machine learning for public health.

**Broader Impact Concerns:**

The broader impact is potentially positive because probabilistic epidemic forecasting can support preparedness, resource allocation, and risk communication under uncertainty. The paper’s focus on prediction intervals and forecast ensembles is appropriate for high-stakes public-health settings.

**Claims And Evidence:**

No

**Claims Explanation:**

The empirical evidence is broad and generally convincing at the level of benchmark performance. The paper evaluates the proposed models on six epidemic datasets with different spatial scales and temporal resolutions, and compares against a large set of temporal, spatiotemporal, and probabilistic baselines.

The theoretical claims are mathematically valuable, but their practical relevance is less fully established. The geometric ergodicity and stationarity results are proved for simplified closed-loop recurrent processes and rely on assumptions such as bounded spatial transformations, contractive LSTM forget gates, and added vanishing noise to hidden/cell states. Since the practical implementation does not inject hidden-state noise and the empirical forecasting task is finite-sample, multi-step, and trained on real epidemic data, the theory should be presented as a stability result for an idealized model class rather than as direct evidence of reliable long-term epidemic forecasting. The authors should be more clear on this point.

**Requested Changes:**

- The authors should clarify the contribution of this paper more precisely. The paper should explicitly distinguish what is new relative to existing engression, LSTM-engression, GCN-LSTM, STARMA/STAR-inspired layers, and energy-score-based probabilistic forecasting.
- The theoretical section should be better aligned with the implemented models. The authors should clearly state which assumptions are verifiable or enforced in practice, especially the contractivity assumption, boundedness of spatial transformations, and the use of hidden/cell-state noise only in theory.
- The evaluation should include stronger ablation studies. At minimum, I would like to see ablations for pre-additive versus post-additive noise, energy score versus MSE or quantile loss, and spatial module versus no spatial module.
- Baseline fairness should be made more transparent. The paper should describe the hyperparameter search spaces, validation protocols, computational budgets, and whether each baseline was tuned comparably.

---

> ### Author Response · Authors · 2026-05-22
> **Response to Reviewer v4zJ**
>
> We sincerely thank Reviewer ‘v4zJ’ for the invaluable review of our manuscript. Based on their constructive suggestions, we have carefully revised the manuscript as follows (updates are colored in red).
>
> 1.	We clearly clarify our main contributions in the updated manuscript in the Introduction (pg. 3-4). Particularly, we emphasized the following:
>
> a.	This paper introduces the first spatiotemporal deep generative frameworks – MVEN, GCEN, and STEN, that integrate a pre-additive noise mechanism with energy score optimization for probabilistic forecasting, which is highly relevant for epidemic series. We propose spatiotemporal engression models, where existing engression-LSTM and GCN-LSTM models appear as special cases of our ‘more general’ models.
>
> b.	We provide a mathematical foundation for our proposed models by establishing the asymptotic stability and properties of closed-loop, pre-additive spatiotemporal recurrent networks. To the best of our knowledge, this is the first time asymptotic stationarity is studied for pre-additive noise-based spatiotemporal deep learning models.
>
> c.	We demonstrate that our models provide a highly efficient, lightweight alternative to resource-heavy probabilistic forecasting models (such as state-of-the-art diffusion models), making them useful for probabilistic forecasting on low-frequency epidemic data.
>
> d.	We provide a comprehensive benchmark resource by rigorously evaluating our models across six diverse epidemiological datasets.
>
>
> 2.	We explicitly explain each assumption and when/how they can be enforced in practice. We added two points in Remark 3 (pg. 14). To address the specific concerns of the reviewer, we state the following:
>
> a.	The spatial transformations can be made bounded by considering bounded activation functions, such as sigmoidal and tanh (as we had already mentioned post Assumption A6).
>
> b.	We refer to Jaeger (2001) for the justification behind adding small noise into the hidden/cell states to establish irreducibility in theory. In practice, the noise added to the spatial embeddings already propagates through the hidden states, so we do not add further noise in the hidden/cell states in the practical implementations.
>
> c.	The LSTM forget gate output lies in (0,1) because of sigmoidal activation, and naturally favors a contractive state update. Contractivity can be further enforced in practice through methods like standard input scaling (which we perform during our experiments).
>
> Hence, based on the reviewer’s suggestion, we have aligned the theoretical section with the implemented models. Indeed, the theoretical results are presented as a stability result for an idealized model class under mild assumptions, which can be enforced in practice. We have been more explicit regarding their practical relevance in Sec. 5.2 (pg. 14) of the updated manuscript.
>
>
> 3.	We have added a new section on ablation study (Sec. 8; pg. 26-27) in the revised manuscript, comparing:
>
> a.	The proposed models (with pre-additive noise) vs. their post-additive variants.
>
> b.	Training the proposed models with energy score loss vs. MSE loss.
>
> c.	The importance of the spatial module (GCEN and STEN vs. MVEN).
>
> The detailed results are presented in Tables 14-16 (Appendix G) in the updated manuscript. The results can be summarized by the MCB plot added in Fig. 12, which depicts that the proposed models are ‘statistically superior’ to the ablation variants, based on MAE (a point metric) and CRPS (a probabilistic metric). This justifies the pre-additive mechanism and energy score loss optimization for probabilistic forecasting. The importance of the spatial module is also reflected in Fig. 12, with GCEN and STEN outperforming MVEN on the CRPS and MAE metrics. To check the calibration of the ablation variants, we also added PIT Q-Q plots in Fig. 14 (Appendix G); a detailed discussion is provided in Sec. 8.
>
>
> 4.	To ensure absolute fairness and eliminate user tuning bias, we strictly adhered to the configurations provided by the recently used benchmark and foundational papers published in venues such as JMLR (Darts) and JRSS-C (STESN). This protocol ensures that all baselines are evaluated under their intended, reproducible out-of-the-box configurations. The input and output sequence lengths were kept uniform across all architectures, including our proposed models. The computational resources for all models are identical (GPU P100 for the DL models) and mentioned in Appendix E.1 (Implementation Details). In the revised manuscript (pg. 45-46), we have added the necessary details to Appendix E.1, describing the fixed architectures, validation protocols, and the computational budget utilized for all experiments.
>
>
> We thank the reviewer for their constructive and insightful comments. We believe that incorporating their suggestions has greatly improved the manuscript.

---

### Review · Reviewer_bwPu · 2026-05-15

**Summary Of Contributions:**

This paper studies probabilistic forecasting for epidemic spatiotemporal data. Instead of only giving one point forecast, the paper aims to generate multiple possible future trajectories and use them to estimate uncertainty. The authors build on engression and propose three models: MVEN, GCEN, and STEN. The paper evaluates these models on six epidemic datasets and argues that proposed methods can improve both point forecasting and probabilistic forecasting.

**Audience:**

Yes

**Audience Explanation:**

Yes. I think this paper would be of interest to part of the TMLR audience, especially researchers working on time-series forecasting, probabilistic forecasting, spatiotemporal modeling, and generative models.

**Claims And Evidence:**

Yes

**Claims Explanation:**

Yes, to a reasonable extent. The paper provides a fairly broad empirical evaluation across multiple epidemic datasets and compares against a range of temporal, spatiotemporal, and probabilistic forecasting baselines. The results generally support the claim.

**Requested Changes:**

Pros: \
1: The problem is important and well motivated. For epidemic forecasting, point forecasts alone are often not enough, and probabilistic forecasts with uncertainty ranges are more useful for public health planning. \
2: The model design is clear. MVEN, GCEN, and STEN form a clear family of models, moving from temporal-only modeling to graph-based spatial modeling. \
3: The evaluation is broad. The paper tests methods on six epidemic datasets with different temporal granularities and forecast horizons, which makes the empirical study more convincing.

Cons: \
1: The paper is quite long and sometimes hard to follow. Several model descriptions repeat similar ideas, such as noise injection, LSTM forecasting, and ensemble generation. A more concise presentation would make the contribution easier to understand. \
2: My main concern is that the methodological novelty feels somewhat not enough. The proposed models mainly combine existing ingredients: engression-style noise, LSTM temporal modeling, and either graph convolution or spatial aggregation. This combination is useful, but the paper should be clearer about the novelty design, not just like a combination here. \
3: The paper proposes three related models, but it is not always clear how a practitioner should choose among them before running full experiments. The empirical results show that the best choice is like dataset-dependent. A more concrete model-selection discussion would be useful.

---

> ### Author Response · Authors · 2026-05-22
> **Response to Reviewer bwPu**
>
> We sincerely thank Reviewer ‘bwPu’ for the invaluable review of our manuscript. Based on their constructive suggestions, we have carefully revised the manuscript as follows (updates are colored in red).
>
> 1.	We appreciate the reviewer’s suggestion to make the presentation more concise. We have condensed the last paragraph of STEN (Sec. 4.2) in the updated manuscript to avoid redundancy in explaining the noise injection and temporal processing parts, which are already explained in detail in the preceding section (GCEN). Please note that the main manuscript is within 29 pages (excluding references) with relevant details/explanations/results for readers put into the appendix.
>
> 2.	We clearly clarify our main contributions in the updated manuscript in the Introduction (pg. 3-4). Particularly, we emphasized the following:
>
> (a)	This paper introduces the first spatiotemporal deep generative frameworks – MVEN, GCEN, and STEN that integrate a pre-additive noise mechanism with energy score optimization for probabilistic forecasting, which is highly relevant for epidemic series. Therefore, we propose spatiotemporal engression models, where existing engression-LSTM and GCN-LSTM models appear as special cases of our ‘more general’ models;
>
> (b) We provide a mathematical foundation for our proposed models by establishing the asymptotic stability and properties of closed-loop, pre-additive spatiotemporal recurrent networks. To the best of our knowledge, this is the first time asymptotic stationarity is studied for pre-additive noise-based spatiotemporal deep learning models;
>
> (c) We demonstrate that our models provide a highly efficient, lightweight alternative to resource-heavy probabilistic forecasting models (such as state-of-the-art diffusion models), making them useful for probabilistic forecasting on low-frequency epidemic data;
>
> (d) We provide a comprehensive benchmark resource by rigorously evaluating our models across six diverse epidemiological datasets.
>
> 3.	We fully agree that a concrete model selection discussion is essential. We discussed the model-selection strategy in Remark 1 (pg. 11) of Sec. 4. Empirically, GCEN demonstrates the strongest overall predictive performance across our benchmark datasets, driven by its sophisticated graph convolutional layers. Although GCEN works well for medium to large spatiotemporal datasets, these graph operations function as a black box, masking the explicit pathways of spatial dissemination. STEN is preferable when structural explainability is important and data is scarce. Its learned weights directly isolate and quantify the specific lag contributions of spatial neighbors to disease transmission (Sec. 7.2), delivering actionable epidemiological insights alongside reliable probabilistic forecasts. Lastly, as a purely temporal architecture, MVEN serves as the optimal selection in scenarios where reliable network topology or spatial data is sparse, poorly resolved, or entirely unavailable.
>
> Once again, we thank the reviewer for their constructive and insightful comments. We believe that incorporating their suggestions has greatly improved the manuscript.

---

### Comment · Reviewer_Kh1u · 2026-04-30

The authors argue for 'engression' (which I admit I'd not heard of before but which also seems new based on the references), which involves pre-additive noise (which also does not seem particularly new as a concept) to do probabilistic forecasting, and include geo-spatial information.

I read the paper with interest and optimistic curiosity, although have a number of lingering questions on my mind.

1) Figure 1B is somehow not convincing and I'm not totally sure what the authors are trying to convey here. Firstly, there is no ground truth interval in this plot, so there is no way to know if the model's output is reasonable. This might not be an issue were it not for the fact that the underlying data gives little clue as to what the output should look like. Similarly, the expected values of all the other models are completely different, which also makes the plot a bit confusing, if the goal is to convey a top-level demonstration of what the new approach provides which the others do not.

2) The pre-additive approach reminds me immediately of variational approaches. However, whilst variational approaches often constrain the distribution and variance of the injected noise as part of the loss function (e.g. KL between inferred latent and prior), I'm struggling to see how the injected noise in this case is used in a 'principled' way.  I see that the sharpness term controls variability so that the output distribution (across samples) broadly matches the distribution of observed points, but this seems very heuristic / implicit. Perhaps I'm missing something and there is a way to link the energy function to an explicit likelihood function - could the authors confirm?

3) regarding the proofs - I'm glad there are included but (a) I haven't checked them and (b) (more importantly)   I'm worried the authors overstate their relevance to real-world problems.  Are the authors confident that the assumptions associated with these proofs translate into most real-world scenarios?

4) the authors seem to evaluate on the same objective they are training with, which is often valid, but in this case it feels a bit circular beacuse the metric hasn't (AFAIK) been defined elsewhere as a standard for evaluating forecasting. As such, to propose a new metric, optimise for that metric, and then say your method is better than other methods on that metric, seems like a form of goalpost setting.   I think to make it more convincing there should be some eevaluation of distirbution matching / coverage.


Otherwise, i appreciated the breadth of evaluations (different diseases , spatial structures, time granularities), and I also felt like the paper was easy to read and very thorough.

---

> ### Author Response · Authors · 2026-05-11
> **Response to Reviewer 'Kh1u'**
>
> We sincerely thank the learned reviewer for their invaluable comments on our manuscript. We provide point-by-point responses to the comments below (References are from the main manuscript):
>
> 1.	The primary goal of Figure 1 is to illustrate why relying on a single conditional quantity (such as the mean in point forecasting) is fundamentally inadequate for highly stochastic epidemic dynamics and why a distributional approach is required. Fig. 1B and 1A are in-sample lag plots (current week $y_t$ vs. previous week $y_{t-1}$) rather than a standard time-series plot, and the grey scatter points themselves represent the ground truth. A vertical slice at any fixed $y_{t-1}$ empirically reveals the true conditional distribution of the future state $y_t$, with the point spread defining the empirical ground truth interval. Our proposed MVEN generates the light orange cloud (Fig. 1(B)), covering this empirical spread better than the point-based baselines and capturing heteroskedasticity in the data. Since a single input ($y_{t-1}$) maps to highly varied outputs ($y_t$), point-based models struggle to estimate a stable conditional mean (hence, the difference in expected values of baselines). Conversely, MVEN avoids a rigid fit, acting instead as a distributional lens. MVEN’s expected value (solid orange line; the cloud's median) naturally tracks the underlying scatter point density, while pre-additive noise generates the full ensemble of plausible states (light orange cloud). In Sec. 2, we visualize current vs. lag-1 values to motivate our deep engression approach over point-based forecasters.
>
> 2.	While both variational and engression methodologies model complex distributions, the former require a tractable parametric likelihood and an explicit prior. In contrast, engression minimizes the energy score (ES) loss, a mathematically proven strictly proper scoring rule (Gneiting and Raftery, 2007). A strictly proper score is uniquely optimized iff the predictive distribution exactly matches the true underlying distribution; therefore, minimizing the empirical ES loss provides a mathematical guarantee: the network will utilize pre-additive noise to shape the forecast ensemble until it converges to the true empirical distribution. Thus, we provide a rigorous non-parametric alignment rather than a heuristic approximation. Regarding the link between the energy function and an explicit likelihood: standard MLE is simply a special case of ‘optimum score estimation’ where the optimized strictly proper scoring rule is the logarithmic score. Using logarithmic score (and thus MLE) can be highly unstable/intractable for nonparametric continuous distributions, but we bypass the need to force spatiotemporal data into restrictive parametric likelihoods using ES loss.
>
> 3.	The theoretical guarantees of geometric ergodicity (GE) and asymptotic stationarity hold substantial practical relevance for public health agencies and decision-makers. They ensure our forecasts remain stable and reliable even under high volatility. This robustness is critical for decision-making under uncertainty, as forecast instability or divergence could lead to severe adverse policy and public health consequences. Regarding the assumptions, we extend Zhao et al. (2020)’s work on the GE of temporal LSTMs for the pre-additive noise model (spatiotemporal setup). GE for LSTMs requires bounded and continuous output and activation functions (sigmoid and tanh); our assumptions on network weights and noise align with practical deployment. Finally, while real-world deployment is open-loop, analyzing the closed-loop (autonomous) system is the mathematical standard for proving stability. By proving closed-loop GE, we guarantee the model's core engine is inherently non-explosive and exponentially forgets initial conditions. Real-world implications are detailed and verified through closed-loop simulations in Sec. 5.2-5.3 (page 13).
>
> 4.	We would like to point out that the CRPS is a popularly used strictly proper scoring rule generalizing MAE to probabilistic forecasts (Gneiting and Raftery, 2007) and serves as a special case of the broader ES. Also, our evaluation encompasses 10 distinct metrics, including five point and multiple probabilistic metrics (Sec. 6.2), across which our architecture consistently outperforms baselines. To ensure robust generalization, models are trained by optimizing the ES loss (Q2 answer), hyperparameters are tuned by minimizing CRPS on a strictly held-out validation set, and final reported results are derived from an independent, temporally separated test set (avoiding data leakage). Lastly, we provide PIT Q-Q plots (Fig. 8, page 22) and report the empirical coverage of 95% prediction intervals (Table 10, page 44). For a rigorous assessment, we evaluate these intervals using the Winkler Score, which simultaneously rewards desired coverage and heavily penalizes excessively wide intervals, providing an honest assessment of predictive uncertainty.

---

### Decision · Action_Editor_a22t · 2026-06-15

**Recommendation:** Accept as is

**Audience:**

Yes

**Audience Explanation:**

Yes. The paper addresses an important problem—probabilistic spatiotemporal forecasting of epidemics—and combines methodological innovation with extensive empirical validation. Researchers working on probabilistic forecasting, spatiotemporal modeling, epidemiology, and uncertainty quantification are likely to find the proposed methods and theoretical results of interest.

**Claims And Evidence:**

Yes

**Claims Explanation:**

Yes. The paper provides both theoretical and empirical evidence to support its claims. It establishes geometric ergodicity and asymptotic stationarity of the proposed models, and extensive experiments across six epidemic datasets demonstrate consistent improvements in both point and probabilistic forecasting performance over strong baselines.

---

> ### Author Response · Authors · 2026-06-20
> **Response to Action Editor a22t**
>
> We sincerely thank the action editor and learned reviewers for their constructive comments. We believe that incorporating their suggestions have greatly improved the presentation of this work. We have submitted the camera-ready version for your final approval.